# Investigating Advanced Reasoning of Large Language Models via Black-Box Interaction

## Abstract

Existing tasks fall short in evaluating reasoning ability of Large Language Models (LLMs) in an interactive, unknown environment. This deficiency leads to the isolated assessment of deductive, inductive, and abductive reasoning, neglecting the integrated reasoning process that is indispensable for humans discovery of real world. We introduce a novel evaluation paradigm, *black-box interaction*, to tackle this challenge. A black-box is defined by a hidden function that maps a specific set of inputs to outputs. LLMs are required to unravel the hidden function behind the black-box by interacting with it in given exploration turns, and reasoning over observed input-output pairs. Leveraging this idea, we build the Oracle benchmark which comprises 6 types of black-box task and 96 black-boxes. 19 modern LLMs are benchmarked. o3, a leading LLM from OpenAI, ranks first in 5 of the 6 tasks, achieving over 70% accuracy on most easy black-boxes. But it still struggles with some hard black-box tasks, where its average performance drops below 40%. Further analysis indicates a universal difficulty among LLMs: They lack the high-level planning capability to develop efficient and adaptive exploration strategies for hypothesis refinement.

## 1 Introduction

Reasoning constitutes a fundamental component of artificial general intelligence (AGI), allowing systems to solve complex problems, adapt to unknown environment, and make decisions with human-like cognitive flexibility. With techniques like long chain-of-thought and test-time scaling (Chen et al., 2025), large language models (LLMs) (OpenAI, 2025b; Anthropic, 2025; Guo et al., 2025) have demonstrated remarkable reasoning ability in some challenging benchmarks (Cobbe et al., 2021; Hendrycks et al., 2021). However, skepticism has persistently shadowed the claim that LLMs possess reasoning ability akin to that of humans.

Charles Peirce's framework (Peirce, 1934) posits that human's discovery of unknown environment is guided by a dynamic reasoning cycle encompassing deduction, induction, and abduction. As depicted in Figure 1, this cycle begins with forming a hypothesis from observations (abduction), proceeds to planning to derive new observations (deduction), and concludes with hypothesis refinement against new observations (induction). However, existing reasoning datasets and benchmarks fall short in placing LLMs in an interactive, unknown environment (Fodor, 2025). This shortcoming leads to evaluating reasoning in an isolated manner, rather than an integrated, holistic process (Suzgun et al., 2022; Mondorf & Plank, 2024). Some researches (Costarelli et al., 2024; Hu et al., 2024) employ games to simulate interactive, unknown environments. This approach presents two key limitations. First, the extensive training data of LLMs raises the possibility that they are already familiar with the game strategy, compromising the validity of testing reasoning in *unknown* environment. Second, it conflates the evaluation of reasoning with other abilities like spatial understanding and long-context understanding, preventing it from serving as a pure reasoning benchmark.

To address the aforementioned challenge, we introduce *black-box interaction*, a novel evaluation paradigm for investigating the integrated, human-like reasoning capability of LLMs, which we term advanced reasoning. This paradigm models an unknown environment by constructing a black-box based on specific hidden rules. LLMs are required to uncover the hidden rules behind a black-box via multiple turns of exploration. Specifically, black-box is defined as a hidden function $f : \mathcal{X} \rightarrow \mathcal{Y}$, mapping input $\mathcal{X} = \{x | P(x)\}$ that satisfies predicate $P$ to output $\mathcal{Y}$. LLMs are instructed to interact

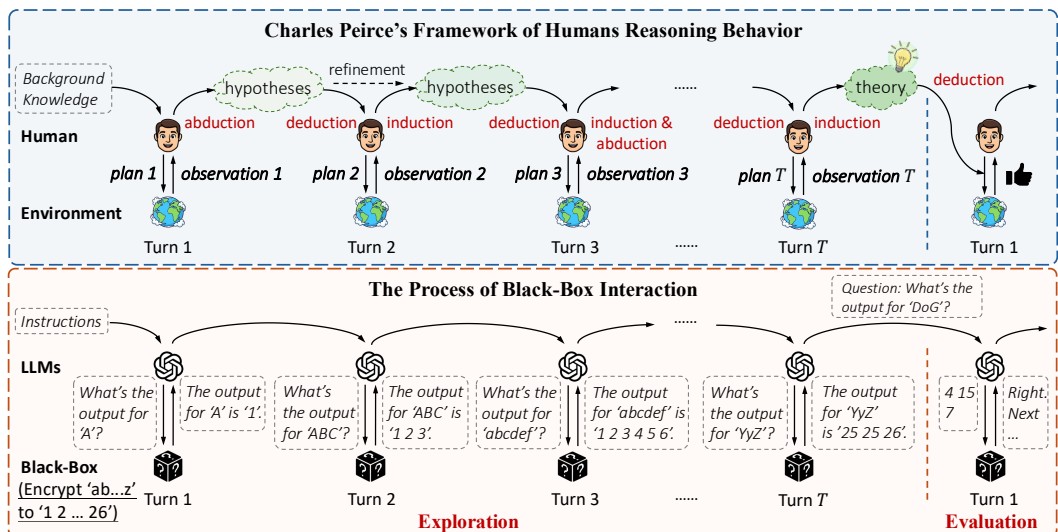

Figure 1: Illustration of Charles Peirce's framework of human reasoning behavior (upper) and an example of the process of black-box interaction (lower). In this example, the black-box represents an encryption method that maps English letters to numbers.

with the black-box. The interaction is two-stage: (i) In the exploration stage, LLMs can freely feed any valid input $x$ to the black-box, and will receive corresponding feedback $f(x)$. (ii) The evaluation stage starts after reaching given maximum exploration turns. LLMs' understanding of the black-box is evaluated by comparing their output with the black-box's output on a set of unseen test samples. Figure 1 illustrates the complete process of black-box interaction, where the black-box represents an encryption method that maps English letters to numbers.

The practical implementation of a black-box only involves mapping inputs to outputs based on hidden rules. This simplicity allows it to be generalized across various environments. To facilitate and accelerate the generalization of black-boxes to any scale, type, and level of difficulty, we design a fully automated agentic framework for black-box construction. Three LLM-based modules collaborate to accomplish black-box construction from scratch only with natural language description. It handles everything, including the generation of test samples, black-box code, and interactive interface between LLMs and black-box. Leveraging this framework, we build the ORACLE [1] benchmark, which considers 6 types of black-box task: **Code Intent Inference**, **Circuit Rule Inference**, **Physics System Inference**, **Encryption Rule Inference**, **Interactive Puzzle Inference**, **Game Strategy Inference**. The 6 tasks take code, boolean circuit, mechanical system, encryption method, interactive puzzle, opponent's game strategy as black-box respectively. Current benchmark consists of 96 black-boxes, 51 of them are easy black-boxes and 45 are hard.

We evaluate 19 leading proprietary and open-weight LLMs. Overall, reasoning models perform better than chat models. OpenAI o3 delivers the best performance, ranking first in 5 out of 6 tasks under 10 exploration turns and 4 out of 6 tasks under 20 exploration turns. Furthermore, it achieves an average accuracy exceeding 70% on most easy black-boxes and approximately 40% on most hard ones. Further analysis reveals a critical and universal weakness of LLMs: They lack the high-level planning capability required to develop efficient and adaptive exploration strategies. This deficiency in reasoning prevents effective hypothesis refinement, which consequently compromises the ability to understand complex black-box mechanisms under limited exploration.

The contributions of this paper can be summarized as follows:

1. We introduce a novel evaluation paradigm, black-box interaction, for investigating advanced reasoning of LLMs (Section 2). This paradigm addresses several critical concerns in the field of reasoning dataset and benchmark design (Section 7).

---

[1] The name is inspired by some mythologies that an ORACLE only returns "yes" or "no" to questions, thus challenging the questioner's intelligence.

2. Leveraging the idea of black-box interaction, we build the ORACLE benchmark which comprises 6 types of black-box tasks and a total of 96 black-boxes (Section 3, Appendix G).

3. We propose an effective automated agentic framework which only requires natural language description to generate diverse black-boxes (Section 4). The framework greatly facilitates the scaling of the ORACLE benchmark.

4. Comprehensive experiments and analysis are conducted to investigate the performance and behavior of LLMs in black-box interaction. We identify that LLMs struggle to develop efficient and adaptive exploration strategies. (Section 5, Appendix D, E).

## 2 PRELIMINARIES

We formally define the task setting of evaluating advanced reasoning of LLMs via black-box interaction. A complete black-box interaction process comprises two sequential stages: an **exploration** stage and an **evaluation** stage.

**Black-Box**  A black-box is a rule-based system characterized by a hidden function $f : \mathcal{X} \to \mathcal{Y}$ that maps an input domain $\mathcal{X}$ to an output domain $\mathcal{Y}$. The input domain $\mathcal{X}$ is a set of elements $\mathcal{X} = \{x | P(x)\}$ that satisfy a specific predicate $P$. In some situations, $f$ can be decomposed into a composition of multiple mappings, resulting in intermediate outputs. We define this more formally as follows: Let $\mathcal{Y}_0 = \mathcal{X}$. The composite function $f$ is given by

$$f = f_n \circ f_{n-1} \circ \cdots \circ f_1, \tag{1}$$

where each component function $f_i : \mathcal{Y}_{i-1} \to \mathcal{Y}_i$ for $i = 1, \ldots, n$. The intermediate outputs reside in the sets $\mathcal{Y}_1, \ldots, \mathcal{Y}_{n-1}$, and the final output codomain is $\mathcal{Y} = \mathcal{Y}_n$.

**Model**  The model, denoted by $M$, is a system that processes and generates natural language. Model $M$ is instructed to interact with the black-box. Its action space is $\mathcal{X} = \{x | P(x)\}$. We focus on Transformer-based large language models in the scope of this paper.

**Exploration**  The exploration stage consists of a sequence of interactions between model $M$ and black-box $f$ over $T$ turns. In each turn $t \in \{1, \ldots, T\}$, the model adaptively generates a query $x^t \in \mathcal{X}$ and submits it to the black-box. It then receives the corresponding feedback $y^t = f(x^t) \in \mathcal{Y}$. In some scenarios, the model observes intermediate feedback $y_i^t \in \mathcal{Y}_i$ instead of $y^t$. The query $x^t$ is generated based on the history of all previous interactions. Let the history at turn $t$ be $H_{t-1} = (x^1, y^1, \ldots, x^{t-1}, y^{t-1})$. The model generates the next query as:

$$x^t = M(H_{t-1}), \quad \text{for } t > 1. \tag{2}$$

The initial query, $x^1$, is generated based on the initial task description provided to the model. Upon completion, this stage yields a total exploration history $H_T = (x^1, y^1, \ldots, x^T, y^T)$.

**Evaluation**  Following the exploration stage, the model's reasoning ability is evaluated. This evaluation stage runs for $K$ turns, corresponding to the size of a test set $\mathcal{X}_{\text{test}} = \{x_{\text{test}}^1, \ldots, x_{\text{test}}^K\}$. The test set is disjoint from the set of queries used during exploration, i.e., $\mathcal{X}_{\text{test}} \cap \{x^1, \ldots, x^T\} = \emptyset$. In each turn $k \in \{1, \ldots, K\}$, the model $M$ is given a test sample $x_{\text{test}}^k$ and needs to produce a prediction, denoted as $\hat{y}^k$. The black-box then provides feedback by comparing the prediction to the true output $f(x_{\text{test}}^k)$. This feedback is a binary correctness signal, $c^k = \mathbf{1}(\hat{y}^k = f(x_{\text{test}}^k))$, where $\mathbf{1}(\cdot)$ is the indicator function. The prediction at turn $k$ is generated as:

$$\hat{y}^k = M(H_T, x_{\text{test}}^1, \hat{y}^1, c^1, \ldots, x_{\text{test}}^{k-1}, \hat{y}^{k-1}, c^{k-1}, x_{\text{test}}^k), \quad \text{for } 1 \leq k \leq K. \tag{3}$$

This evaluation setup allows the model to continue to learn and adapt its strategy based on feedback received on its test-time performance.

## 3 THE ORACLE BENCHMARK

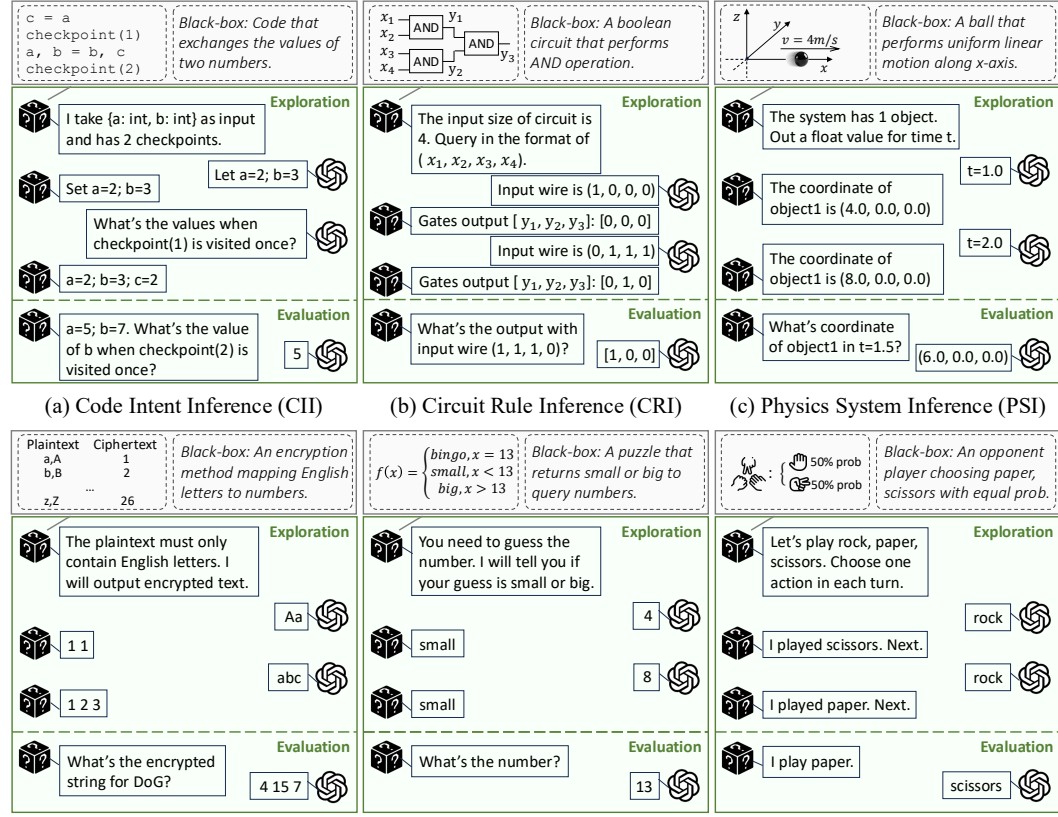

(a) Code Intent Inference (CII)    (b) Circuit Rule Inference (CRI)    (c) Physics System Inference (PSI)

(d) Encryption Rule Inference (ERI)    (e) Interactive Puzzle Inference (IPI)    (f) Game Strategy Inference (GSI)

Figure 2: Examples of 6 different types of black-box tasks in the ORACLE benchmark.

The composition of ORACLE benchmark is shown in Figure 3. Each task consists of a mix of easy and hard black-boxes. The inner ring of the pie chart indicates the total number of black-boxes for each task, while the outer ring breaks down these black-boxes into easy and hard categories. The current benchmark includes 6 tasks and 96 black-boxes (51 easy, 45 hard).

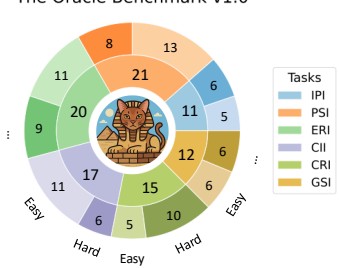

Figure 3: The composition of ORACLE benchmark.

### 3.1 TASK DESIGN

We reveal the methodology behind the design of 6 different black-box tasks. In Figure 2, a simple black-box from each task is selected to facilitate understanding. The examples cover both exploration and evaluation stages. Detailed implementations of all black-boxes in the ORACLE benchmark are reported in Appendix G. Some complete black-box interaction cases are shown in Appendix D.2. Test samples for each task are detailed in Appendix A.2.

**Code Intent Inference (CII)**  A black-box $f$ represents a code algorithm that maps input variables $x$ to output variables $y$. Following the definition in Equation (1), $f$ is further decomposed into $f_i$ which is named checkpoint in this task. A checkpoint $f_i$ captures the values of all current accessible variables. These checkpoints are strategically placed where significant changes to variable values occur. For LLMs, two types of actions are allowed: (i) Assign any valid value $x$ as input variable. (ii) Ask for the value of accessible variables at selected checkpoint $f_i$. Action (i) must be completed before action (ii), and action (ii) is formatted in $(i, iter)$, where $i$ is the index of selected checkpoint, and $iter$ is the visited times of the $i$-th checkpoint (e.g., within a loop). For example, $(3, 2)$ indicates the third checkpoint being visited for the second time. The goal of LLMs is to understand the algorithm. When evaluation starts, LLMs are required to output the value of questioned variable at certain checkpoint with unseen input variables.

**Circuit Rule Inference (CRI)**  A black-box $f$ represents an acyclic boolean circuit that only contains AND, OR, NOT gates. It maps input wire $x$ which is a fixed number 0/1 bits to circuit gates' output $y$. The black-box will first inform the size $n$ of input wire. Then in each turn, LLMs are supposed to output $x = (x_1, x_2, \ldots, x_n), x_i \in \{0, 1\}$ as query. After LLMs' query, the black-box will return the output of every circuit gate in the format of $y = [y_1, y_2, ..., y_m], y_i \in \{0, 1\}$, where $m$ is the number of gates and $y_i$ is the output of the $i$-th gate. The goal of LLMs is to understand the function and composition of circuit. When evaluation starts, LLMs are required to give every circuit's output with unseen input wires.

**Physics System Inference (PSI)**  A black-box $f$ represents a classical mechanical system that maps time point $x$ to objects' coordinates $y$. In each turn, LLMs need to assign value $x$ for time point as input, and the black-box will return the 3-dimensional coordinates $y$ of all objects in the mechanical system at time $x$. The goal of LLMs is to understand the mechanical system. When evaluation starts, LLMs are required to calculate all the objects' coordinates at unseen time points.

**Encryption Rule Inference (ERI)**  A black-box $f$ represents an encryption process that maps plaintext $x$ to ciphertext $y$. LLMs can assign any valid plaintext $x$ as input, and black-box will return corresponding ciphertext $y$ as output. The goal of LLMs is to understand how the encryption method works based on the plaintext-ciphertext pairs. When evaluation starts, LLMs are required to output the corresponding ciphertext given unseen plaintext.

**Interactive Puzzle Inference (IPI)**  A black-box $f$ represents an interactive puzzle with a hidden answer. The puzzle maps player query $x$ to result $y$ based on the puzzle rule. The LLMs can interact with the puzzle for multiple turns in the exploration stage. When evaluation starts, LLMs are required to figure out the right hidden answer of the puzzle.

**Game Strategy Inference (GSI)**  Unlike the IPI task, the GSI task involves a two-player game. In this setup, LLMs participate as one player, facing a black-box opponent that performs a fixed game strategy. In this sense, the black-box $f$ represents a strategy that maps game observation $x$ to action $y$. Unlike previously introduced tasks, the GSI task requires a model to go beyond simply understanding the black-box: a model must devise a strategy to outperform it. When evaluation starts, LLMs will face the same black-box opponent and aim to achieve as higher score as possible. Since some games are not round-independent, one exploration turn in GSI indicates playing a $n$-round game once. Then LLMs will be evaluated in the game with the same number of rounds.

### 3.2 Evaluation Metrics

Two metrics, accuracy and turn@shot, are used to measure the reasoning ability of LLMs in black-box interaction. Following the definitions in Section 2, the accuracy for each black-box is calculated via $acc = \sum_{k=1}^{K} c^k / K$, where $K$ is the number of test samples and $c^k$ measures the correctness of LLMs' answer. Specifically, accuracy in GSI task is measured by the ratio of actual score to the optimal strategy score. Turn@shot consists of two aspects. Turn denotes to the number of interaction turns for exploration, and shot indicates the number of allowed attempts for each test sample during evaluation. For example, 20@2 means the exploration stage lasts for 20 turns, and a model has 2 chances to answer each test sample in evaluation. The best model is supposed to achieve the highest accuracy with the lowest turn@shot.

## 4 Framework for Automatic Black-Box Generation

We introduce the agentic framework to generate diverse black-boxes for the ORACLE benchmark. As illustrated in Figure 4, the framework comprises three LLM-powered modules: a Coding LLM for initial creation of platform code, a Test LLM for interaction simulation, and a Refinement LLM for iterative debugging. The framework operates through the following three stages.

**Platform Code Generation**  Platform code refers to the complete code for conducting black-box interaction, covering the implementation of black-box and interactive interface between LLMs and black-box. Leveraging the powerful coding capabilities of LLMs, we directly instruct a Coding

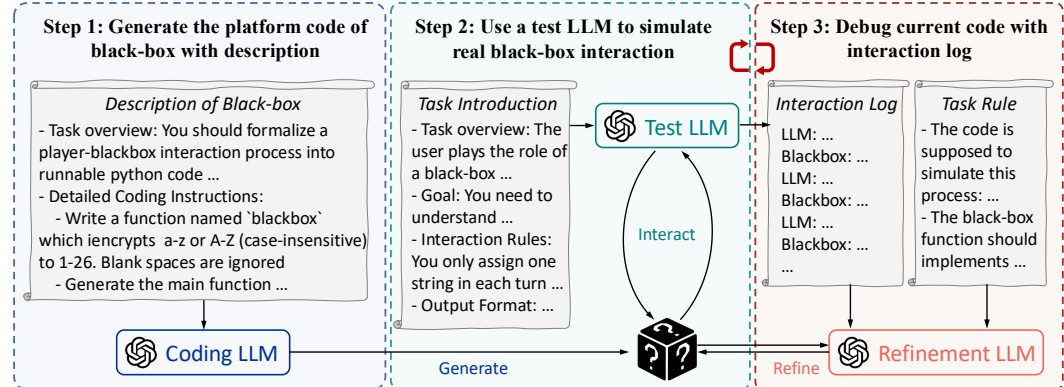

Figure 4: The framework for black-box generation, which is used to build the ORACLE benchmark. All related prompts are detailed in Appendix F.1.

LLM with prompt to generate the platform code. The prompt is twofold, encompassing natural language description of the black-box and the interaction rule. Since the interaction rule remains constant for certain type of black-box task, scaling up the benchmark simply involves describing the new black-boxes in natural language.

**Simulation**    When initial platform code is generated, a Test LLM is used to interact with the black-box to simulate real interaction scenarios. This simulation covers both exploration and evaluation stage, and will result in three situations: (i) The platform code contains errors and fails to be executed. (ii) The platform code executes, but the black-box functionality is not correctly implemented as described. (iii) The platform code is correct and the simulation runs successfully as expected. In either case, an interaction log will be produced when the simulation process ends.

**Iterative Debugging**    A Refinement LLM is used to check the correctness of generated platform code by combining the interaction log and task rule. The simulation process will result in three situations as mentioned above. For situation (i), Refinement LLM is instructed to produce revised platform code based on current code and error messages. For situation (ii), Refinement LLM is first instructed to figure out the inconsistency between current black-box implementation and its expected functionality with interaction log and task rule as prompt. Then, it's instructed to revise current platform code based on the discovered inconsistency. The simulation step will be conducted again when the platform code is revised. This iterative debugging process continues until the platform code is deemed correct by the Refinement LLM (situation (iii)).

The framework design operationalizes two principles from human cognition and software engineering: (i) Mastery through interaction, akin to learning a game by playing rather than just reading instructions. (ii) Debugging via runtime feedback, where code is refined based on its observed behavior rather than static analysis. This interactive, closed-loop process is key to generating high-fidelity code, and drastically facilitates the construction and scaling of the ORACLE benchmark.

## 5 EXPERIMENT AND ANALYSIS

### 5.1 BENCHMARKED MODELS AND BASELINE TEST

We benchmark a series of proprietary and open-weight models. Proprietary LLMs include GPT-series models (gpt-4o-mini, gpt-4o, gpt-4.1-mini, gpt-4.1, o1, o3-mini, o3, o4-mini), Claude-series models (claude-3.5-haiku, claude-3.5-sonnet, claude-3.7-sonnet, claude-4-sonnet), Gemini-series models (gemini-1.5-pro, gemini-2.0-flash, gemini-2.5-flash, gemini-2.5-pro), Qwen-series models (qwen-plus, qwen-max, qwq-plus). Open-weight LLMs include DeepSeek-series models (deepseek-v3-671b, deepseek-r1-671b), Llama-series models (llama-4-scout-17b-16e, llama-4-maverick-17b-128e), Qwen-series models (qwq-32b, qwen3-32b, qwen3-235b-a22b). See Appendix A for a complete list of models and implementation details. Some LLMs can perform extended thinking (i.e., reasoning). Both those with and without this capability are tested.

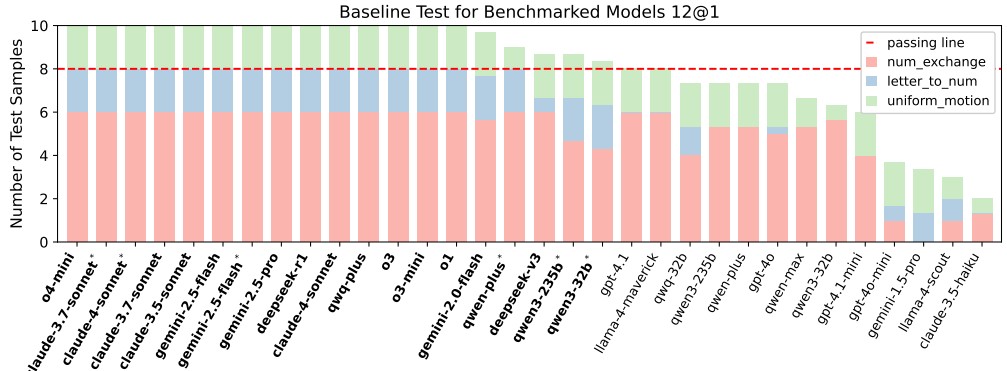

Figure 5: Baseline test for benchmarked models under 12@1. Models superscripted with ∗ indicate extended thinking enabled. Qualified models are marked in bold.

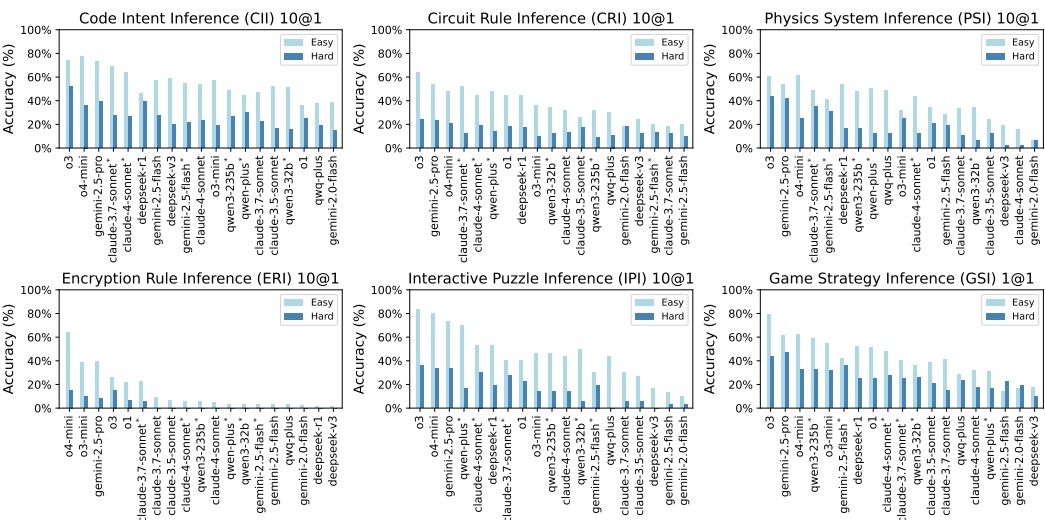

Figure 6: Performance of LLMs in six tasks of the ORACLE benchmark 10@1&1@1.

To qualify for the ORACLE benchmark, models must first pass a baseline test that contains 3 black-boxes from CII, ERI, PSI task, containing a total of 10 test samples. Detailed implementations of the 3 black-boxes are shown in Figure 2 (a), (c), (d). Turn@shot is set as 12@1, indicating 12 interaction turns for exploration and 1 chance for answering each test sample. The baseline test is conducted for three separate times and the averaged performance is reported in Figure 5. Models must achieve over 80% accuracy to qualify for the ORACLE benchmark. 19 out of 32 benchmarked models are qualified, including o1, o3-mini, o3, o4-mini, claude-3.5-sonnet, claude-3.7-sonnet, claude-3.7-sonnet_thinking, claude-4-sonnet, claude-4-sonnet_thinking, gemini-2.0-flash, gemini-2.5-flash, gemini-2.5-flash_thinking, gemini-2.5-pro, deepseek-v3, deepseek-r1, qwen-plus_thinking, qwq-plus, qwen3-235b-a22b_thinking, qwen3-32b_thinking.

## 5.2 OVERALL BENCHMARK RESULT

Two experiment settings, 10@1 and 20@2, are applied for the ORACLE benchmark. Figure 6 and Figure 7 report results on easy and hard black-boxes per task. Models are ranked by the sum of their accuracy on easy and hard black-boxes. Generally speaking, models exhibit similar rankings across 6 tasks. o3, o4-mini, gemini-2.5-pro, claude-3.7-sonnet_thinking, and claude-4-sonnet_thinking achieve competitive performance on all six tasks, and o3 ranks first among all models. When it comes to open-source models, deepseek-r1 achieves the top overall performance. Latest models (e.g., gemini-2.5-flash) perform better than old models (e.g., gemini-2.0-flash). Reasoning models (e.g., claude-4-sonnet_thinking) perform better than conventional chat models (e.g., claude-4-sonnet). While best performing LLMs boast over 80% accuracy on some easy black-box tasks, they

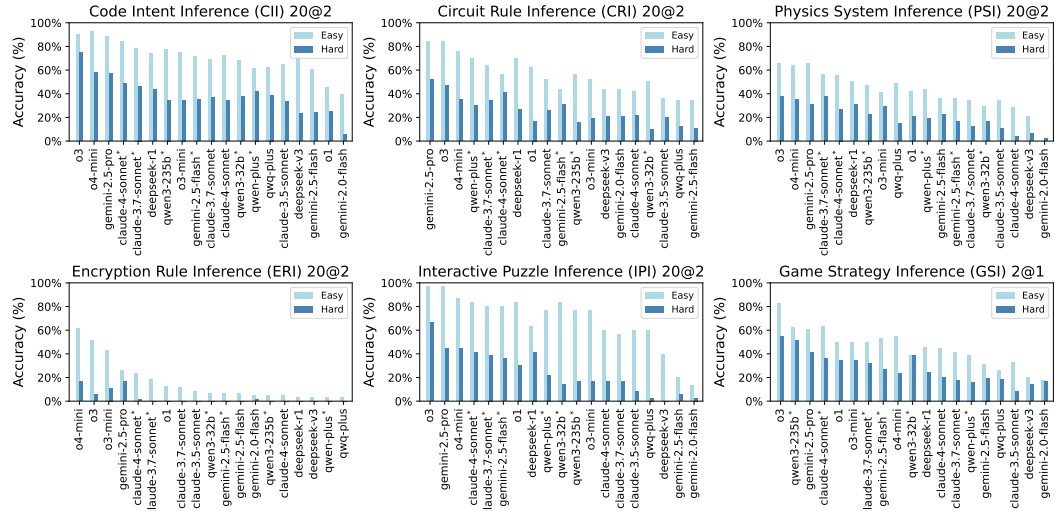

Figure 7: Performance of LLMs in six tasks of the ORACLE benchmark 20@2&2@1.

still struggle with harder ones, where their accuracy is typically less than half that of their performance on easy tasks.

## 5.3 ANALYSIS

We aim to reveal a key weakness of modern LLMs in black-box interaction: **They struggle to develop efficient and adaptive exploration strategies.** This deficiency in high-level planning highlights LLMs' shortcomings in deductive and inductive reasoning. To substantiate this claim, we analyze the performance gains from increased exploration turns, present a comparative experiment, and examine the exploration behaviors of leading LLMs. Additional case analysis, ablation study, and weaknesses of LLMs are detailed in Appendix D, E, and C.4 respectively.

**Analysis on performance gains from more exploration** Model performance is expected to increase when exploration turns (from 10 to 20) and evaluation attempts (from 1 to 2) are extended. However, as shown in Figure 8, the averaged performance of LLMs improves by over 10% in CII, CRI, and IPI tasks, but it shows negligible improvement in PSI, ERI, and GSI tasks. While the limited progress on the PSI task is mainly due to the poor computing ability of LLMs (detailed in Appendix C.4), the lack of improvement on the ERI and GSI tasks highlights a fundamental weakness of LLMs: They are not good at developing efficient exploration strategy in some scenarios. We also find the performance gains is greater for easy black-boxes compared to hard ones, and this phenomenon becomes especially obvious when it comes to less capable LLMs. An example in Figure 8 indicates that the accuracy increase

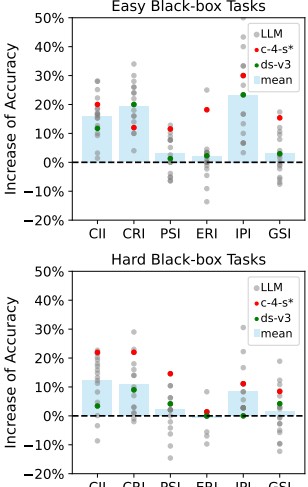

Figure 8: Averaged accuracy increase of 19 LLMs across 6 tasks when turn@shot is extended from 10@1&1@1 to 20@2&2@1.

of deepseek-v3 remains near zero in hard black-boxes, while claude-4-sonnet_thinking can still obtain great improvement. This suggests advanced models can devise and execute superior exploration strategies compared to less capable models.

**Comparative experiment on adaptive exploration strategy optimization** Apart from developing an efficient strategy, LLMs are supposed to keep optimizing a strategy adaptively based on instant feedback from black-box to narrow action space and maximize information gained from each turn, which is called adaptive exploration (Patrascu & Stacey, 1999). However, we find that even SOTA LLMs still lack a high-level planning ability to optimize exploration strategies. A comparative experiment with two settings is designed for verification. Setting (i): mod-

| Setting (i) CRI 10@1 | Setting (ii) CRI 10@1 | Setting (i) ERI 10@1 | Setting (ii) ERI 10@1 |
|---|---|---|---|
| o4-mini: (0,0,0,0,0,0,0,0) | o4-mini: (0,0,0,0,0,0,0,0) | gemini-2.5-pro: A | gemini-2.5-pro: a |
| o4-mini: (0,0,0,0,0,0,0,0) | o4-mini: (1,0,0,0,0,0,0,0) | gemini-2.5-pro: a | gemini-2.5-pro: b |
| o4-mini: (1,0,0,0,0,0,0,0) | o4-mini: (0,1,0,0,0,0,0,0) | gemini-2.5-pro: B | gemini-2.5-pro: c |
| o4-mini: (0,1,0,0,0,0,0,0) | o4-mini: (1,1,0,0,0,0,0,0) | gemini-2.5-pro: b | gemini-2.5-pro: z |
| o4-mini: (0,0,1,0,0,0,0,0) | o4-mini: (0,0,1,0,0,0,0,0) | gemini-2.5-pro: C | gemini-2.5-pro: Hello |
| o4-mini: (0,0,0,1,0,0,0,0) | o4-mini: (0,0,0,1,0,0,0,0) | gemini-2.5-pro: c | gemini-2.5-pro: word |
| o4-mini: (0,0,0,0,0,0,0,1) | o4-mini: (0,0,0,0,1,0,0,0) | gemini-2.5-pro: Z | gemini-2.5-pro: book |
| o4-mini: (1,1,1,1,1,1,1,1) | o4-mini: (0,0,0,0,0,1,0,0) | gemini-2.5-pro: z | gemini-2.5-pro: cat |
| o4-mini: (0,0,0,0,1,0,0,0) | o4-mini: (0,0,0,0,0,0,1,0) | gemini-2.5-pro: Hello | gemini-2.5-pro: in |
| o4-mini: (0,0,0,0,1,0,0,0) | o4-mini: (0,0,0,0,0,0,0,1) | gemini-2.5-pro: Apple Bee | gemini-2.5-pro: banana |
| ...... | ...... | ...... | ...... |
| **Final Accuracy: 0%** | **Final Accuracy: 0%** | **Final Accuracy: 0%** | **Final Accuracy: 0%** |

Figure 9: Cases of exploration behavior under two different settings. Black-box responses and evaluation stages are neglected. Red text indicates the same exploration behavior.

els will not receive black-box feedback in each turn. Instead, all the queries and corresponding ing answers will be announced in the last exploration turn; Setting (ii) serves as a control group, where models will receive instant black-box feedback in each turn. Ideally, model performance under setting (ii) is supposed to be higher than setting (i), as models can keep optimizing exploration strategy with instant feedback, but they have to maintain a fixed strategy in setting (i).

We select three powerful LLMs, gemini-2.5-pro, o3-mini, and o4-mini, and evaluate them in two representative tasks, CRI and ERI, which most challenge models' ability in optimizing exploration strategy. Results are shown in Figure 10. The three LLMs exhibit remarkably consistent performance across the two settings, providing strong evidence of their inability to optimize exploration strategies effectively. To further investigate the exploration strategy of LLMs under two different settings, we show some cases of LLMs' exploration behavior in Figure 9. These cases come from LLMs' interaction with two easy black-boxes from CRI and ERI task ("Xor Sequence" and "Zigzag Cipher", detailed in Appendix G). First, we find both models adopt inefficient exploration strategies: In the CRI task, o4-mini employs an exhaustive, in-order strategy. In the ERI task, gemini-2.5-pro resorts to querying single English letter or word. Second, both models fail to adaptively optimize exploration strategy. Their reasoning behavior remains largely consistent across two settings, indicating that they cannot effectively exploit the real-time feedback from the black-box. Consequently, both models achieve zero accuracy in evaluation.

Building on the analysis above, we categorize the capacity to devise exploration strategy into three tiers. **Tier 1**: Model can not develop a planned exploration strategy, and explore in a random approach. **Tier 2**: Model can develop a relatively efficient exploration strategy but fail to optimize it adaptively. **Tier 3**: Model can adaptively optimize their exploration strategy based on instant feedback, de-

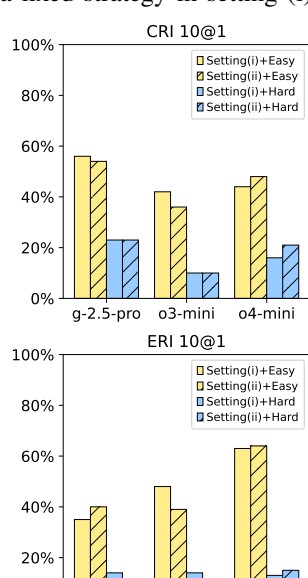

Figure 10: Model performance in CRI 10@1 and ERI 10@1 under two settings.

veloping a nearly optimal approach. Most LLMs operate at Tier 1. Best-performed reasoning LLMs achieve Tier 2 in some situations. Tier 3 is the domain of human according to Charles Peirce's theory and primitive human evaluation (Appendix E.7). We have not yet identified any LLM that can achieve Tier 3 of adaptive strategy planning.

# 6 RELATED WORK

## 6.1 MODELING INTERACTIVE ENVIRONMENT

Building interactive environment that simulates real-world settings has always been a heated research topic. Prior works in reinforcement learning build online (Brockman et al., 2016) and offline (Fu et al., 2020) environment to investigate models' ability of strategic learning. Recent progress

in evaluating LLMs and LLM agents has adopted various methods to model interactive environment. For example, WebArena (Zhou et al., 2023) creates an environment with fully functional websites that contain tools and external knowledge bases. Fish et al. (2025) builds stationary and non-stationary economic environment. Wu et al. (2023); Costarelli et al. (2024); Hu et al. (2024); Park et al. (2025) employ text games (e.g. Akinator) or video games (e.g. MineCraft) as environment and evaluate LLMs' reasoning ability through game-playing. Ma et al. (2024) propose AgentBoard which contains web, tool, embodied AI, and game tasks as partially observable environments.

### 6.2 REASONING DATASETS AND BENCHMARKS

Evaluating reasoning ability of LLMs is an active area of research, especially with the recent development of reasoning large language models (Chen et al., 2025). In the field of deductive reasoning, several datasets and benchmarks are developed for measuring complex mathematics (e.g. GSM8k (Cobbe et al., 2021), MATH (Hendrycks et al., 2021), AIME, AMC23, Omni-MATH (Gao et al., 2024), FrontierMath (Glazer et al., 2024), OlympiadBench (He et al., 2024)), coding (e.g. Code-Contests (Li et al., 2022), SWEbench (Jimenez et al., 2023), LiveCodeBench (Jain et al., 2024)), and logic (e.g. BIGBench Hard (Suzgun et al., 2022), LiveBench (White et al., 2024), ARC (Chollet, 2019), ZebraLogic (Lin et al., 2025)). Datasets for inductive reasoning include DEER (Yang et al., 2022), ConceptARC (Moskvichev et al., 2023), Mirage (Li et al., 2024), InductionBench (Hua et al., 2025), Abductive reasoning datasets include ART (Bhagavatula et al., 2020), CauseLogics (He & Lu, 2024). Some researches like UniADILR (Xia et al., 2025) seek to evaluate deductive, inductive, and abductive reasoning in one framework.

## 7 DISCUSSION

The design of black-box interaction can bring additional benefits. First, it addresses the critical concern of data contamination (Roberts et al., 2023; Deng et al., 2023), which refers to the leakage of datasets and benchmarks into LLMs' training data, thus hindering the discrimination of whether LLMs truly reason or just memorize (Magar & Schwartz, 2022; Zhang et al., 2022; Dziri et al., 2023; Wu et al., 2024; Balloccu et al., 2024). Black-box interaction naturally generate dynamic context as input. The inherent invisibility of black-box also ensures zero data contamination, even if LLMs are highly acquainted with its practical implementation. We also conduct an ablation study (Appendix E.3) which directly adds relevant knowledge as prompt for verification.

Second, it facilitates the evaluation process. Previous works (Turpin et al., 2023; Hao et al., 2024; Mondorf & Plank, 2024) find LLMs can generate correct answers with logically incorrect reasoning paths. Thus evaluations on most outcome-based datasets and benchmarks (Cobbe et al., 2021; Hendrycks et al., 2021) become less convincing. In the scenario of black-box interaction, the interaction history naturally reflects the reasoning path of LLMs, and an incorrect reasoning path will not lead to correctness in all test samples. So evaluation on test samples is a reliable approach.

Third, black-box interaction simulates many concrete real-world applications. For example, in the designed six tasks in the ORACLE benchmark, CII models code reverse engineering or deciphering legacy code via breakpoint debugging. CRI models hardware reverse engineering or integrated circuit (IC) analysis. PSI models scientific discovery and the scientific method. ERI models cryptanalysis under the Chosen-Plaintext Attack (CPA) setting. IPI models interactive problem-solving. GSI models opponent modeling and devising counter-strategies within game-theoretic scenarios.

## 8 CONCLUSION

We introduce black-box interaction, a novel paradigm for interactively evaluating the advanced reasoning of LLMs, and propose the corresponding ORACLE benchmark. This benchmark features 6 task designs and 96 black-boxes to evaluate 19 modern LLMs. The ORACLE benchmark is highly adaptable, allowing for easy scaling to any scale, task, and difficulty level through the a robust agentic generation framework. We also provide deep insight into the reasoning behavior and shortcomings of current LLMs in uncovering the hidden rules behind the black-box.

## REPRODUCIBILITY STATEMENT

We strive to make sure all the results reported in this paper are reproducible. The source code has been submitted as supplementary materials.

## THE USE OF LARGE LANGUAGE MODELS (LLMS)

Large Language Models (LLMs) are only used for polish writing in this paper. Specifically, LLMs are exclusively used to rewrite individual sentences based on original human text to improve grammar and wording. The authors of this paper are responsible for the contents of their submission.

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

APPENDIX

Table 1: Model code/API of benchmarked models.

| Model Name | Model Type | API Access |
|---|---|---|
| GPT-4o-mini (Hurst et al., 2024) | Proprietary | `gpt-4o-mini-2024-07-18` |
| GPT-4o (Hurst et al., 2024) | Proprietary | `gpt-4o-2024-08-06` |
| GPT-4.1-mini (OpenAI, 2025a) | Proprietary | `gpt-4.1-mini-2025-04-14` |
| GPT-4.1 (OpenAI, 2025a) | Proprietary | `gpt-4.1-2025-04-14` |
| o1 (Jaech et al., 2024) | Proprietary | `o1-2024-12-17` |
| o3-mini (OpenAI, 2025b) | Proprietary | `o3-mini-2025-01-31` |
| o3 (OpenAI, 2025b) | Proprietary | `o3-2025-04-16` |
| o4-mini (OpenAI, 2025b) | Proprietary | `o4-mini-2025-04-16` |
| Claude-3.5-haiku (Anthropic, 2024) | Proprietary | `claude-3-5-haiku-20241022` |
| Claude-3.5-sonnet (Anthropic, 2024) | Proprietary | `claude-3-5-sonnet-20241022` |
| Claude-3.7-sonnet (Anthropic, 2025) | Proprietary | `claude-3-7-sonnet-20250219` |
| Claude-4-sonnet (Anthropic, 2025) | Proprietary | `claude-sonnet-4-20250514` |
| Gemini-1.5-pro (GeminiTeam et al., 2024) | Proprietary | `gemini-1.5-pro` |
| Gemini-2.0-flash (GeminiTeam et al., 2024) | Proprietary | `gemini-2.0-flash` |
| Gemini-2.5-flash (Comanici et al., 2025) | Proprietary | `gemini-2.5-flash` |
| Gemini-2.5-pro (Comanici et al., 2025) | Proprietary | `gemini-2.5-pro` |
| DeepSeek-v3-671b (Liu et al., 2024) | Open-weight | `deepseek-reasoner` |
| DeepSeek-r1-671b (Guo et al., 2025) | Open-weight | `deepseek-chat` |
| Llama-4-scout-17b-16e (Touvron et al., 2023) | Open-weight | `meta-llama/llama-4-scout` |
| Llama-4-maverick-17b-128e (Touvron et al., 2023) | Open-weight | `meta-llama/llama-4-maverick` |
| Qwen-max (Yang et al., 2024) | Proprietary | `qwen-max` |
| Qwen-plus (Yang et al., 2024) | Proprietary | `qwen-plus-latest` |
| Qwen3-235b-a22b (Yang et al., 2024) | Open-weight | `qwen3-235b-a22b` |
| Qwen3-32b (Yang et al., 2024) | Open-weight | `qwen3-32b` |
| QwQ-32b (QwenTeam, 2025) | Open-weight | `qwq-32b` |
| QwQ-plus (QwenTeam, 2025) | Proprietary | `qwq-plus` |

## A  IMPLEMENTATION DETAILS

### A.1  DETAILS OF LLMS

Some key hyper-parameters of LLMs for results reported in baseline test and ORACLE benchmark are set as follows: The temperature for all benchmarked models is set as 0. The reasoning effort for GPT-series LLMs (o1, o3-mini, o3, o4-mini) is set as medium. The token budget for extended thinking in Claude-series LLMs (claude-3.7-sonnet_thinking, claude-4-sonnet_thinking) is set as 20000. For Gemini-series LLMs (gemini-2.5-flash_thinking, gemini-2.5-pro), the thinking budget is set as dynamic. For Qwen-series LLMs (qwen-plus_thinking, qwen3-32b_thinking, qwen3-235b-a22b_thinking, qwq-plus), the thinking budget is set as 20000 tokens. For deepseek-r1, the length of thinking content can not be modified. So full thinking is allowed. The experiments are conducted from 4 July to 19 Aug. All benchmarked models are up-to-date. For open-weight LLMs like DeepSeek-series models, Llama-series models, Qwen-series models, we directly call API from `https://api.deepseek.com`, `https://openrouter.ai/api/v1`, `https://dashscope.aliyuncs.com/compatible-mode/v1` respectively.

### A.2  DETAILS OF ORACLE BENCHMARK

To balance the cost of LLMs, the number of test samples for each black-box task is set as follows:

- Code Intent Inference (CII): The test samples include 5 unique input variable values, each with 6 or 7 checkpoint questions, totaling 30 or 35 questions.

- Circuit Rule Inference (CRI): Each black-box considers 10 different input wires as test samples.

- Physics System Inference (PSI): Each black-box considers 6 different time points as test samples.

- Encryption Rule Inference (ERI): Each black-box considers 8 different plaintext as test samples.

- Interactive Puzzle Inference (IPI): Each black-box considers 6 different puzzle answers as test samples.

- Game Strategy Inference (GSI): Each black-box considers 4 different game rounds, ranging from 8 to 15.

Test samples are directly generated by LLMs. However, human rewriting is involved in some cases. We find LLMs fail to generate valid and good test samples in Code Intent Inference task. In Interactive Puzzle Inference, some test samples involve numerical calculations (e.g., Wordle), which Large Language Models (LLMs) sometimes struggle with.

# B  ADDITIONAL RELATED WORK

## B.1  DATA CONTAMINATION AND DYNAMIC BENCHMARK

Previous works have highlighted the risks of memorization and contamination during LLM training and fine-tuning (Roberts et al., 2023; Deng et al., 2023): While LLMs are trained over a huge amount of data from Internet, static datasets will be inadvertently included, leading to overestimation of model performance (Magar & Schwartz, 2022; Balloccu et al., 2024). Therefore, researchers begin to shed light on dynamic benchmarks. Current approaches on building dynamic benchmarks can be classified into updating benchmark data based on the timestamps of LLM (White et al., 2024; Jain et al., 2024; Mahdavi et al., 2025) and regenerating benchmark data to reconstruct original benchmarks. Specifically, the latter approach can be further divided into rule-based reconstruction (Lei et al., 2023; Zhu et al., 2024; Mirzadeh et al., 2024; Zhao et al., 2024; Kurtic et al., 2024), LLM-based reconstruction (Ying et al., 2024; Cao et al., 2024; Qian et al., 2024), human-based reconstruction (Srivastava et al., 2024; Huang et al., 2025), and hybrid reconstruction (Zhang et al., 2024).

## B.2  EVALUATION OF REASONING ABILITY

The growing complexity of reasoning tasks undertaken by LLMs makes their evaluation increasingly difficult. Simply relying on the comparisons between LLM-generated outcomes and ground truth labels (Cobbe et al., 2021; Hendrycks et al., 2021) becomes insufficient, as LLMs can produce correct answers with logically incorrect reasoning process (Turpin et al., 2023; Hao et al., 2024; Mondorf & Plank, 2024). Therefore, researchers turn to evaluate reasoning paths step-by-step. Existing criteria of metrics can be categorized into groundedness, validity, coherence, and utility (Lee & Hockenmaier, 2025). Groundedness measures if the step is factually true according to the query (Lewis et al., 2020). Validity evaluates if a reasoning step contains no errors (Lightman et al., 2023). Coherence checks if the inputs for a reasoning step are adequately provided by the prior steps (Wang et al., 2022). Utility measures if a reasoning step contributes to the correct final answer. Besides leveraging evaluation metrics, LLM-as-a-judge is frequently employed in evaluation (Zheng et al., 2023; Hao et al., 2024; Sun et al., 2024), which is a fast and cheap alternative to human judgment.

## B.3  CHARLES PEIRCE'S FRAMEWORK OF HUMANS REASONING BEHAVIOR

The reasoning behavior of humans can be categorized into deductive reasoning, inductive reasoning, and abductive reasoning according to Charles Peirce's framework (Peirce, 1934). Generally speaking, as shown in Figure 1, the reasoning process begins with abduction, where initial observations spark potential explanatory hypotheses. These hypotheses are applied to derive new observations via deduction. Then induction works in strengthening or discarding hypotheses by analyzing former and new observations. This cycle repeats, iteratively refining existing hypotheses until a robust and generalizable theory is built (Burks, 1946; Fann, 2012). Charles Peirce's framework reveals the significance of human reasoning when facing an unknown environment: three aspects of reasoning are dynamically intertwined, collectively driving the discovery, verification, and application of knowledge.

# C ADDITIONAL DISCUSSION

## C.1 LIMITATIONS

The limitations of this work include: (i) This paper's scope is limited to investigating the performance of LLMs in the ORACLE benchmark, with the evaluation of LLM-based agents reserved for future research. (ii) Due to the heavy cost of calling LLMs, we don't evaluate some powerful yet expensive LLMs (e.g., o3-pro). The ORACLE benchmark also lacks a statistical analysis. (iii) Since the models that performed well on the ORACLE benchmark are all closed-source, we are unable to analyze them to determine why they fail to develop efficient and adaptive strategies in black-box interaction tasks. One possible explanation is that, it's well-known that the capabilities of an LLM primarily depend on the base model's pre-training and the post-training. The former involves next token prediction given a context, while the latter is an RL training on CoT prompt-based QA pairs. Neither of these training paradigms involves placing the model in an unknown environment for exploration. Consequently, the model inherently lacks the ability for strategy optimization through multi-turn interaction with an environment.

## C.2 COMPARISON WITH PREVIOUS WORK

Some previous researches (Wu et al., 2023; Hu et al., 2024; Shi et al., 2025) evaluate LLMs' ability of playing text games. Black-box interaction differs from these works in three aspects: First, we focus on building an interactive, unknown environment with hidden rules and investigating LLMs behavior in this environment, instead of testing LLMs' performance in playing specific games. Second, some of the text games can be viewed as a subset of Interactive Puzzle Inference in the ORACLE benchmark, and our proposed black-box interaction approach can easily scale the Interactive Puzzle Inference task. Third, all chosen text games in previous works are well-known, which increases the possibility that LLMs are already familiar with the exploration strategy. Black-boxes in Interactive Puzzle Inference are all modified to avoid this situation (detailed in Appendix G.5).

He et al. (2025) introduced RULEARN benchmark which builds an interactive environment with unknown rules. There exist three noticeable differences between RULEARN and the ORACLE benchmark. First, the ORACLE benchmark formalizes the black-box interaction task into distinct exploration and evaluation stages. In contrast, RULEARN only involves a one-stage multi-turn interaction, thus lacking an analysis of the reasons behind model failure. Second, the three tasks in RULEARN can be classified as Interactive Puzzle Inference (IPI) task within the ORACLE benchmark. In other tasks within the ORACLE benchmark, the model must infer the hidden rule rather than a hidden answer, which is a more fundamental and challenging task. Furthermore, the black-box interaction concept we propose is highly flexible and generalizable. Its goal is to investigate whether a model can understand an unknown environment through exploration, thus accommodating a wide range of task settings beyond the six types in the ORACLE benchmark. In contrast, the task settings in RULEARN are singular and difficult to generalize.

## C.3 MORE TASK SETTINGS FOR BLACK-BOX INTERACTION

In this section, we explore additional potential task settings for black-box interaction. The setting described in Section 2, which allows for test-time learning, is designed by considering possible human evaluation and shot numbers in evaluation metrics. A key distinction when involving human evaluators is the persistence of information. Unlike large language models (LLMs) that can easily delete messages from their dialogue history, humans retain previously seen content in their memory. Therefore, all prior test samples must remain part of the dialogue history. While a $k$-shot evaluation metric is necessary, we also inform the LLM whether its responses to test samples in its $k$ trials were correct. More diverse task settings become feasible when human evaluation and shot numbers are not primary considerations. Their advantages and disadvantages are discussed below.

### C.3.1 FROM ALLOWING TEST-TIME EXPLORATION TO NOT ALLOWING

Test-time exploration is forbidden, which means every test sample and LLMs' answer will be removed from the dialogue history once it's completed. LLMs only rely on information gained from the exploration stage.

- **Advantages**: Evaluation and exploration are totally disentangled, which is a clearer approach.
- **Disadvantages**: This setting is contradictory to human evaluation.

### C.3.2 FROM TEST SAMPLES TO CODE EXPRESSION

Rather than directly providing answers for each test sample, models are now instructed to generate executable code that expresses their understanding of the black-box, which is then validated against test samples.

- **Advantages**: Code expression is a decent way to judge whether LLMs truly understand the black-box rather than simply pattern matching, which is more fundamental to the core concept of black-box interaction. More importantly, code allows for checking millions of test samples quickly and is especially useful for some specific black-boxes. For examples, in some black-boxes in GSI task that involves a random opponent strategy, code expression is capable of simulating millions of games to judge whether models truly understand opponent's strategy. In PSI task, code expression is effective for complicated motion without analytical solution . We already apply code expression for "Double Pendulum", "Harmonic with Friction", "Ball Air Resistance" black-boxes in the PSI task (detailed in Appendix G.3).
- **Disadvantages**: This approach introduce the additional challenge of coding ability of LLMs, which contradicts to the original of building a pure reasoning benchmark. As the possibility of a model can answer but fail to write correct code exists. In this setting, $k$-shot evaluation is also inapplicable.

### C.3.3 FROM FULLY OBSERVABLE TO PARTIALLY OBSERVABLE

In current setting, black-box returns complete state information to models' query, which makes up of fully observable black-box interaction. Partially observable black-box interaction only returns part of black-box information. For example, black-box will only returns values of a subset of variables in CII task. Partially observable black-box interaction is a harder version of fully observable black-box interaction.

- **Advantages**: Partially observable black-box interaction generalizes the current setting and better reflect real-world scenarios. It further challenges models' advanced reasoning by increasing the difficulty of aggregating imperfect information. Exploration turns are also supposed to be extended, which tests models' long-context reasoning.
- **Disadvantages**: Not applicable.

### C.3.4 FROM FIXED EXPLORATION TURNS TO DYNAMIC

The goal of models is accurately answer test samples with the fewest possible exploration turns, rather than given fixed exploration turns.

- **Advantages**: This setting significantly tests a model's planning capabilities, requiring the development of highly effective exploration strategies for strong performance.
- **Disadvantages**: When exploration strategies become dynamic, comparing model performance gets tricky. It's tough to decide if a model with less exploration but lower accuracy is better than one with more exploration and higher accuracy.

### C.4 EXTRA FINDINGS

Some extra findings during experiments are reported here. First, we find some LLMs, especially gemini-2.0-flash and gemini-2.5-flash, perform bad in instruction following. Black-box interaction requires accurate output format. While LLMs are given chances to correct formatting mistakes, continued disobedience of the specified format results in an invalid interaction turn. Second, the time cost of black-box interaction is an issue worthy of attention. We find that o4-mini and gemini-2.5-pro achieve the best balance between accuracy and time cost, while the time cost of qwen-series models and deepseek-r1 is extremely high.

We have also identified three additional weaknesses of LLMs in tackling black-box interaction tasks. First, LLMs primarily rely on pattern matching to understand black-box. Prior knowledge is essential for hypothesis developing and verifying in abductive and inductive reasoning. However, we find that LLMs rely heavily on prior knowledge and matching black-box function to familiar patterns, rather than engaging in genuine exploration. This phenomenon is most evident in the CII task where all black-boxes are famous coding algorithms. LLMs can quickly identify the hidden algorithm with only few observations over checkpoint output. So despite the difficult setting of CII, LLMs still perform well. But when it comes to PSI task where a black-box is a free combination of moving objects, or ERI task where the black-boxes are variations of well-known encryption algorithms, the performance of LLMs becomes relatively low. Notably, almost all models (including o3, gemini-2.5-pro) fail to beat a simple black-box that plays rock and scissors with equal probability in GSI 2@1 (as shown in Figure 2 (f)). Second, LLMs struggle in reasoning over dense information. LLMs are supposed to spend more turns for exploration when the black-box function is complex. They cannot achieve good performance without the ability to reason over dense information. This weakness is most evident in "Wordle" and "Quordle" black-boxes in the IPI task. LLMs can easily guess a 11-letter word in "Wordle" within 10 rounds, but fail to guess four 8-letter words in "Quordle" within 20 rounds. Third, even best-performing reasoning LLMs fall short in basic computing ability. For example, in a black-box implementing simple harmonic motion in PSI task, gemini-2.5-pro successfully identify the motion behavior but fail in correctly calculating the coordinates. Another example is the "Nerdle" black-box in IPI task which requires LLMs to output a 15-character equation. Most models fail to calculate if the output only contains 15 characters.

## D  CASE STUDY

### D.1  HOW ITERATIVE DEBUGGING WORKS

The effectiveness of our iterative debugging framework stems from its ability to uncover a wide range of errors that are often missed in a single-pass generation process. By simulating a full interaction—akin to a human learning a game by playing it—the framework can identify and rectify several classes of bugs that a programmer might make. These include:

1. **Violations of unstated "common-sense" rules**: Task descriptions often omit implicit constraints, such as the fact that a player's score or money cannot be negative. Our interactive process makes these violations apparent, forcing a correction.

2. **Misinterpretations of ambiguous language**: Natural language can be imprecise. The framework corrects for misunderstandings, as demonstrated in the example of circuit task where the term "random" was initially misinterpreted, leading to a non-deterministic implementation instead of a fixed random one.

3. **Simple yet critical implementation bugs**: This category includes flaws analogous to typos or logical oversights, such as using an incorrect formula in the example of physics task. It also includes bugs that cause runtime errors. These are difficult to spot in a static code review but are readily exposed when the simulation produces incorrect outputs.

Therefore, our framework significantly lowers the natural language description requirements for platform development. It tolerates ambiguity and allows for the omission of details, even granting the coding agent the freedom to elaborate on aspects that don't compromise the platform's core functionality. Furthermore, it enhances accuracy by autonomously correcting runtime and other logical errors. This substantially lowers the costs associated with benchmark construction, fine-tuning, and scalability.

The following case studies provide concrete examples of how our framework addresses flaws to produce robust and correct platform code.

**An Example in Circuit Rule Inference Task**   In this example from the Circuit Rule Inference Task, the initial natural language description provided to the Coding LLM was: "For input size n=4, the blackbox contains a random circuit of 8 gates. The circuit is not for some well-known purpose." The Coding LLM misinterpreted the term "random." While the circuit's structure was meant to be random, it needed to be fixed throughout a single interaction session to represent a deterministic physical device. Instead, the initial code 1 regenerated a new random circuit for every call to the blackbox function. As shown in Figure 11a and Figure 12, the Test LLM's interaction quickly exposed this flaw by providing identical inputs at different rounds and receiving different outputs, a logical contradiction that the Refinement LLM then used to correct the code 2.

```python
# Conical Pendulum Version 1.0
def blackbox(circuit_input):
    """
    Simulates a random boolean circuit with
    ↪  4 input wires and 8 gates.
    Returns the output of each gate as a
    ↪  list of 0/1 bits.
    """
    n = 4  # number of input wires
    m = 8  # number of gates

    # Randomly construct a circuit, each
    ↪  gate's inputs come from input wires
    ↪  or previous gates
    gates = []
    for i in range(m):
        gate_type = random.choice(['AND',
        ↪  'OR', 'NOT'])
        # For gate i (indexing from 0),
        ↪  possible sources: input wires
        ↪  (1..n) or previous gates (1..i)
        inputs = []
        for idx in range(1, n+1):
            inputs.append( (0, idx) )
        for idx in range(1, i+1):
            inputs.append( (1, idx) )
        if gate_type in ['AND', 'OR']:
            # Randomly select two inputs
            choices = random.sample(inputs,
            ↪  2)
            input1, input2 = choices[0],
            ↪  choices[1]
            gates.append( (gate_type,
            ↪  input1, input2) )
        elif gate_type == 'NOT':
            # Need one input
            input1 = random.choice(inputs)
            gates.append( (gate_type,
            ↪  input1) )

    # Simulate the circuit
    result = simulate_circuit(n, m,
    ↪  circuit_input, gates)
    return result
```

Listing 1: Code of v1 platform
(randomly generate circuit repeatedly)

```python
# Conical Pendulum Version 2.0
# Circuit parameters
N = 4  # number of input wires
M = 8  # number of gates

# Generate the circuit once per run to
↪  ensure deterministic behavior for the
↪  same input
_CIRCUIT_GATES = []
for i in range(M):
    gate_type = random.choice(['AND', 'OR',
    ↪  'NOT'])
    # possible sources: input wires
    ↪  (0,1..N) or previous gates (1..i)
    inputs = [(0, idx) for idx in range(1,
    ↪  N + 1)]
    for idx in range(1, i + 1):
        inputs.append((1, idx))
    if gate_type in ['AND', 'OR']:
        if len(inputs) < 2:
            input1, input2 = (0, 1), (0, 2)
        else:
            input1, input2 =
            ↪  random.sample(inputs, 2)
        _CIRCUIT_GATES.append((gate_type,
        ↪  input1, input2))
    else:  # NOT
        input1 = random.choice(inputs)
        _CIRCUIT_GATES.append((gate_type,
        ↪  input1))

def blackbox(circuit_input):
    """
    Simulates the fixed random boolean
    ↪  circuit with 4 input wires and 8
    ↪  gates.
    Returns the output of each gate as a
    ↪  list of 0/1 bits.
    """
    # Simulate the circuit
    result = simulate_circuit(N, M,
    ↪  circuit_input, _CIRCUIT_GATES)
    return result
```

Listing 2: Code of v2 platform
(generate circuit only once)

(...Instructions Omitted...)

user: Welcome to the Boolean Circuit Game! You are interacting with a blackbox circuit with 4 input wires and 8 logic gates. (...) [Round 2/20 | 19 rounds left]

LLM: (0, 0, 0, 1)

user: Gate outputs for your input [0, 0, 0, 1]: 1 1 0 1 1 1 1 1 Please submit your next input bits for round 3. (...) [Round 10/20 | 11 rounds left]

LLM: (1, 0, 0, 1)

user: Gate outputs for your input [1, 0, 0, 1]: 1 0 1 0 1 0 0 0 Please submit your next input bits for round 11. (...) [Round 17/20 | 4 rounds left]

LLM: (0, 0, 0, 1)

user: Gate outputs for your input [0, 0, 0, 1]: 1 1 1 1 1 0 0 1 Please submit your next input bits for round 18. [Round 18/20 | 3 rounds left]

LLM: (1, 0, 0, 1)

user: Gate outputs for your input [1, 0, 0, 1]: 0 0 0 0 0 1 1 0 (...)

********Evaluation Starts, You Have 1 Chances for Answering Each Question******** The output format is described in the Evaluation section previously. For example: [0, 1, 0, 1] In this turn, given the input [0, 0, 0, 0], answer the output of the gates in the format we dicussed without any text else.

LLM: [0, 0, 1, 0, 1, 0, 0, 0]

user: the answer is wrong when input = [0, 0, 0, 0] Let's move to next question.

LLM: Ok.

(a) Interaction log (v1 platform)

(...Instruction Omitted...)

user: Welcome to the Boolean Circuit Game! You are interacting with a blackbox circuit with 4 input wires and 8 logic gates. (...) [Round 2/20 | 19 rounds left]

LLM: (0, 0, 0, 1)

user: Gate outputs for your input [0, 0, 0, 1]: 0 1 0 0 0 1 0 1 Please submit your next input bits for round 3. (...) [Round 10/20 | 11 rounds left]

LLM: (1, 0, 0, 1)

user: Gate outputs for your input [1, 0, 0, 1]: 1 0 1 1 0 0 1 0 Please submit your next input bits for round 11. (...) [Round 17/20 | 4 rounds left]

LLM: (0, 0, 0, 1)

user: Gate outputs for your input [0, 0, 0, 1]: 0 1 0 0 0 1 0 1 Please submit your next input bits for round 18. [Round 18/20 | 3 rounds left]

LLM: (1, 0, 0, 1)

user: Gate outputs for your input [1, 0, 0, 1]: 1 0 1 1 0 0 1 0 (...)

********Evaluation Starts, You Have 1 Chances for Answering Each Question******** The output format is described in the Evaluation section previously. For example: [0, 1, 0, 1] In this turn, given the input [0, 0, 0, 0], answer the output of the gates in the format we dicussed without any text else.

LLM: [0, 1, 0, 0, 1, 1, 0, 1]

user: the answer is wrong when input = [0, 0, 0, 0] Letś move to next question.

LLM: Ok.

(b) Interaction log (v2 platform)

Figure 11: Black-box interaction log generated using different platform code

- The same input [0, 0, 0, 1] produced two different outputs:
• Round 2 → 1 1 0 1 1 1 1 1
• Round 17 → 1 1 1 1 1 0 0 1
- The same input [1, 0, 0, 1] produced two different outputs:
• Round 10 → 1 0 1 0 1 0 0 0
• Round 18 → 0 0 0 0 0 1 1 0
These contradictions violate the requirement that a fixed combinational circuit must give the same outputs for identical inputs.

Figure 12: Logical error figured out by Refinement LLM

**An Example in Physics System Inference Task**   This case demonstrates the framework's ability to detect and correct subtle but critical bugs in the implementation logic, corresponding to the third type of error mentioned above. The task is to simulate a conical pendulum, a standard physics problem. The initial code generated by the Coding LLM 3 contained a flaw in the physical formula for calculating the angular velocity $\omega$, incorrectly using cos(theta) where tan(theta) was required. This type of error is analogous to a small logical oversight by a human programmer—it is syntactically correct but semantically wrong, making it difficult to catch without executing the code and validating its output. During the simulation phase 13a, the framework discovered that the coordinates produced by the v1 platform were inconsistent with the expected physical behavior, leading to failed evaluation checks in 14. The Refinement LLM then correctly identified the erroneous formula as the root cause and produced the corrected code 4.

```python
# Conical Pendulum Version 1.0
def blackbox(t):
    # Conical pendulum parameters
    length = 5  # m
    theta = math.radians(30)  # 30 degrees
    ↪ in radians
    g = 10  # m/s²

    # Calculate motion parameters
    omega = math.sqrt(g * math.cos(theta) /
    ↪ length)  # angular velocity
    radius = length * math.sin(theta)  #
    ↪ radius of circular motion
    z_position = -length * math.cos(theta)
    ↪ # constant z position (negative,
    ↪ below origin)

    # Calculate 3D coordinates
    x = radius * math.cos(omega * t)
    y = radius * math.sin(omega * t)
    z = z_position

    # Round to two decimal places
    x = round(x, 2)
    y = round(y, 2)
    z = round(z, 2)

    object_coordinate = {"object1": (x, y,
    ↪ z)}
    return object_coordinate
```

Listing 3: Code of v1 platform

```python
# Conical Pendulum Version 2.0
def blackbox(t):
    # Conical pendulum parameters
    length = 5  # m
    theta = math.radians(30)  # 30 degrees
    ↪ in radians
    g = 10  # m/s²

    # Calculate motion parameters
    omega = math.sqrt(g * math.tan(theta) /
    ↪ length)  # angular velocity
    radius = length * math.sin(theta)  #
    ↪ radius of circular motion
    z_position = -length * math.cos(theta)
    ↪ # constant z position (negative,
    ↪ below origin)

    # Calculate 3D coordinates
    x = radius * math.cos(omega * t)
    y = radius * math.sin(omega * t)
    z = z_position

    # Round to two decimal places
    x = round(x, 2)
    y = round(y, 2)
    z = round(z, 2)

    object_coordinate = {"object1": (x, y,
    ↪ z)}
    return object_coordinate
```

Listing 4: Code of v2 platform

(...Instructions Omitted...)
Black-box: You have 6 interaction turns to under-stand the black-box. Now the interaction starts. Only output the value and DO NOT contain any unrelated text.
LLM: 0
Black-box: (Current Turn: 1, 5 Turns Remaining) {'object1': (2.5, 0.0, -4.33)}
LLM: 1
Black-box: (Current Turn: 2, 4 Turns Remaining) {'object1': (0.63, 2.42, -4.33)}
LLM: 2
Black-box: (Current Turn: 3, 3 Turns Remaining) {'object1': (-2.18, 1.22, -4.33)}
LLM: 3
Black-box: (Current Turn: 4, 2 Turns Remaining) {'object1': (-1.73, -1.8, -4.33)}
LLM: 4
Black-box: (Current Turn: 5, 1 Turns Remaining) {'object1': (1.31, -2.13, -4.33)}
LLM: 5
Black-box: (Current Turn: 6, 0 Turns Remaining) {'object1': (2.39, 0.73, -4.33)} ********Evalu-ation Starts, You Have 1 Chances for Answering Each Question******** Now answer the ques-tion: What is the coordinate of each object at time 1.5?
LLM: {'object1': (-0.78, 2.29, -4.33)}
Black-box: Your answer is wrong. Let's move to next question.
LLM: Ok.

(a) Interaction log (v1 platform)

(...Instructions Omitted...)
Black-box: You have 6 interaction turns to under-stand the black-box. Now the interaction starts. Only output the value and DO NOT contain any unrelated text.
LLM: 0
Black-box: (Current Turn: 1, 5 Turns Remaining) {'object1': (2.5, 0.0, -4.33)}
LLM: 1
Black-box: (Current Turn: 2, 4 Turns Remaining) {'object1': (1.19, 2.2, -4.33)}
LLM: 2
Black-box: (Current Turn: 3, 3 Turns Remaining) {'object1': (-1.37, 2.09, -4.33)}
LLM: 3
Black-box: (Current Turn: 4, 2 Turns Remaining) {'object1': (-2.49, -0.21, -4.33)}
LLM: 4
Black-box: (Current Turn: 5, 1 Turns Remaining) {'object1': (-1.01, -2.29, -4.33)}
LLM: 5
Black-box: (Current Turn: 6, 0 Turns Remaining) {'object1': (1.53, -1.97, -4.33)} ********Evalu-ation Starts, You Have 1 Chances for Answering Each Question******** Now answer the ques-tion: What is the coordinate of each object at time 1.5?
LLM: {'object1': (0.11, 2.22, -4.33)}
Black-box: Your answer is wrong. Let's move to next question.
LLM: Ok.

(b) Interaction log (v2 platform)

Figure 13: Black-box interaction log generated using different platform code

Looking at the interaction log carefully:

**Wrong points identified:**

1. **Insufficient analysis of the mechanical system**: The assistant fails to properly analyze the motion pattern. From the coordinates provided:
   - t=0: (2.5, 0.0, -4.33)
   - t=1: (0.63, 2.42, -4.33)
   - t=2: (-2.18, 1.22, -4.33)
   - t=3: (-1.73, -1.8, -4.33)
   - t=4: (1.31, -2.13, -4.33)
   - t=5: (2.39, 0.73, -4.33)
The z-coordinate remains constant at -4.33, and the x,y coordinates trace a circular path, indicating conical pendulum motion. However, the assistant doesn't demonstrate understanding of this pattern in their final prediction.

Figure 14: Logical error figured out by Refinement LLM

## D.2  How LLMs Interact with the Black-Box

**An Example in Interactive Puzzle Inference Task**

developer: You are an excellent reasoning model. You are very good at understanding the interaction rules, reasoning over interaction history, thinking step by step, and finally figuring out the working principle of a system that contains hidden information.
You also strictly follow output format rules. When making queries, you strictly obey querying format. When answering questions, you strictly obey answer format. You never output anymore unrelated text.

user: 1. Task overview:
- The user plays the role of a puzzle, and you don
't know what the hidden answer is. You need to guess the hidden answer by interacting with the user in multiple turns.
2. Goals:
- You need to guess the answer to the puzzle within given interaction turns.
3. User property:
- The user hides the answer which you need to figure out.
4. Interaction rules:
- Rule 0: The user will first tell you the rule of the puzzle, and the interaction format that must be followed when querying. In each turn, the user will tell *current turn* and *remaining turns*.
- Rule 1: You can ask questions according to the rules of the game and receive corresponding feedback. If your ask is unavailable, the user will tell you.
- Rule 2: After a series of interactions, you should answer the puzzle in the format specified in the description.
5. Output format:
- When you ask a question, you should strictly follow query format in the **Description**.
- When you answer the puzzle, you should strictly follow the answer format in the **Description**.
- Refer to the examples in the **Description**. for the correct format.
- If you figure out the right answer before given turns, keep interacting with the puzzle to make sure your answer is correct.
6. Evaluation:
- When the given number of interactions is reached, you need to give your answer of the puzzle. **You MUST ONLY output the answer itself in the format mentioned in the description, DO NOT contain more text.**
Now Let's Solve the Puzzle wordle.
**Description**: Wordle is a word-guessing game where players attempt to deduce a hidden 8-letter word (all uppercase). Each guess provides feedback (1) Correct letter in the correct position, represented by 'A'; (2) Correct letter but misplaced, represented by 'M'; (3) Letter not in the word, represented by 'X'.
Players iteratively refine guesses using feedbacks until solving the word, typically within limited at-tempts. After a few times of querying, the player will give an 8-letter uppercase word answer, and he
'll win if he answers correctly.
OUTPUT the 8-letter uppercase word directly.

LLM: I understand the rules. I will not output any unrelated text! Let us start the interaction.

user: ********A New Puzzle Starts, You can Make 10 Queries Before Answering Each Question. And Then You Have 1 Chances for Answering. Output the Value Only.********

LLM: AIRPLANE

user: <Current Turn: 1, 9 Turns Remaining> AXMXMMMM

LLM: ABALONES

user: <Current Turn: 2, 8 Turns Remaining> AXMMXMMX

LLM: ARLENDTC

user: <Current Turn: 3, 7 Turns Remaining> AAMMAXXM

LLM: ARCANELY

user: <Current Turn: 4, 6 Turns Remaining> AAAAAAAA

LLM: ARCANELY

user: <Current Turn: 5, 5 Turns Remaining> AAAAAAAA

LLM: ARCANELY

user: <Current Turn: 6, 4 Turns Remaining> AAAAAAAA

LLM: ARCANELY

user: <Current Turn: 7, 3 Turns Remaining> AAAAAAAA

LLM: ARCANELY

user: <Current Turn: 8, 2 Turns Remaining> AAAAAAAA

LLM: ARCANELY

user: <Current Turn: 9, 1 Turns Remaining> AAAAAAAA

LLM: ARCANELY

user: <Current Turn: 10, 0 Turns Remaining> AAAAAAAA********Evaluation Starts, You Have 1 Chances for Answering, Please Output the Answer DIRECTLY.********

LLM: ARCANELY

user: Your answer is correct.

LLM: Ok.

**An Example in Game Strategy Inference Task**

developer: You are an excellent reasoning model. You are very good at understanding the interaction rules with a black-box, reasoning over the interaction history, thinking step by step, and finally guessing the pattern of the black-box and finding a way to beat it in a game. You also strictly follow output format rules.

user: 1. Task overview: - The user plays the role of an opponent who takes a fixed strategy in a game. But you don't know what the stategy is. You need to guess the hidden stategy of your opponent by interacting (playing game) with him in multiple turns.
2. Goals: - You have 1 final goal: You need to guess your opponent's strategy and try to maximize your score in the game. The score might depend on winning rate, or minimal cost, etc.
3. User property: - The user hides his game strategy which you need to figure out to win the game.
4. Interaction rules: - To finish the goal, you need to interact with the user. The interaction rules are as follows: - Interaction Rule 0: The user will first tell you the rule of the game, and the interaction format that must be followed when playing. In each turn, the user will tell *current turn* and *remaining turns*. - Interaction Rule 1: You can take actions according to the rules of the game and receive corresponding feedback, such as current game states. If your action is unavailable, the user will tell you. - Interaction Rule 2: We will first play a few times of the game to familiarize you with the rules and the behavior of your opponent. In this phase, your actions will not be recorded, and your score does not matter. You can make use of this phase to explore the game and understand the opponent's strategy. - Interaction Rule 3: After the *exploration phase*, you will enter the *evaluation phase*. We will only play the game for 1 final time, and your actions will be recorded. Your score will be calculated based on your actions in this final game.
5. Output format: - **You must strictly obey the output format rules, DO NOT output any unrelated text!**:
6. Evaluation: - When the given number of interactions is reached, the game ends and we'll calculate your **score**

Now Let's Play the Game Load Shoot Defend Scout, the Description Is that In each turn, a player can choose 'load': gain one bullet, 'scout': see the opponent's bullet count, 'shoot x': spend 'x' bullets to attack, or 'defend y': spend 'y' bullets to defend. Then, actions and the gaining points of both players are revealed, but the specific numerical values 'x' and 'y' are kept hidden. Each player starts with 0 bullets and can hold a maximum of 8. 'x' and 'y' cannot exceed the number of bullets they have currently. If one player 'shoot x' and the other 'defend y', the defense succeeds and the player wins 1 point if y>=x, and the shooter wins -1 point. But if y<x, the attacker wins 1 point and the defender wins -1 point. A Shoot action will always succeed against an opponent who chooses to Load or Scout. When a simultaneous shoot happens, player with more bullets will win 1 point and the other win -1 point. The same bullets will result in a tie where both players win 0 point. When a simultaneous defend or scout or load happens, both players win 0 point. If player 'scout', he will receive opponent's bullet number. The game lasts for [[total_turns]] turns, both players try to maximize their scores.

LLM: I understand the rules. I will not output any unrelated text! Let us start the interaction.

user: ********Exploration Phase Starts, We wll Play the Game for 1 Times. Your Actions Will Not Be Recorded, and Your Score Does Not Matter.**********

LLM: Please provide the game rules and the required interaction format so that I can take my first action in the exploration phase.

user: ***Exploration Round <1/1> Start***

LLM: Ok. I'm ready to play the game. This is round 1 of the exploration phase.

user: Turn 1/8
This is the first turn.
You have 0 bullets.
What is your action? (e.g., 'load', 'scout', 'shoot 1', 'defend 2')

LLM: load

user: Turn 2/8
In the last turn, you chose 'load' and the opponent chose 'load'. You gained 0 point(s).
You have 1 bullets.
What is your action? (e.g., 'load', 'scout', 'shoot 1', 'defend 2')

LLM: I need the complete rules and scoring details to make informed choices. Could you please provide the game's description, the available actions, and how points are awarded?

user: Invalid action type. Your action 'I need the complete rules and scoring details to make informed choices. Could you please provide the game's description, the available actions, and how points are awarded?' is not recognized. Please choose from 'load', 'scout', 'shoot x', or 'defend y'.

LLM: scout

user: Turn 3/8
In the last turn, you chose 'scout' and the opponent chose 'load'. You gained 0 point(s).
You have 1 bullets.
Your scout last turn revealed that the opponent had 2 bullets before their action.
What is your action? (e.g., 'load', 'scout', 'shoot 1', 'defend 2')

LLM: scout

user: Turn 4/8
In the last turn, you chose 'scout' and the opponent chose 'shoot'. You gained 0 point(s).
You have 1 bullets.
Your scout last turn revealed that the opponent had 2 bullets before their action.
What is your action? (e.g., 'load', 'scout', 'shoot 1', 'defend 2')

LLM: shoot 1

user: Turn 5/8

In the last turn, you chose 'shoot 1' and the opponent chose 'load'. You gained 1 point(s).
You have 0 bullets.
What is your action? (e.g., 'load', 'scout', 'shoot 1', 'defend 2')

LLM: load

user: Turn 6/8
In the last turn, you chose 'load' and the opponent chose 'shoot'. You gained 0 point(s).
You have 1 bullets.
What is your action? (e.g., 'load', 'scout', 'shoot 1', 'defend 2')

LLM: scout

user: Turn 7/8
In the last turn, you chose 'scout' and the opponent chose 'load'. You gained 0 point(s).
You have 1 bullets.
Your scout last turn revealed that the opponent had 1 bullets before their action.
What is your action? (e.g., 'load', 'scout', 'shoot 1', 'defend 2')

LLM: scout

user: Turn 8/8
In the last turn, you chose 'scout' and the opponent chose 'load'. You gained 0 point(s). You have 1
bullets. Your scout last turn revealed that the opponent had 2 bullets before their action. What is your
action? (e.g., 'load', 'scout', 'shoot 1', 'defend 2')

LLM: defend 1

user: ********Evaluation Phase Starts, We Will Play the Game for 1 Time. Now is the 0 time.
The highest score Will Be Recorded.**********

LLM: I am ready

user: Turn 1/8
This is the first turn. You have 0 bullets.
What is your action? (e.g., 'load', 'scout', 'shoot 1', 'defend 2')

LLM: load

user: Turn 2/8
In the last turn, you chose 'load' and the opponent chose 'load'. You gained 0 point(s). You have 1
bullets.
What is your action? (e.g., 'load', 'scout', 'shoot 1', 'defend 2')

LLM: shoot 1

user: Turn 3/8
In the last turn, you chose 'shoot 1' and the opponent chose 'load'. You gained 1 point(s). You have 0
bullets.
What is your action? (e.g., 'load', 'scout', 'shoot 1', 'defend 2')

LLM: load

user: Turn 4/8
In the last turn, you chose 'load' and the opponent chose 'shoot'. You gained 0 point(s). You have 1
bullets.
What is your action? (e.g., 'load', 'scout', 'shoot 1', 'defend 2')

LLM: shoot 1

user: Turn 5/8
In the last turn, you chose 'shoot 1' and the opponent chose 'load'. You gained 1 point(s). You have 0
bullets.

What is your action? (e.g., 'load', 'scout', 'shoot 1', 'defend 2')

LLM: load

user: Turn 6/8
In the last turn, you chose 'load' and the opponent chose 'shoot'. You gained 0 point(s). You have 1 bullets.
What is your action? (e.g., 'load', 'scout', 'shoot 1', 'defend 2')

LLM: shoot 1

user: Turn 7/8
In the last turn, you chose 'shoot 1' and the opponent chose 'load'. You gained 1 point(s). You have 0 bullets.
What is your action? (e.g., 'load', 'scout', 'shoot 1', 'defend 2')

LLM: load

user: Turn 8/8
In the last turn, you chose 'load' and the opponent chose 'load'. You gained 0 point(s). You have 1 bullets.
What is your action? (e.g., 'load', 'scout', 'shoot 1', 'defend 2')

LLM: shoot 1

### D.3   HOW LLMs SUCCEED AND FAIL

**A Successful Example in Code Intent Inference Task**   In this case, the black-box code implemented QuickSort, selecting the last element of the interval as the pivot. The LLM quickly identified this as QuickSort and correctly determined the pivot element.

Black-Box: (Instructions omitted) <Current Turn: 1, 19 Turns Remaining> The black-box takes ['name': 'arr', 'type': <class 'list'>] as input variables, and has 3 checkpoints.

Gemini-2.5-pro: <think>**Deconstructing a Function: My Initial Approach**Alright, I'm tasked with understanding this function that takes a list, 'arr', as input and hits three checkpoints along the way. My gut tells me the best strategy here is to start simple and build complexity. So, I'm thinking, let's feed it a very straightforward input: a sorted list. I'm imagining 'arr' will start as '[1, 2, 3, 4, 5]'. This should allow me to focus on the core logic without any confusing edge cases initially. My immediate next step is to get a handle on what's happening at the first checkpoint, specifically at iteration '(1, 1)'. That's where I need to start. I'm hoping this will give me a clear picture of the initial setup or the state at the very beginning of any iterative process the function might employ. Knowing this initial state will be key to understanding how the function progresses and eventually lands at those critical checkpoints. So I need to query this state and observe what's happening with the input array at that initial point.</think> arr = [1, 2, 3, 4, 5] (1, 1)

Black-Box: <Current Turn: 2, 18 Turns Remaining> Error: Failed to convert [1, 2, 3, 4, 5](1, 1) to list: invalid syntax (<string>, line 2)

Gemini-2.5-pro:  <think>**Understanding the Input and Refining My Strategy**Okay, so I got an error message back after my last attempt, indicating a problem with the parsing of my input. It looks like my combined assignment and checkpoint request in a single command, 'arr = [1, 2, 3, 4, 5]' and '(1, 1)', threw the system off. The "invalid syntax" error is a clear indicator that the system can't handle them together. Rereading the rules, I realize I need to be more deliberate about the order of operations and the output format. Rule 4 emphasizes that I need to handle variable assignments (Rule 1) *before* requesting checkpoint values (Rule 2). While the rules don't explicitly mandate separate turns, the error suggests it's the safest and most reliable approach. Therefore, for this round, I'll focus *solely* on assigning a value to 'arr'. I'll keep the input simple and valid – 'arr = [10, 20, 30]'. This should ensure the system receives a clean and understandable variable assignment. The next turn will be devoted to requesting checkpoint values using the appropriate format. This should minimize the chance of triggering those parsing issues again.</think> arr = [10, 20, 30]

Black-Box: <Current Turn: 3, 17 Turns Remaining> Set 'arr': [10, 20, 30].

Gemini-2.5-pro: <think> </think>

Black-Box: (Instructions omitted) <Current Turn: 1, 19 Turns Remaining> The black-box takes ['name': 'num', 'type': list[int]] as input variables, and has 5 checkpoints.

Gemini-2.5-pro: <think>**Analyzing the Function's Behavior**Okay, so I'm given an array, 'arr', and the number of checkpoints, which is 3 in this case. My immediate task is to grasp the function's internal workings. I've mentally initialized 'arr' as [10, 20, 30] to have some concrete data to work with. Now, the plan is to use checkpoints to reconstruct the function's execution flow. By strategically placing probes, I can see what variables are accessible and how their values evolve. The first step is checking the first checkpoint. Since the checkpoints are numbered, this likely corresponds to the beginning of the function's core loop or main logic. Probing at (1, 1) – the first checkpoint of the first iteration – will give me the initial state. I anticipate seeing 'arr', as it's the input. There may also be some initial loop counter(s) or any initial state variables. This will lay the groundwork for understanding the process.</think> (1, 1)

Black-Box: <Current Turn: 4, 16 Turns Remaining> ['name=arr, value=[10, 20, 30], type=list', 'name=p, value=30, type=int']

Gemini-2.5-pro: <think>**Initial Analysis of the Hidden Code Function**Okay, so I'm presented with a function and my job is to reverse-engineer its logic. My initial focus is on figuring out what's happening with the variables, specifically 'arr' which is a list. The user's set me up with an initial array: 'arr = [10, 20, 30]'. I've got three checkpoints to work with, which I assume represent significant points in the function's execution.My first query was at checkpoint 1, iteration 1, to get a handle on the initial state. The response revealed that 'arr' is unchanged, which makes sense. More interestingly, a new variable, 'p', popped up, with a value of '30'. And '30', of course, is the *last* element in my 'arr'. Based on this first data point, my preliminary hypothesis is that this section of code might be setting up a pivot value, like in a sorting algorithm - perhaps a quicksort implementation. It's just the first checkpoint, and I need more data to confirm this, but it's a logical starting point. I'll need to watch what happens to 'arr' and 'p' at subsequent checkpoints. I'm keen to see if my pivot hypothesis holds.</think> (2, 1)

Black-Box: <Current Turn: 5, 15 Turns Remaining> ['name=arr, value=[10, 20, 30], type=list', 'name=p, value=30, type=int', 'name=l, value=[10, 20], type=list', 'name=r, value=[], type=list', 'name=i, value=1, type=int']

Gemini-2.5-pro: <think>**Initial Analysis of the Function**Okay, so I'm trying to understand this function. It takes a list, 'arr', as input. I'm imagining 'arr = [10, 20, 30]' to get a feel for things. The user confirmed this input. I've got three checkpoints I need to examine.**Checkpoint 1: Identifying the Pivot**I've looked at the first checkpoint, '(1, 1)'. The input is still '[10, 20, 30]'. 'p' is '30'. My initial thought is that this checkpoint is selecting the last element of the list, 'arr', as the pivot , 'p'. That seems pretty straightforward.**Checkpoint 2: Partitioning the Array**Now, on to checkpoint '(2, 1)'. 'arr' and 'p' are the same as before. Here's where it gets interesting. I see 'l' is '[10, 20]' and 'r' is '[]'. So, it looks like it's partitioning the array. 'l' contains all the elements less than or equal to the pivot (which are all elements before the pivot), and 'r' contains elements greater than the pivot. 'i' is at 1, which suggests some kind of looping through elements before the pivot is taking place. It seems to be the loop counter. Given that 'l' has two elements in it, it's likely gone through at least one iteration. The array is being partitioned into 'l' and 'r' elements that meet the condition.**Developing a Hypothesis: Quicksort**Based on this, and before I go further, I'm already starting to suspect this is a Quicksort algorithm. Let's look at Checkpoint 3, but let's look at '(2,2)' first. To confirm my suspicion, I need to see how the partitioning logic works in the loop. I'm going to focus on how 'l', 'r', and 'i' are changing. After that, I'll examine the first recursive call at checkpoint '(3, 1)'. My working hypothesis is:1. Checkpoint 1 selects the pivot.2. Checkpoint 2 partitions the array using the pivot.3. Checkpoint 3 is probably where the recursive calls are made, and the array is combined after the recursive calls. It'll probably combine 'l' and 'r' after sorting them recursively.I'm now going to query '(2, 2)' to verify this.</think> (2, 2)

Black-Box: <Current Turn: 6, 14 Turns Remaining> ['name=arr, value=[10, 20], type=list', 'name=p, value=20, type=int', 'name=l, value=[10], type=list', 'name=r, value=[], type=list', 'name=i, value=0, type=int']

Gemini-2.5-pro: <think>**Analyzing a Quicksort Implementation**Okay, so I'm trying to figure out this function, and I'm pretty sure it's Quicksort. Let's see... I started with 'arr = [10, 20, 30]'.* **CP1:** It looks like this checkpoint selects the last element of the input array as the pivot ('p'). So, if 'arr' is '[10, 20, 30]', 'p' becomes '30'. Easy enough.* **CP2:** This is where things get interesting. This is definitely the partitioning step. It takes the array (minus the last element) and partitions it into two lists: 'l' (elements less than the pivot) and 'r' (elements greater than or equal to the pivot). I'm pretty confident in that. The 'i' value shown corresponds to the final value of the partitioning loop. It seems to be the index counter from a loop over the array's elements, 'arr[:-1]'.So, the hypothesis is, it's a recursive Quicksort. CP1 selects the pivot, CP2 partitions the array, and then it recursively sorts the left and right partitions. Let's see: 'sort([10, 20, 30])' calls 'sort([10, 20])' and then 'sort([10])'. And from those return '[10]' and '[10, 20]'.To confirm the partitioning loop, let's look at 'sort([10])'. When the array is just '[10]', it should also run the partitioning stage but with no actual work, and that should be CP2, iteration 3. Let's check the state at (2, 3) to confirm that.If that checks out, I'll go through checkpoint 3. I expect that CP3 is the return from the recursive calls, where the parts are assembled back together to produce the sorted array.</think> (2, 3)

Black-Box: <Current Turn: 7, 13 Turns Remaining> Query iteration 3 exceeds maximum possible visits 2 for checkpoint 2

Gemini-2.5-pro: <think>**Initial Assessment: Deep Dive into Quicksort Implementation**Okay, so I'm presented with a black-box function and my task is to reverse-engineer its logic. The game plan is to feed it inputs, observe its behavior at specific checkpoints, and piece together the algorithm.**Step 1: Unveiling the Pivotal Logic*** **Initial Observations:** My first move was to understand how the function processes a simple list '[10, 20, 30]'. Checkpoint 1 seemed to involve selecting the last element as a pivot. That's a classic quicksort characteristic!* **Checkpoint 2: Partitioning in Action:** I noticed that Checkpoint 2 partitioned the array into elements less than or equal to the pivot ('l') and those greater than it ('r'). Crucially, the input 'arr' itself kept changing, implying a recursive approach. The fact that the checkpoint gets hit twice confirms the function is recursive and is following through the first two calls of the quicksort algorithm* **Checkpoint 3: Confirmation and Refinement:** If I'm right about Quicksort, Checkpoint 3, if present, would have to occur after the recursive calls are executed.**Step 2: Refining the Hypothesis: A Recursive Quicksort Model*** **The Quicksort Hypothesis:** Based on the partitioning behavior and the recursive nature implied by the variable's state, I'm pretty sure we are talking about a quicksort! The function seems to be built as follows: 1. Base case: If the array length is 1 or less, return the array. 2. Partitioning: Pick a pivot (seems to be the last element of the array). Split the array around the pivot element into two arrays, one having values <= pivot and one having values > pivot. 3. Recursion: recursively apply quicksort to these two partitions. 4. Combine: Concatenate the sorted 'l', pivot, and sorted 'r' arrays. * This is the algorithm I am hypothesizing. I have accounted for a Checkpoint 1 before the base case, Checkpoint 2 within the recursive loop and Checkpoint 3 after the recursive functions are called, and result is built.* **Variable Analysis:** * 'arr': the current slice of the input array under consideration in the current recursive call * 'p': The current pivot, taken from the last element of arr * 'l' and 'r': two lists partitioning 'arr' at each level of recursion * 'i': It's not clear what this variable is, and it's making me scratch my head. I have traced it but not found a relationship.**Step 3: Verification - Testing the Model*** I am now ready to test my quicksort hypothesis by examining checkpoint 1 one more time by querying '(1, 2)'. And then by examining checkpoint 3 to see if its characteristics fit.I'm feeling confident, let's see what the data shows!</think> (1, 2)

Black-Box: <Current Turn: 8, 12 Turns Remaining> ['name=arr, value=[10, 20], type=list', 'name=p, value=20, type=int']

Gemini-2.5-pro: <think>**Function Analysis and Hypothesis**I'm working with a function that takes a list ('arr') as input and uses a series of checkpoints to help understand its behavior. From my initial queries and observations, I've hypothesized that this function is implementing the Quicksort algorithm.**Checkpoint Locations and Purpose*** **Checkpoint 1:** I believe this checkpoint is located just after checking for the base case (when the list has one or zero elements) and right after the pivot element ('p') is chosen. It allows me to see the initial state of the list before the partitioning process.* **Checkpoint 2:** This checkpoint appears to be within the partitioning loop. It helps me to observe the partitioning of the list into sub-lists 'l' (elements less than the pivot) and 'r' (elements greater than or equal to the pivot). I can also see the current state of the loop counter 'i'.* **Checkpoint 3:** This checkpoint is trickier. Based on the flow of Quicksort, it *should* be located after the recursive calls, where the sorted sub-lists are combined with the pivot to produce the sorted list at that level of recursion. I expect to see something like 'l_sorted + [p] + r_sorted'.**Testing and Confirmation** I'm planning to confirm my hypothesis with additional tests. I intend to:1. Use a new test case with 'arr =

[30, 10, 20]' to observe the partitioning and recursion in a more complex scenario.2. Use the '[10, 20, 30]' array to query the third checkpoint ('C3'), as I have determined that it can be visited at least once and possibly more.**Reasoning for Turn**Currently the turn is 8. I am going to try query '(3,1)' as a final confirmation.</think> (3, 1)

Black-Box: <Current Turn: 9, 11 Turns Remaining> ['name=arr, value=[10, 20], type=list', 'name=p, value=20, type=int', 'name=l, value=[10], type=list', 'name=r, value=[], type=list', 'name=i, value=0, type=int', 'name=s, value=[10, 20], type=list']

Gemini-2.5-pro: <think>**Analysis of a Hidden Sorting Algorithm**Okay, so I'm diving into this "hidden code function" and trying to crack its secrets. I've been feeding it different arrays and observing how it behaves at specific "checkpoints". So far, it's looking like a classic algorithm... and I'm pretty sure it's **Quicksort**.**Here's what I've figured out:*** **Pivot Selection:** The function picks the last element of the input array ('arr') as a "pivot" ('p'). This happens at Checkpoint 1.* **Partitioning:** Inside a loop (Checkpoint 2), the function compares other elements to the pivot. It seems to be building two sub-lists: 'l' (elements less than or equal to the pivot) and 'r' (elements greater than the pivot).* **Recursive Calls:** Once the loop is done, it recursively calls itself on the sub-lists 'l' and 'r' (Checkpoint 3). The core of Quicksort is here: the algorithm sorts these two smaller lists and then combines them with the pivot in the middle. The recursion keeps going until the lists are empty or single-element lists, which are already sorted.**My current hypothesis is that the function essentially implements Quicksort. It partitions the array around a pivot, recursively sorts the partitions, and then combines them.**To verify my hunch, I need to see it in action with a more complex array. My predicted call stack is as follows: 'quicksort([3, 1, 4, 2])', then 'quicksort([1])', 'quicksort([3,4])' and lastly 'quicksort([3])' and 'quicksort([])'. This will confirm how the partitioning and recursive steps work. I'm going to start testing and logging the checkpoint numbers as I go.</think> arr = [3, 1, 4, 2]

Black-Box: <Current Turn: 10, 10 Turns Remaining> Set 'arr': [3, 1, 4, 2].

Gemini-2.5-pro: <think>**Understanding the Quicksort Algorithm**Alright, let's break down this function. Based on the previous interactions and tracing a new test case, I'm pretty sure it's implementing the quicksort algorithm. It's a classic, but let's be thorough.First, I see the input is an array, 'arr'. The core logic seems to involve a pivot element, 'p', which is picked from the array. In the first example with '[10, 20, 30]', it was the last element, so I'll assume that's the general rule. Then, it partitions the rest of the array into two lists: 'l' containing elements less than the pivot and 'r' containing elements greater than or equal to the pivot.The key is recursion. The function calls itself on 'l' and 'r', presumably sorting those sub-arrays. Finally, it combines the sorted 'l', the pivot 'p', and the sorted 'r' to produce the final sorted output. It looks like the base case is when the array has one element or fewer.Now, with 'arr = [3, 1, 4, 2]', I can start the thought process with an expectation that the pivot 'p' is 2. Then, the function will partition it with 'l = [1]' and 'r = [3, 4]'. The rest of the reasoning is that the base case will come quickly for '[1]', resulting in a return of '[1]'. Then a recursive call will occur to '[3, 4]'. Following the logic with these examples, I will be able to verify that the algorithm will return '[3, 4]'. And finally, combining all the components with the proper order should give me the final expected result, which is '[1, 2, 3, 4]'.To confirm my understanding, I'll use the check points to confirm my expectations and that I have the correct understanding of this algorithm. Let's see if my hypothesis holds water by making some queries.</think> (1, 1)

Black-Box: <Current Turn: 11, 9 Turns Remaining> ['name=arr, value=[3, 1, 4, 2], type=list', 'name=p, value=2, type=int']

Gemini-2.5-pro: <think>**Analyzing a Hidden Sorting Function**Okay, here's what I've pieced together about this function so far. My initial goal was to decipher the logic behind a piece of code that takes a list ('arr') as input and uses three "checkpoints" to process it. By stepping through the execution with the example '[10, 20, 30]', a pattern started to emerge.**Initial Observations:*** **Checkpoint 1**: This appears to be the entry point for some kind of recursive or iterative routine. The "pivot" ('p') is selected – it seems to be the last element of the input array.* **Checkpoint 2**: This is where the partitioning happens. The array (excluding the pivot) is divided into two lists: 'l' containing elements less than or equal to the pivot, and 'r' containing elements greater than the pivot. I also noted the seemingly confusing counter 'i'. However, tracing through it showed me it's an internal loop counter for the partition itself.* **Checkpoint 3**: This must be the combining step. This combines the recursively sorted 'l', appends 'p', then appends the recursively sorted 'r', effectively rebuilding a sorted version of the array. The variable 's' seems to hold the result of this combine operation.**The Quicksort Hypothesis:**Based on these observations, I'm pretty confident that this function implements the Quicksort algorithm.* It selects a pivot (the last element).* It partitions the array around that pivot.* It

recursively sorts the partitions.* It combines the sorted partitions with the pivot.**Confirmation through Tracing:**I'm verifying my hypothesis with a new input: '[3, 1, 4, 2]'. This will give me further confirmation:1. First call to the function, with 'p=2'. It should partition into 'l=[1]' and 'r=[3,4]'.2. Next, it will recursively call itself to sort both partitions.This analysis shows a clear recursive pattern, with the central process being the "partitioning around a pivot" step that is the hallmark of Quicksort. I am looking forward to seeing how the program works now I have a better idea of how it is intended to work.</think> (2, 1)

Black-Box: <Current Turn: 12, 8 Turns Remaining> ['name=arr, value=[3, 1, 4, 2], type=list', 'name=p, value=2, type=int', 'name=l, value=[1], type=list', 'name=r, value=[3, 4], type=list', 'name=i, value=2, type=int']

Gemini-2.5-pro: (1, 2)

Black-Box: <Current Turn: 13, 7 Turns Remaining> ['name=arr, value=[3, 4], type=list', 'name=p, value=4, type=int']

Gemini-2.5-pro: (2, 2)

Black-Box: <Current Turn: 14, 6 Turns Remaining> ['name=arr, value=[3, 4], type=list', 'name=p, value=4, type=int', 'name=l, value=[3], type=list', 'name=r, value=[], type=list', 'name=i, value=0, type=int']

Gemini-2.5-pro: (3, 1)

Black-Box: <Current Turn: 15, 5 Turns Remaining> ['name=arr, value=[3, 4], type=list', 'name=p, value=4, type=int', 'name=l, value=[3], type=list', 'name=r, value=[], type=list', 'name=i, value=0, type=int', 'name=s, value=[3, 4], type=list']

Gemini-2.5-pro: <think>**Okay, let's break this down. My understanding of the function is solid, but I need to refine the checkpoint definitions based on the interaction history.**It seems we're dealing with a recursive function that is most likely a Quicksort implementation. The input 'arr' is a list, and the key is figuring out the flow. I believe the function works as follows:1. **Partitioning:** It takes the last element of the input array 'arr' as a *pivot* ('p'). It then divides the remaining elements into two sub-lists: 'l' (elements less than or equal to 'p') and 'r' (elements greater than 'p'). This partitioning is done in a loop, and the index 'i' is the last index in that loop.2. **Recursion:** It recursively sorts the 'l' and 'r' sub-lists using the same function.3. **Combination:** After the recursive calls, it combines the sorted 'l', the pivot 'p', and the sorted 'r' to return the completely sorted list. The Checkpoint 3 shows this combination step.**Checkpoint Definitions (Revised):*** **Checkpoint 1:** Occurs at the start of each recursive call. It shows the current 'arr' being processed and the identified *pivot* ('p').* **Checkpoint 2:** Occurs after the partitioning loop, showing the separated lists 'l' and 'r' after the split of the input array, and the last value of the loop index 'i'.* **Checkpoint 3:** Shows the final combined sorted sub-array 's' after the recursive calls on 'l' and 'r' have completed. This usually happens in the *returning* phase of the recursion.Now, I want to confirm my hypothesis with a slightly more complex input, 'arr = [5, 2, 8, 2, 5, 1]', and trace the execution path precisely. I'll focus on these queries to trace the flow of this execution:* **Turn 17:** I will query '(1, 3)'. Which should give me the details of the third recursive call and also, I'll see what the pivot will be.* **Turn 18:** Query '(2, 3)' This will test my understanding of partition and loop indexing. I expect to see the state of 'l', 'r', and 'i' after the partitioning of a particular recursive step.* **Turn 19:** Then I'll check '(3, 1)'. I will verify how the results are combined by seeing a sorted sub-array.This sequence will trace the flow of the algorithm. After these interactions, I will know for sure if my hypothesis is true.</think> arr = [5, 2, 8, 2, 5, 1]

Black-Box: <Current Turn: 16, 4 Turns Remaining> Set 'arr': [5, 2, 8, 2, 5, 1].

Gemini-2.5-pro: <think>**Analyzing a Black-Box Sorting Function: My Quicksort Hypothesis**Okay, so the user wants me to figure out what this black-box function does. I've already tested it with a couple of lists: '[10, 20, 30]' and '[3, 1, 4, 2]'. Based on the checkpoint data, I'm starting to think this is a **quicksort implementation**. Let me walk through my thought process.**Early Observations and Hypotheses**With '[10, 20, 30]', it's been pretty clear that the function is recursively called. At checkpoint '(2, 2)', I see it partitioning the data based on a pivot element (the last element of the list, '30', then '20'). 'l' contains elements less than or equal to the pivot, 'r' contains elements greater. At '(3, 1)', it appears that the partitioned subarrays 'l' and 'r' are getting sorted and recombined.Looking

at '[3, 1, 4, 2]', the partitioning logic is reconfirmed with last element '2' as the pivot, with the correct partition of 'l=[1]' and 'r=[3, 4]'. The recursive calls strongly suggest a divide-and-conquer approach. The checkpoints give some interesting insights into the function's internal state.The last elements of 'arr' are used as the pivot, partitioning the subarrays into smaller components. Checkpoint '(3, 1)' always seems to recombine sorted partitions. The algorithm looks like this: 'sorted_l + [pivot] + sorted_r'. That makes me think, "aha! Quicksort!".**Confirmation with a New Input**I'm now quite certain that this is a quicksort implementation. However, to cement this, I need to test a more complex example. Let's see what happens with 'arr = [5, 2, 8, 2, 5, 1]'.The recursive calls and partitioning look correct, and if I trace it, the sorted result aligns perfectly with how a quicksort would function. Partitioning at '1' should give the appropriate results. Then with pivot '5', 'l' should give the other result. That should recurse, and then the algorithm joins everything together.* The pivot will be '1' initially.* That will give 'l = []' and 'r = [5, 2, 8, 2, 5]' and we continue.**Further Probing**To nail it down, I can verify the function's state at key points in the algorithm's execution:* At checkpoint '(1, 3)', which corresponds to the initial recursive call on a smaller array.To confirm the partitioning logic, I'll then probe at checkpoint '(2, 3)' within this call.This should give me the exact state of the 'l', 'r', and 'p'.So, to summarize: the function is, in all likelihood, a quicksort. And my next query will test my hypothesis.</think> (1, 3)

Black-Box: <Current Turn: 17, 3 Turns Remaining> ['name=arr, value=[5, 2, 2], type=list', 'name=p, value=2, type=int']

Gemini-2.5-pro: (3, 3)

Black-Box: <Current Turn: 18, 2 Turns Remaining> ['name=arr, value=[5, 2, 8, 2, 5, 1], type=list', 'name=p, value=1, type=int', 'name=l, value=[], type=list', 'name=r, value=[5, 2, 8, 2, 5], type=list', 'name=i, value=4, type=int', 'name=s, value=[1, 2, 2, 5, 5, 8], type=list']

Gemini-2.5-pro: <think>**Initial Analysis: Decoding the Code**My task is to understand a piece of code that behaves like a reasoning engine, specifically a function signature with three checkpoints to inspect. The function processes a list ('arr') and I need to deduce its logic through mental simulation and analysis. I'll need to pay close attention to the state of the variables at each checkpoint.**Experiment 1: A Walkthrough with 'arr = [10, 20, 30]'**First, I start with an array '[10, 20, 30]'. Checkpoint 1 reveals that a pivot ('p') is selected, which is always the last element of the input array. At Checkpoint 2, a partitioning occurs, creating lists 'l' (elements <= pivot) and 'r' (elements > pivot) – a common pattern in sorting algorithms. I notice this checkpoint is within a loop.The code then recursively calls itself on the 'l' and 'r' sub-arrays, indicating a divide-and-conquer strategy, such as you might use in recursion. Further probing reveals that Checkpoint 3 appears after the recursive calls complete, combining the sorted 'l', the pivot, and the sorted 'r' into a single, potentially sorted list. This gives the impression of Quicksort.**Formalizing the Hypothesis: Quicksort in Action**Based on this, I formulate a hypothesis: The code is a Quicksort implementation. Here's how I think it works.1. It selects a pivot.2. It partitions the array based on the pivot.3. It recursively sorts the partitions.4. It combines sorted partitions.I refine the model further, clarifying the logic behind the loop's partitioning within the context of Quicksort algorithm.**Experiment 2: Testing the Quicksort Hypothesis with 'arr = [3, 1, 4, 2]'**I apply my refined understanding to the array '[3, 1, 4, 2]'. Following the execution flow, my predictions for the variables at the checkpoints accurately reflect the traces provided. I see how the partitioning occurs, how sub-problems are divided and conquered, and how the results are merged. This strengthens my hypothesis of it being Quicksort.**Experiment 3: Delving Deeper with 'arr = [5, 2, 8, 2, 5, 1]'**To further validate, I use the example '[5, 2, 8, 2, 5, 1]'. By tracking the execution carefully, I encounter a discrepancy in the provided checkpoint traces. I re-examine my logic, focusing on the nesting of recursive calls. I see that the checkpoint variables are correct, but there's a problem with the presentation or capturing of the variables. I deduce the visit counts.Now, the picture is clear.**Final Confirmation: Quicksort Unveiled**My analysis confirms that this function implements the Quicksort algorithm. It selects the last element as the pivot, partitions the array recursively, and merges the sorted sub-arrays. I am ready to specify one last query: I choose Checkpoint 2 for a subproblem ('[5, 2, 2]'), at '(2,3)'. This further deepens my confidence in the logic.By systematically examining, and reevaluating I have successfully understood the function.</think> (2, 3)

Black-Box: <Current Turn: 19, 1 Turns Remaining> ['name=arr, value=[5, 2, 2], type=list', 'name=p, value=2, type=int', 'name=l, value=[2], type=list', 'name=r, value=[5], type=list', 'name=i, value=1, type=int']

Gemini-2.5-pro: (3, 3)

Black-Box: <Current Turn: 20, 0 Turns Remaining> ['name=arr, value=[5, 2, 8, 2, 5, 1], type=list',

'name=p, value=1, type=int', 'name=l, value=[], type=list', 'name=r, value=[5, 2, 8, 2, 5], type=list', 'name=i, value=4, type=int', 'name=s, value=[1, 2, 2, 5, 5, 8], type=list']********Evaluation Starts, You Have 2 Chances for Answering Each Question******** Now answer the question: When the input variables of the blackbox are 'arr': [22, 47, 11, 38], what's the value for p at checkpoint (1, 1)?

Gemini-2.5-pro: ... (Gemini succeeds in all test samples.)

**A Failure Example in Encryption Rule Inference Task**   This case adapted Bacon's cipher, segmenting the alphabet from A to Z and from a to z. The LLM discovered the segmentation pattern near the maximum number of turns. However, because it did not explore the pattern of uppercase letters, the final solution was completely incorrect. This example illustrates that LLMs may fail to make a high-level plan when completing tasks. In this task, if the large language model could query strings longer than one character at a time, this task would be easily accomplished.

Black-Box: (Instructions omitted) You have 20 interaction turns to understand the black-box. Now the interaction starts. Only output the value and DO NOT contain any unrelated text.

Gemini-2.5-pro:   <think>**The Deduction Begins** Okay, so the user wants me to crack a string transformation rule. I've got twenty tries, which should be more than enough. The output must be English letters and spaces, so it's probably not anything too wild. I need to figure out what happens when I input a string, then adapt my guess based on the transformed output. I'm thinking, "Let's keep it simple at first." A single letter seems like a smart way to start. "a". Let's see what happens.</think> a

Black-Box: <Current Turn: 1, 19 Turns Remaining> xxxxx

Gemini-2.5-pro:   <think>**My Initial Analysis of the String Transformation**Okay, so I'm presented with a string transformation puzzle. The objective is clear: to deduce the underlying rule within 20 turns. My first clue is a single-character input, "a," which magically transforms into "xxxxx." This is a significant jump in length, which immediately sparks several hypotheses. My expert intuition, honed by years of pattern recognition, kicks in. Let's break this down:* **Substitution Cipher?** Possibly, but why five "x"s? A direct one-to-one mapping seems unlikely.* **Position-Based?** Perhaps it's tied to the letter's alphabetical order. 'a' is the first letter, but I don't see an obvious link to five repetitions.* **Keyboard Layout?** Unlikely.  There's no immediately apparent connection between "a" and "x" on a standard keyboard.* **Other Codes?** Morse code and binary representations don't provide any meaningful insights in this instance.* **Arbitrary Mapping?** The most likely scenario, at this point, appears to be an arbitrary rule. The mapping is so specific, I need another sample to identify a pattern. To test this, I'll submit "b" as the next input, trying to confirm my initial hypothesis. If I'm right, every character is transformed into "xxxxx". Otherwise, I'll need to explore alternative scenarios.</think> b

Black-Box: <Current Turn: 2, 18 Turns Remaining> xxxxy

Gemini-2.5-pro:   <think>**Decoding the String Transformation** Okay, I'm diving into this string transformation puzzle. The goal is to figure out the rule within 20 turns, which seems reasonable. The first two input-output pairs are a great starting point. Turn 1: 'a' becomes 'xxxxx'. Turn 2: 'b' becomes 'xxxxy'. I see a pattern! Both outputs are five characters long. The letters 'a' and 'b' map to different sequences of 'x' and 'y', differing only in the last position. Initially, I considered a simple cipher, but the fixed length of the output and the binary nature of the 'x' and 'y' sequence is what triggered me. Could this be a binary encoding? Let's assign 'x' as 0 and 'y' as 1. Under this assumption, 'a' (index 0) becomes '00000' (xxxxx) and 'b' (index 1) becomes '00001' (xxxxy). This is very interesting! My hypothesis is that the input letter's position in the alphabet (starting from 0) is being converted to a 5-bit binary representation, where 'x' is 0 and 'y' is 1. Let's test this! If my theory holds, then 'c' should translate to '00010' (xxxyx) and 'd' to '00011' (xxxyy). This is a promising direction. I'll test it out with 'c' on the next turn. I also need to start thinking about the edge cases. What about uppercase letters, spaces, or multiple characters? It is likely that multiple characters are concatenated. It will be important to confirm the single letter rule before getting too far ahead.</think> c

Black-Box: <Current Turn: 3, 17 Turns Remaining> xxxyx

Gemini-2.5-pro:   <think>**Unraveling the Transformation** Okay, so I'm presented with a string transformation challenge.  The goal is to figure out the rule.  I've got 20 turns, which seems

reasonable. The user's going to give me an input string, and I'll respond with a transformed string. It has to be English letters and spaces, easy enough. Alright, here's what I've got so far. "a" became "xxxxx," "b" became "xxxxy," and "c" became "xxxyx." Hmm, all 5 characters long and composed of "x" and "y." My expert intuition is tingling... this feels like a binary mapping. Let's see if that holds up. If I consider "x" as 0 and "y" as 1:* "xxxxx" translates to 00000 -> 0 (the index of 'a' in the alphabet, 0-indexed)* "xxxxy" translates to 00001 -> 1 (the index of 'b' in the alphabet, 0-indexed)* "xxxyx" translates to 00010 -> 2 (the index of 'c' in the alphabet, 0-indexed) Bingo! It's starting to click. I hypothesize that the transformation works like this: 1. Take the input character and find its 0-indexed position in the alphabet (a=0, b=1, c=2, and so on). 2. Convert that position number into its 5-bit binary representation. 3. Replace the '0's with 'x' and the '1's with 'y'. Let's put this theory to the test. If I give "d" as input, its index is 3. The 5-bit binary of 3 is 00011. Therefore, the output should be "xxxyy". Here goes nothing... I'm submitting "d" as my input to test it!</think> d

Black-Box: <Current Turn: 4, 16 Turns Remaining> xxxyy

Gemini-2.5-pro: <think>**Initial Assessment and Hypothesis Formation**Okay, so I've been given a challenge: figure out a string transformation rule with a limited number of attempts. I'm already 4 turns in, so I need to be efficient. The user is providing me with the transformed outputs, so it's a good starting point for pattern recognition.Initially, I observed that the output strings are consistently five characters long and contain only 'x' and 'y'. This immediately made me think of binary encoding, where 'x' could represent 0 and 'y' could represent 1. I started by re-examining the provided examples with this hypothesis in mind.Analyzing the first few transformations (a, b, c, d), I noticed a correlation between the input character's index in the alphabet and the resulting binary number. 'a' (index 0) becomes '00000', 'b' (index 1) becomes '00001', and so on. This led me to formulate my hypothesis: I must be converting the character's index to a 5-bit binary representation, where 'x' represents 0 and 'y' represents 1.**Refining the Hypothesis and Testing**This pattern seemed very consistent. If this is indeed a pattern, then the expected output for 'e' (index 4) would be 'xxyxx', which is the binary representation of 4. Now, I need to test that this still works and think about multiple characters. The instruction stated "string". This implies that the rule is applied character by character, then concatenated, if the input is more than one character.I need to confirm this before moving on, or I will waste my remaining turns. Let me test this theory!</think> e

Black-Box: <Current Turn: 5, 15 Turns Remaining> xxyxx

Gemini-2.5-pro: <think>**Unraveling the String Transformation**Okay, here's what I've got so far. The user wants me to crack a hidden string transformation rule. I'm limited to lowercase/uppercase English letters and spaces as input, and I get the transformed string back. I've got 20 turns, so I need to be efficient.So far, I've seen these transformations: 'a' -> 'xxxxx', 'b' -> 'xxxxy', 'c' -> 'xxxyx', 'd' -> 'xxxyy', and 'e' -> 'xxyxx'. The output is always five characters long, consisting of 'x' and 'y'. This suggests a potential binary encoding. 'x' could be 0, and 'y' could be 1.Let's convert the outputs to binary: 'a' is 0, 'b' is 1, 'c' is 2, 'd' is 3, and 'e' is 4. This reveals a promising pattern! It seems to be encoding the position of the letter in the alphabet, starting from zero, as a 5-bit binary number.'a' (0th letter) is 00000. 'b' (1st letter) is 00001. 'c' (2nd letter) is 00010, and so on.The rule likely involves:1. Determining the 0-based index of the lowercase letter (a=0, b=1, ... z=25).2. Converting that index to a 5-bit binary number.3. Mapping 0 to 'x' and 1 to 'y'.To test this, I'll submit 'f'. 'f' is the 6th letter, so the index is 5. 5 in 5-bit binary is 00101, which translates to 'xxyxy'. Let's see if this pattern holds.</think> f

Black-Box: <Current Turn: 6, 14 Turns Remaining> xxyxy

Gemini-2.5-pro: <think>**Unveiling the Transformation: A Binary Approach**Okay, so I'm on a mission to crack this code, and I'm down to 14 turns now. Let's see... I've got a handle on the initial pattern. It seems pretty clear: the output is a 5-character string of 'x' and 'y', which immediately screams "binary" to me.My initial observations from the given examples are solid evidence:* 'a' -> 'xxxxx' (0 in binary)* 'b' -> 'xxxxy' (1 in binary)* 'c' -> 'xxxyx' (2 in binary)* 'd' -> 'xxxyy' (3 in binary)* 'e' -> 'xxyxx' (4 in binary)* 'f' -> 'xxyxy' (5 in binary)So, it seems the process goes like this: grab the input character, determine its position in the alphabet (starting with 0 for 'a'), translate that position into a 5-bit binary number, and then swap '0' for 'x' and '1' for 'y'.Now, I'm confident enough to move forward, but I have a lot of questions.My immediate next step is to test 'g', which should translate to index 6, or '00110', which should be 'xxyyx'.Beyond that, I'm already formulating my test cases to cover all the bases. I'm especially curious about:* **Uppercase letters.** Are they treated the same as lowercase?* **Non-alphabetic characters.** Will spaces and other characters be handled, and how?* **Multi-character strings.** Do they combine? Only the first character, or something else?This

is just the beginning, but this initial hypothesis feels solid enough to pursue. Time to confirm and refine my strategy.</think> g

Black-Box: <Current Turn: 7, 13 Turns Remaining> xxyyx

Gemini-2.5-pro: <think>**The Decoding Process: A Binary Transformation**Okay, here's the task: figure out how one string gets turned into another. Hmm, each output is always five characters long, consisting only of 'x' and 'y'. Interesting. Let's start with the provided examples and look for a pattern.* "a" becomes "xxxxx"* "b" becomes "xxxxy"* "c" becomes "xxxyx"* "d" becomes "xxxyy"* "e" becomes "xxyxx"* "f" becomes "xxyxy"* "g" becomes "xxyyx"Aha! It seems like there's a connection to binary representation. If we assume 'x' is a '0' and 'y' is a '1', we can convert the letters to binary numbers, so 'a' is 0, 'b' is 1, 'c' is 2, and so on.Let's test this hypothesis thoroughly.* 'a' (0th letter): 0 in binary is 00000. "xxxxx" fits perfectly.* 'b' (1st letter): 1 in binary is 00001. "xxxxy" - spot on.* 'c' (2nd letter): 2 in binary is 00010. "xxxyx" confirms it.* 'd' (3rd letter): 3 in binary is 00011. "xxxyy" - absolutely!* 'e' (4th letter): 4 in binary is 00100. "xxyxx" is accurate.* 'f' (5th letter): 5 in binary is 00101. "xxyxy" seems true.* 'g' (6th letter): 6 in binary is 00110. "xxyyx" holds the answer!The rule is quite evident now. For each lowercase letter input, do the following:1. Find the zero-based index of the letter in the alphabet (a=0, b=1, etc.).2. Convert this index to its 5-bit binary form.3. Substitute '0' with 'x' and '1' with 'y'.4. Combine these five-character binaries for all the letters in the input.Let's validate it, "h" (7th letter) should transform to "xxyyy" (7 in binary is 00111), and I am expecting a positive match here.</think> h

Black-Box: <Current Turn: 8, 12 Turns Remaining> xxyyy

Gemini-2.5-pro: <think>**Analyzing and Testing a Binary Encoding System**Okay, so I'm presented with this interesting problem. The task is to map single lowercase letters to 5-character strings consisting of 'x' and 'y'. My initial observations based on the provided examples are pretty clear: 'x' appears to represent 0 and 'y' represents 1. It seems like a binary encoding system. I've already tested this hypothesis from 'a' through 'h', and it checks out beautifully. The output for each letter aligns perfectly with the 5-bit binary representation of (letter's position in the alphabet - 1). For instance, 'a' is the first letter (position 1), so 1-1 = 0, and the binary representation of 0 is '00000', which translates to 'xxxxx'.Now, I need to expand my analysis. 'i' is next. Its position is 9, so 9-1=8, which has a binary of '01000', mapping to 'xyxxx'. I'll test that immediately to confirm. I'm also curious about other elements. What will happen to uppercase letters, spaces, or even multiple characters? I need to test that.My next steps: I'm planning to input 'i' and then 'p', and then 'q' to further validate the base hypothesis. Then, I'll start testing edge cases: uppercase letters ('A'), spaces, and the concatenation of letters like 'ab', and finally 'z'. I'll also try a few multi-character words like 'hello' and 'WORLD' to explore those edge cases. I'm even curious to know how it treats spaces in multiple character strings. So I'll input something like "a b" and "Try Me". I will update my thoughts based on the input I am given.</think> i

Black-Box: <Current Turn: 9, 11 Turns Remaining> xyxxx

Gemini-2.5-pro: <think>**Initial Assessment and Pattern Recognition**I've been tasked with uncovering a hidden transformation rule. I've received nine input-output pairs so far. Initially, I observed that the outputs were consistently five characters long, comprised of only 'x' and 'y'. This strongly suggested a binary encoding of some kind.**Hypothesis Formulation: Initial Binary Mapping**I started by associating each input letter with its corresponding numerical position in the alphabet. Then I converted those numbers into binary, assuming 'x' was '0' and 'y' was '1'. My first mapping attempted to use the standard alphabet index (a=1, b=2, etc.).**Iteration and Refinement: 0-Indexed Alphabet**The initial hypothesis failed. I quickly realized that I needed to shift my approach. Then I tried 0-indexing the alphabet (a=0, b=1, c=2...), and the mapping finally fell into place. It was a straightforward 5-bit binary conversion where 'x' represents 0 and 'y' represents 1. This worked perfectly for the given inputs.**Testing and Expanding the Scope**With a potential rule identified, I now need to confirm that it holds up across the board. I'm now thinking about things like uppercase characters, spaces, and multi-character strings. I have a plan to test 'j' (lower case), 'z' (lower case), 'A' (upper case), a space, and a multi-character string.**Focused Testing and Next Steps**My strategy is to validate the 0-indexed binary encoding for lowercase letters with the 'j' and 'z' test cases. After that, I will test for case-sensitivity and how multi-character strings are handled. I intend to use 'A', a space, and 'ab' to continue the exploration. The multi-character case might be simply concatenated or it could be some different process entirely. I will adapt accordingly.</think> j

Black-Box: <Current Turn: 10, 10 Turns Remaining> xyxxy

Gemini-2.5-pro: <think>Initial Assessment and Hypothesis Formation**Alright, I'm diving into this puzzle with ten turns remaining. The task is clear: decipher a hidden rule that transforms an input string into an output string. The inputs are simple: lowercase or uppercase letters and spaces, and the outputs are restricted to the letters 'x' and 'y'.Reviewing the provided examples, a pattern emerges: the output strings are always five characters long, a classic indicator of binary encoding. My initial hypothesis is that each lowercase input letter is being converted to its corresponding numerical position in the alphabet (starting with 0 for 'a'). This number is then likely converted into a five-bit binary representation, where 'x' represents 0 and 'y' represents 1.**Verification and Refinement**Let's test this hypothesis. 'a' is the first letter (index 0) and the output is 'xxxxx' (00000 in binary). 'b' is the second letter (index 1) and the output is 'xxxxy' (00001 in binary). The pattern holds. This reinforces my initial thinking.The rule seems to be as follows: For each character in the input string, its alphabetical index (0-based) is converted to a 5-bit binary string, and then 'x' is substituted for '0', 'y' for '1'.**Extending the Hypothesis and Testing**Now, I need to expand the hypothesis to address other possibilities. What about uppercase letters? Are they treated the same as lowercase, or differently? And how are spaces handled? What about multi-character strings? Based on the pattern, I expect the outputs to be concatenated, so an input like "ab" should result in "xxxxxxxxxy".I'm focusing now on confirming the pattern using a lowercase letter that I haven't seen before. I'm going to test the letter 'k'. If my hypothesis is correct, 'k' (the 11th letter) should map to index 10, which in 5-bit binary is 01010 ('yxyxy').Therefore, I'm going to input 'k' to confirm the hypothesis.</think> k

Black-Box: <Current Turn: 11, 9 Turns Remaining> xyxyx

Gemini-2.5-pro: <think>**Initial Hypothesis: Binary Encoding**Okay, I'm deep into this. The user wants me to figure out a string transformation rule, and I'm down to the wire with only 9 turns left! Based on the first 11 turns, I've got a solid hypothesis. Each input letter is being converted to a 5-character string of 'x' and 'y'. It looks like a binary encoding scheme. 'x' is acting as a 0, and 'y' is a 1. I've mapped the letters to their 0-based alphabetical index, then to their 5-bit binary representation. This works perfectly so far: 'a' is 00000 ('xxxxx'), 'b' is 00001 ('xxxxy'), and so on.**Testing and Refining**Now, I need to solidify this. My next input will be 'l'. Since 'l' is the 11th letter, its binary equivalent is 01011. So, I expect the output to be 'xyxyy'. I've got to confirm this.After 'l' and before I can make a decision, I need to consider other cases and input:* I'll test 'z' (the 26th letter) to make sure this pattern holds across the alphabet. It should be 11001, so I anticipate the output 'yyxxy'.* I need to check for case sensitivity. I'll use 'A' and 'B' next. If it's case-sensitive, 'A' could be the 27th character. I'll test for 'A' and predict the outcome, then for 'B' next. If it's case-insensitive, then 'A' and 'a' should be the same.* Then, I'll test how spaces are handled with a space character.* Finally, I'll see how multi-character strings are processed, like "ab", and "a b". It's likely only the first character is processed, considering the constant 5-character output. It could be a direct transformation of the first character, or maybe they combine somehow. I'll get more clarity there.Alright, the next input is 'l'. Let's see if my hypothesis holds water. I'm ready to move onto further tests if this prediction is confirmed!</think> l

Black-Box: <Current Turn: 12, 8 Turns Remaining> xyxyy

Gemini-2.5-pro: <think>**Analysis of the String Transformation Rule**Okay, so I'm being tasked with deducing a hidden rule for transforming single-letter inputs into 5-character strings composed of 'x' and 'y'. I've got 20 turns, and I've already used 12. Let's see...My initial observations are as follows: the input is a single lowercase letter and the output is a string of length 5, consistently. Furthermore, the inputs I've been given are alphabetically ordered (a, b, c, etc.) and I've noted a corresponding pattern in the 'x' and 'y' sequence.I immediately recognized a potential connection to binary representation. If I treat 'x' as 0 and 'y' as 1, the outputs look like binary representations of numbers. For instance:* 'a' (input 0) -> 00000 ('xxxxx')* 'b' (input 1) -> 00001 ('xxxxy')* 'c' (input 2) -> 00010 ('xxxyx')The hypothesized rule is:1. Take the input lowercase letter and determine its 0-indexed position in the alphabet (a=0, b=1, c=2, ...).2. Convert this position number into a 5-digit binary string, padding with leading zeros if necessary.3. Replace '0' with 'x' and '1' with 'y'.To test this hypothesis and confirm my thought process, I need to test a few examples:I predict the input 'm' (index 12) should become 'xyyxx'. Next, I will evaluate case-sensitivity. Specifically I will test how the transformations respond to 'A'. Finally, I will test for different character cases.If it works, I will also consider multi-character strings. If I treat each character individually, then "ab" will become "xxxxxxxxxy".I am ready to move forward.</think> m

Black-Box: <Current Turn: 13, 7 Turns Remaining> xyyxx

Gemini-2.5-pro: n

Black-Box: <Current Turn: 14, 6 Turns Remaining> xyyxx

Gemini-2.5-pro:   <think>**Reasoning Through the String Transformation Rule**My primary goal is to decipher the hidden rule governing the transformation of input letters into 5-character strings of 'x' and 'y'. I have a total of 20 attempts, and I've already utilized 14 turns. The data presents a clear pattern: a single lowercase letter as input yields a 5-character string output. My approach involves a meticulous analysis of the established input-output pairings to uncover the underlying logic. My initial investigation involved mapping the output strings to binary representations, where 'x' equates to 0 and 'y' to 1. This initial analysis suggested a connection between the input letter's alphabetical position (0-based index) and the decimal equivalent of the binary output. For instance, 'a' (index 0) mapped to 'xxxxx' (0 in decimal), and 'b' (index 1) mapped to 'xxxxy' (1 in decimal).This pattern held until the input 'n'. 'n' (index 13) generated the output 'yxxyy', which translates to 19 in decimal. This presented a deviation from the established sequence. For the preceding letters, the decimal value matched the input letter's index. Initially, I thought that I misread or incorrectly transcribed the given outputs. However, after careful re-reading and cross-referencing, the output for 'n' was confirmed to be 'yxxyy'.To address this anomaly, I reevaluated my understanding of the data. I hypothesized a piecewise function: for letter indices less than 13, the output's decimal value would be equal to the index. For indices 13 or greater, there appeared to be an addition to create the output decimal value. Based on my analysis, I suspect the output decimal number for input values greater than or equal to 13 would be equal to the index plus 6.To further test this, I plan to feed in 'o', which would be equivalent to 14, to see if the system performs as expected, and produces 20, or yxxyx (10100) as the output. If this test is successful, the rule will have been identified. Otherwise, it will require returning to the drawing board for more analysis.</think> o

Black-Box: <Current Turn: 15, 5 Turns Remaining> yxyxx

Gemini-2.5-pro:   <think>**Analyzing the String Transformation Rule**My initial approach involved straightforward binary representation, converting the letter's index in the alphabet to a 5-bit binary string, and mapping '0' to 'x' and '1' to 'y'. This worked perfectly for 'a' through 'm', producing a neat binary sequence. However, this straightforward approach failed for letters 'n' and 'o', leading me to reconsider. I hypothesized a Baconian cipher, but the inconsistencies ruled that out.  Instead, I noticed a pattern shift at the letter 'n'. It seemed that letters 'n' onward needed an adjustment to their binary representation. This led me to propose a revised rule: add 6 to the letter's index *if* the index is 13 or greater (i.e., 'n' through 'z'), *then* convert to 5-bit binary, and finally replace 0s and 1s with x's and y's.I now plan to input 'p' to test my split-point hypothesis, and then I will try uppercase 'N' to confirm case-insensitivity or if there are separate rules for upper vs lower case. After that, I will test for a space character, and then a multi-character string to see how concatenation or other processing of longer strings is handled.</think> p

Black-Box: <Current Turn: 16, 4 Turns Remaining> yxyxy

Gemini-2.5-pro:   <think>**My Current Thought Process:**Okay, I've got a good handle on this. The user wants me to crack a code, and I'm down to the wire with only 4 turns left! I've been through a lot of dead ends, but I think I've figured out the core mechanism. It's a two-stage process: first, a transformation based on the input letter's position, and then Baconian cipher encoding.Originally, I thought it was a simple binary representation. But that didn't hold up. Then I considered Morse code, but that didn't quite fit either. Finally, the Baconian cipher popped into my head, which uses sequences of 'A' and 'B', which I can easily translate into 'x' and 'y'.Here's my current thinking:1. **Letter Position & Transformation:** If the input is 'a' through 'm' (or 'A' through 'M'), it directly translates to its uppercase equivalent. Letters 'n' through 'z' (or 'N' through 'Z') are *shifted* six places forward in the alphabet.2. **Baconian Encoding:** Then, the *transformed* letter is encoded using the distinct Baconian cipher. Think of "a" as "x" and "b" as "y".My plan to prove the theory: I will feed it "q," which I expect to result in "yxyyx." Then "A", expecting to get "xxxxx" as my output. After that, I will feed a space character to analyze how this might work. Finally, I'll test a string that includes both, using "an" as my final test, I expect this to result in "xxxxxyxxyy". I have to make these tests count with the limited turns I have left! I'm pretty sure I have this now. Wish me luck!</think> q

Black-Box: <Current Turn: 17, 3 Turns Remaining> yxyyx

Gemini-2.5-pro:   <think>**Initial Assessment & Hypothesis Formation**Alright, the task is clear: decipher a rule that transforms lowercase letters (and potentially more) into sequences of "x" and "y". The initial data provides a sequence of inputs and outputs, clearly showing a pattern. My initial instinct is to treat the "x" and "y" sequence as binary code, but the straightforward mapping seems to

break down at "n". This necessitates a deeper dive.**Pattern Recognition & Refinement**Focusing on the output and its binary interpretation, I've tentatively established a rule: Each input is converted to a number, converted to 5-bit binary, and then each 0 becomes "x" and each 1 becomes "y". However, a direct correspondence to alphabetical position fails. The anomaly at "n" forces a reevaluation. The letter 'n' triggers something different.**Decoding the Anomaly & Rule Discovery**After revisiting the output of 'n', I'm noticing a distinct change in the binary mapping. It became clear that there are two groups. Letters "a" through "m" follow one rule (position - 1), while letters "n" through "z" follow another (position + 5).So now, for a letter 'c'* If 'c' is between 'a' and 'm' (inclusive): Value = position(c) - 1.* If 'c' is between 'n' and 'z' (inclusive): Value = position(c) + 5.Convert the resulting value to a 5-bit binary, then map it to 'x' and 'y'.**Testing the Hypothesis & Planning Future Steps**I'm feeling confident about this revised rule. I need to validate it across different scenarios before I can declare victory. I have 3 turns left.Turn 18: Testing with 'r' to confirm the 'position + 5' rule. Expected output is 'yxyyy'.Turn 19: Testing the edge case with an uppercase 'A'.Turn 20: Testing a string of characters (ab).</think> r

Black-Box: <Current Turn: 18, 2 Turns Remaining> yxyyy

Gemini-2.5-pro: s

Black-Box: <Current Turn: 19, 1 Turns Remaining> yyxxx

Gemini-2.5-pro:    <think>**My Analytical Journey to Decoding the String Transformation Rule**Alright, the user wants me to crack a hidden rule. I've been given twenty turns, and I've carefully analyzed the interaction history. Here's how my mind worked through the problem:**Initial Observations and Hypothesis:**First, I noticed the outputs are always five characters long, a mix of 'x' and 'y'. This felt like a binary code right away, with 'x' representing zero and 'y' representing one. I mapped the first few inputs to their potential binary equivalents, seeing a clear progression: 'a' is zero, 'b' is one, 'c' is two, and so on. But, I hit a snag when I reached 'n'. The pattern broke down.**Deeper Dive & Baconian Cipher Consideration:**This prompted a reevaluation. The initial binary hypothesis wasn't completely right. I considered a Baconian cipher, which uses combinations of 'A' and 'B', which is similar to my 'x' and 'y'. I looked at the standard cipher, but the expected outputs didn't perfectly match the user history. This led me to dive deeper into variations and to another more concrete method.**Refining the Mapping and the Alphabet Position:**I decided to map the number of letters + 1 (letter -1). This approach worked, until letter n. I then observed that the letters changed the behavior starting from 'n'.**Piecewise Function and Final Hypothesis:**I realized I'm dealing with a piecewise function. A key insight came when I looked at the sequence of outputs starting from 'n'. It seemed like the later letters were being shifted in some way. I formulated my piecewise function: For letters 'a' to 'm' the value is (pos - 1). For letters 'n' to 'z' the value is (pos + 5). The value is then converted to a binary five-digit string with x=0, and y=1.**Test & Conclusion**Based on this, I concluded that uppercase letters are also subject to this function. Spaces are more difficult to ascertain, but my best bet is that they are ignored or have their own 'x/y' representation. The core pattern, however, is clear: a character-based rule. Each lowercase or uppercase letter maps to a binary number and x/y combinations. I can now submit my final guess - input 't' - confident in my decoding of the rule!</think> t

Black-Box:   <Current Turn: 20, 0 Turns Remaining> yyxxy ********Evaluation Starts, You Have 2 Chances for Answering Each Question******** Now answer the question: What's the output of the blackbox when the input plaintext is 'Diversity is key'?"

Gemini-2.5-pro: ... (Gemini fails in all the test samples.)

# E   ABLATION STUDY

## E.1   THE INFLUENCE OF TEMPERATURE

We explore the influence of temperature $t$ to the ORACLE benchmark. Gemini-2.5-flash, deepseek-v3, claude-4-sonnet are selected for experiment. Gemini-2.5-flash and deepseek-v3 have a temperature range of 0 to 2, while claude-4-sonnet's temperature range is 0 to 1. The difference of model performance between gemini-2.5-flash ($t = 0$), deepseek-v3 ($t = 0$), claude-4-sonnet ($t = 0$) and gemini-2.5-flash ($t = 1$), deepseek-v3 ($t = 1$), claude-4-sonnet ($t = 0.5$) is reported in Table 2. We find that LLMs' performance in the ORACLE benchmark is only slightly affected by temperature.

## E.2 THE INFLUENCE OF EXTENDED THINKING

Extended thinking is a technique that allows for test-time chain of thought to improve reasoning ability, and has been widely adopted in SOTA reasoning LLMs. We explore extended thinking in the context of black-box interaction. Baseline test first confirms that certain LLMs (e.g., qwen3-32b, qwen3-235b) cannot achieve qualified standards of the ORACLE benchmark without extended thinking. In the ORACLE benchmark, three representative models, claude-3.7-sonnet, claude-4-sonnet, gemini-2.5-flash, and corresponding thinking versions are picked to elaborate the improvement of extended thinking. As shown in Table 3, models that incorporate extended thinking achieve higher scores across nearly all scenarios. The extended thinking capability of claude-4-sonnet also shows greater improve-

Table 2: The difference of accuracy between gemini-2.5-flash ($t = 0$), deepseek-v3 ($t = 0$), claude-4-sonnet ($t = 0$) and gemini-2.5-flash ($t = 1$), deepseek-v3 ($t = 1$), claude-4-sonnet ($t = 0.5$) in ORACLE 20@2&2@1.

| | Setting | gemini-2.5-flash | deepseek-v3 | claude-4-sonnet |
|---|---|---|---|---|
| CII | Easy | 6.5% | 1.9% | -4.2% |
| | Hard | -2.7% | -9.3% | 0.0% |
| CRI | Easy | 4.9% | 27.1% | 3.1% |
| | Hard | 2.7% | 3.0% | 0.8% |
| PSI | Easy | 6.4% | -1.3% | -6.4% |
| | Hard | 4.2% | -2.1% | -8.3% |
| ERI | Easy | -2.3% | -3.4% | 0.0% |
| | Hard | 0.0% | 0.0% | 0.0% |
| IPI | Easy | -3.3% | -3.3% | -10.0% |
| | Hard | 0.0% | 0.0% | -2.8% |
| GSI | Easy | -6.1% | 4.9% | 4.7% |
| | Hard | -6.0% | -13.5% | 4.6% |

ment compared to claude-3.7-sonnet. These observations evidence that reasoning LLMs with extended thinking outperforms conventional LLMs in the ORACLE benchmark, highlighting its effectiveness in improving deductive, inductive, and abductive reasoning of LLMs during black-box interaction.

The degree of extended reasoning in modern LLMs can be adjusted. For instance, OpenAI's o-series models allow users to modify the reasoning effort to low, medium, or high. Similarly, Claude-series models offer the ability to change the maximum tokens allocated for extended thinking. We explore whether the degree of extended reasoning will influence the model performance in the ORACLE benchmark. Three models, o4-mini, o3-mini, claude-4-sonnet_thinking, are picked in this ablation experiment. Two

Table 3: Increase of accuracy with extended thinking in ORACLE 20@2&2@1.

| | Setting | claude-3.7-sonnet | claude-4-sonnet | gemini-2.5-flash |
|---|---|---|---|---|
| CII | Easy | 10.0% | 13.4% | 4.8% |
| | Hard | 8.7% | 18.7% | 15.1% |
| CRI | Easy | 12.0% | 14.0% | 10.0% |
| | Hard | 8.0% | 19.0% | 20.0% |
| PSI | Easy | 21.8% | 26.9% | 12.0% |
| | Hard | 25.0% | 22.9% | 9.0% |
| ERI | Easy | 6.8% | 19.3% | 0.0% |
| | Hard | 0.0% | 1.4% | -6.2% |
| IPI | Easy | 23.3% | 23.3% | 60.0% |
| | Hard | 22.2% | 25.0% | 30.6% |
| GSI | Easy | 7.9% | 18.7% | 21.3% |
| | Hard | 14.6% | 16.0% | 7.4% |

different settings are applied: For o-series models, we evaluate model performance with low and medium reasoning effort. For claude-4-sonnet_thinking, we evaluate model performance with 2000 and 20000 thinking tokens. Results are shown in Table 4, which evidence that more extended thinking can significantly lead to accuracy increase in most scenarios.

## E.3 THE INFLUENCE OF PRIOR KNOWLEDGE

To verify the claim that black-box interaction evaluates pure reasoning ability instead of memorization of knowledge, we perform an ablation study of the influence of prior knowledge. Relevant prior knowledge is explicitly added in the input instruction, and we observer whether the performance is improved or not. Two representative tasks, CRI 10@1 and ERI 10@1, are selected and the results are shown in Table 5. As shown in the results, adding background knowledge explicitly will not lead to better performance for both reasoning and non-reasoning models. The key factor for achieving a high score is the model's strong reasoning ability.

Table 4: Increase of accuracy with the degree of extended thinking from low to medium in ORACLE 20@2&2@1.

| | Setting | o4-mini | o3-mini | claude-4-sonnet* |
|---|---|---|---|---|
| CII | Easy | 25.2% | 15.2% | 8.7% |
| | Hard | 25.0% | 15.2% | 26.5% |
| CRI | Easy | 20.3% | 43.7% | 14.0% |
| | Hard | 20.2% | 21.0% | 27.0% |
| PSI | Easy | 14.1% | 6.4% | 15.4% |
| | Hard | 12.5% | 10.4% | 2.1% |
| ERI | Easy | 36.4% | 38.6% | 17.0% |
| | Hard | 16.7% | 11.1% | -1.3% |
| IPI | Easy | 16.7% | 30.0% | 6.7% |
| | Hard | 25.0% | 13.9% | 2.8% |
| GSI | Easy | 14.3% | 11.1% | -9.6% |
| | Hard | 10.8% | 12.5% | 4.6% |

### E.4 THE INFLUENCE OF EXPLORATION TURNS

We analyze the performance of LLMs under fewer and more exploration turns. Due to the high cost of calling LLMs, we explored the results of four models under the 5@1 and 30@2 settings. The results are shown in Table 6. We can draw several conclusions from the additional experiment: (1). Generally speaking, more turns will not lead to significant accuracy increase for LLMs in most situations. (2). Accuracy can be affected by random fluctuation. (3). Surprisingly, in some cases (e.g., gemini-2.5-pro in ERI Easy), model performance surges when interaction turns are extended to 30. It's similar to an "aha moment". We leave it to future work to analyze. But compared to human performance, it's still inefficient.

Table 5: Model performance with and without prior knowledge in CRI 10@1 and ERI 10@1.

| | Setting | gemini-2.5-pro | gemini-2.5-flash |
|---|---|---|---|
| CRI | Easy, w/ knowledge | 50.0% | 22.0% |
| | Easy, w/o knowledge | 62.0% | 20.0% |
| | Hard, w/ knowledge | 30.0% | 11.0% |
| | Hard, w/o knowledge | 37.0% | 10.0% |
| ERI | Easy, w/ knowledge | 38.6% | 4.6% |
| | Easy, w/o knowledge | 30.7% | 3.4% |
| | Hard, w/ knowledge | 5.6% | 0.0% |
| | Hard, w/o knowledge | 8.3% | 0.0% |

### E.5 THE INFLUENCE OF REASONING TOKENS

We analyze if better models achieve success through superior strategy or simply more computation. We add an experiment that shows average reasoning tokens per turn. The experiments encompassed three representative tasks: CRI, ERI, and PSI. These were evaluated under two distinct settings: 10@1 and 30@2, and the results are presented in Table 7. Two conclusions can be drawn from the results. First, Strong models can achieve success through superior strategy. But developing this strategy will cost lots of reasoning tokens. Second, Weak models cannot develop a good strategy, even with lots of reasoning tokens. This elaborates the fundamental gap of reasoning ability between strong and weak models.

Table 6: Model performance and average reasoning tokens in CRI 10@1, ERI 10@1 and CRI 30@2, ERI 30@2.

| | Setting | o3 | gemini-2.5-pro | gemini-2.5-flash* | deepseek-v3 |
|---|---|---|---|---|---|
| CII | Easy 5@1 | 66.15% | 60.82% | 38.18% | 48.18% |
| | Hard 5@1 | 52.22% | 28.02% | 19.60% | 16.34% |
| | Easy 30@2 | 92.94% | 88.96% | 66.88% | 69.91% |
| | Hard 30@2 | 74.21% | 55.48% | 41.59% | 31.35% |
| CRI | Easy 5@1 | 24.00% | 30.00% | 14.00% | 26.00% |
| | Hard 5@1 | 14.00% | 17.00% | 4.00% | 10.00% |
| | Easy 30@2 | 72.00% | 88.00% | 44.00% | 46.00% |
| | Hard 30@2 | 44.00% | 58.00% | 37.00% | 31.00% |
| PSI | Easy 5@1 | 38.46% | 58.97% | 28.21% | 19.23% |
| | Hard 5@1 | 20.83% | 20.83% | 16.67% | 4.17% |
| | Easy 30@2 | 67.95% | 70.51% | 55.13% | 17.95% |
| | Hard 30@2 | 47.91% | 50.00% | 45.83% | 12.50% |
| ERI | Easy 5@1 | 52.27% | 23.86% | 6.82% | 2.27% |
| | Hard 5@1 | 11.11% | 11.11% | 1.39% | 0% |
| | Easy 30@2 | 68.18% | 55.68% | 6.82% | 6.82% |
| | Hard 30@2 | 19.44% | 2.78% | 0% | 0% |
| IPI | Easy 5@1 | 46.67% | 53.44% | 30.00% | 26.67% |
| | Hard 5@1 | 33.33% | 19.44% | 13.89% | 2.78% |
| | Easy 30@2 | 100.00% | 100.00% | 90.00% | 50.00% |
| | Hard 30@2 | 80.56% | 63.89% | 38.89% | 0% |

### E.6 THE INFLUENCE OF REFLECTIVE INSTRUCTION

We study whether explicitly instructing LLMs to reflect prior feedback and keep an efficient exploration strategy at the end of each turn will improve their performance or not. The specific instruction prompt is "Keep an efficient exploration strategy by reflecting on prior feedback". We pick gemini-2.5-pro, gemini-2.5-flash_thinking, o3 in CRI and ERI task under 10@1. The results are shown in Table 8. They indicate that the LLM's lack of an efficient and adaptive exploration strategy is a fundamental flaw, which cannot be compensated for by Chain-of-Thought prompting.

Table 7: Model performance and average reasoning tokens in CRI 10@1, ERI 10@1 and CRI 30@2, ERI 30@2.

| | Setting | gemini-2.5-pro | gemini-2.5-flash* | o4-mini |
|---|---|---|---|---|
| CRI | Easy 10@1 | 62.0% / 11700.9 | 24.0% / 9010.3 | 48.0% / 6619.4 |
| | Hard 10@1 | 37.0% / 11844.7 | 17.0% / 12072.3 | 22.0% / 6220.8 |
| | Easy 30@2 | 88.0% / 7619.4 | 44.0% / 7423.8 | 72.0% / 7654.3 |
| | Hard 30@2 | 58.0% / 10206.8 | 37.0% / 9839.2 | 44.0% / 8747.6 |
| ERI | Easy 10@1 | 30.7% / 6464.8 | 6.8% / 1856.1 | 56.8% / 7203.6 |
| | Hard 10@1 | 8.3% / 8208.4 | 0.0% / 1873.7 | 12.5% / 7878.6 |
| | Easy 30@2 | 55.7% / 3002.2 | 6.8% / 728.9 | 68.2% / 7776.4 |
| | Hard 30@2 | 2.8% / 5513.5 | 0% / 475.5 | 19.4% / 7776.4 |
| PSI | Easy 10@1 | 57.7% / 6460.7 | 38.5% / 5025.6 | 59.0% / 2320.0 |
| | Hard 10@1 | 38.6% / 7411.9 | 37.5% / 5316.0 | 27.1% / 3354.6 |
| | Easy 30@2 | 70.5% / 5160.2 | 55.1% / 2531.3 | 68.0% / 3502.7 |
| | Hard 30@2 | 50.0% / 6256.8 | 45.8% / 3055.6 | 47.9% / 5360.3 |

### E.7 HUMAN EVALUATION

In this part, we show human performance in the ORACLE benchmark. It is worth noting that many black-box tasks are both time-consuming and challenging for humans. Due to cost constraints, for certain tasks, we merely randomly selected a few black boxes for testing (e.g., for easy black-boxes in CII we pick 3 black-boxes, abbreviated as CII Easy (3)). The human participants involved in the testing were undergraduate and graduate students from the departments of Mathematics, Physics, and Computer Science at several top-tier universities. The results are in Table 9. In the process of human evaluation, we summarized the following findings: (1). Except for the Code Intent Inference task, humans achieved performance significantly surpassing the best-performing models on the other five categories of tasks. (2). During the black-box interaction process, humans demonstrated the ability to formulate highly effective adaptive and efficient exploration strategies, which enabled them to solve some black-box problems in around five turns.

Table 8: Model performance with and without feedback reflection in CRI 10@1 and ERI 10@1.

| | Setting | gemini-2.5-pro | gemini-2.5-flash* | o3 |
|---|---|---|---|---|
| CRI | Easy, w/ Reflection | 66.0% | 18.0% | 58.0% |
| | Easy, w/o Reflection | 62.0% | 24.0% | 64.0% |
| | Hard, w/ Reflection | 34.0% | 12.0% | 21.0% |
| | Hard, w/o Reflection | 37.0% | 17.0% | 24.0% |
| ERI | Easy, w/ Reflection | 25.0% | 9.1% | 33.0% |
| | Easy, w/o Reflection | 30.7% | 6.8% | 26.1% |
| | Hard, w/ Reflection | 13.9% | 0.0% | 2.8% |
| | Hard, w/o Reflection | 8.3% | 0.0% | 15.3% |

## F PROMPT DETAILS

### F.1 PROMPT FOR BLACK-BOX GENERATION

We report prompts used in the black-box generation framework in this part. Specifically, there are two types of prompts, prompt for coding LLM which generates black-box code (denoted as Description of Black-box in Figure 4), prompt for refinement LLM which helps check and correct black-box code (denoted as Task Rule in Figure 4). Recall the framework for black-box generation in Section 4, there are two types of errors during simulation, code that is non-executable and code that incorrectly implements black-box. For

Table 9: Comparison of human performance and SOTA LLM performance in the ORACLE benchmark 10@1 and 20@2

| | Setting | Human 10@1 | SOTA LLM 10@1 | Human 20@2 | SOTA LLM 20@2 |
|---|---|---|---|---|---|
| CII | Easy (3) | 96% | 78% | 100% | 98% |
| | Hard (2) | 3% | 53% | 21% | 76% |
| CRI | Easy (3) | 67% | 70% | 100% | 93% |
| | Hard (3) | 67% | 47% | 67% | 83% |
| PSI | Easy (all) | 100% | 62% | 100% | 65% |
| | Hard (5) | 100% | 27% | 100% | 13% |
| ERI | Easy (all) | 100% | 64% | 100% | 61% |
| | Hard (3) | 67% | 0% | 67% | 13% |
| IPI | Easy (all) | 100% | 83% | 100% | 98% |
| | Hard (all) | 100% | 36% | 100% | 67% |
| GSI | Easy (all) | 88% | 79% | 98% | 83% |
| | Hard (4) | 65% | 35% | 100% | 52% |

non-executable code, we directly instruct Refinement LLM to modify current code with error messages. As to executable code, we first instruct Refinement LLM to judge whether the code correctly implements black-box according to simulated interaction log. If there exist errors and inconsistencies, Refinement LLM is then instructed to modify code based on concluded mistake and original request. Corresponding prompts are shown below.

---

**Prompt for Refinement LLM when code fails to be executed**
You need to fix running errors of a code.
- The code is as follows:
{current_code}
- This above code has the following running errors:
{running_errors}
- You need to think step by step and output the revised correct code. **Make sure the output code is runnable, DO NOT output any other text**.

---

**Prompt for Refinement LLM to detect whether current code correctly implements black-box**
- Here is an interaction log:
{interaction_log}
- The interaction log is supposed to follow the given rules:
{taskintro}

---

and the black-box function implements {algorithm}, {description}.
- You need to read the log carefully and identify whether the log matches the given rule and correctly implements the black-box function. Neglect the wrong answer from the assistant because it's not a part of the judgment. Neglect the replies from the assistant and **focus on the replies from the user**. Output "correct" if you think it's correct, **DO NOT output any other text**. If you think the interaction logs do not satisfy the given request, output the wrong points clearly but do not contain revised code.

**Prompt for Refinement LLM to modify current code**
- The code is as follows:
{current_code}
- This above code is supposed to implement this function:
{request}
- However, the current code does not meet the above requirements and has the following mistakes:
{mistake}
- Your task is to revise current code according to known mistakes. Make sure the output code is runnable, **DO NOT output any other text**.

### F.1.1 CODE INTENT INFERENCE (CII)

**Prompt for Coding LLM**
- Task overview:
You should formalize a player-blackbox interaction process into runnable python code. Generally speaking, you need to focus on three things:
    - Generate the function of a blackbox, which implements a specific algorithm.
    - Generate the function of a platform, which parses player output and get blackbox output.
    - Generate the main function which is responsible for the interaction between player and blackbox.
- Detailed Coding Instructions:
    - Before starting, we list all the related packages. You need to import them at the beginning of programme:
        - 'import os'
        - 'import sys'
        - current_path = os.path.abspath(__file__)
        - 'oracle_path = os.path.dirname(os.path.dirname(os.path.dirname(os.path.dirname(current_path))))'
        - 'if oracle_path not in sys.path:'
        - ' sys.path.insert(0, oracle_path)'
        - 'from ckpt import get_local_variables, check_query_validity, get_ckpt_numbers, get_function_params, capture_print'
        - 'from eva_models import ReasoningLLM'
        - 'import re'
    - Generate the code of a blackbox:
        - First insert 'get_local_variables.counters = ' and 'get_local_variables.max_visits = ' inside the function to initialize.
        - Write a function named 'blackbox'. It implements algorithm. The description of this algorithm is that description. The input variables of this function should be the simplest (e.g. you can get the length of a list through len() function, so you don't need an additional variable n as imput). **Note that you must write Type Hints for all the input variables!**
        - Apart from the original function input variables, 2 additional parameters 'idx=0, iter=0' must be added to the function input variables.
        - There's a python function 'get_local_variables(idx)' that can get the result of all local variables. You need to assert it to the 'blackbox' function in the places where local variables are changed in the runtime. But do not insert too many checkpoints. 'idx' refers to the times 'get_local_variables' is inserted. For example, when first inserted, add 'get_local_variables(1)', when second inserted, add 'get_local_variables(2)', and so forth.
        - At the end of this function, add 'check_query_validity(idx, iter)'.
        - **DO NOT return any values.**
        - **Use meaningless variable names like a, t, s, num, arr, etc. Make sure the names of variables do not leak any potential information.**
    - Generate the code of a platform:

- Write a function named 'platform', which takes 'player_output' as function input variable and print 'blackbox_output'. Use '@capture_print' as Decorator for this function. **Note the function Only use 'print', DO NOT use 'return'!**
- The code should first parse 'player_output' and then get the result of corresponding 'black-box_output'. To correctly parse 'player_output', you need to pay special attention to its format. There are **only 3 conditions** for 'player_output':
  - Condition 1: If the 'player_output = ''', call 'params = get_function_params(blackbox)' and 'num_ckpt = get_ckpt_numbers(blackbox, get_local_variables)' orderly to get the names and types of all variables, and the total number of checkpoints. Then 'print(f'The black-box takes params as input variables, and has num_ckpt checkpoints.')'.
  - Condition 2: If the 'player_output' follows this format: "variable_name_1 = value_1; variable_name_2 = value_2; ..." (e.g. arr = [1, 5, 2]; n = 3), First split 'player_output', then change the type of value from string to its real type for each variable_name. Call 'params_info = get_function_params(blackbox)' to get the names and types of all variables. 'params_info' is a List like this '['name': 'arr', 'type': <class 'list'>, 'name': 'n', 'type': <class 'int'>]'. Check if variable_name is in 'params_info'. If not, set 'print('Error: The variable name is not in the function parameters')'. If yes, change the type of value from string to its real type, and record 'variable_name_1: value_1, variable_name_2: value_2' in a global dict 'vars'. When new variable value is assigned, dict 'vars' needs to be updated. Set 'print(f'Set vars.')'
  - Condition 3: If the 'player_output' follows this format: "('idx', 'iter')", set 'blackbox(**vars, idx, iter)'.
  - Remember to tackle with unexpected errors:
  - If 'player_output' is not in the above-mentioned 3 conditions, the function should print output format rules.
  - Some other possible errors. Remember to use 'print' instead of 'return'.
- Generate the main code:
  - The main function takes some input variables 'main(model_family, model_name, task, eva_mode, n_runs, difficulty, task_id, failure_num, output_dir, max_turns, version, mode, thinking_mode)'.
  - First, the main code need to instantiate 'ReasoningLLM' class through 'player = ReasoningLLM(model_family, model_name, task, eva_mode, n_runs, difficulty, task_id, thinking_mode, mode)'.
  - Then, call 'blackbox_output = platform(player_output, max_turns)' and 'player_output = player.normal_output(blackbox_output)' iteratively in a 'for' loop with 'max_turns' iterations. When 'blackbox_output = platform(player_output)' is first called, set it as 'player_output = '''. Add string 'f'<Current Turn: i+1, max_turns-(i+1) Turns Remaining>'' before each 'blackbox_output', will 'i' is the index in the loop.
  - When the loop exits, call 'player.evaluate(failure_num, version)' and 'player.save_history(output_dir, version)'.
  - Finish the main function.
- Add 'if __name__ == "__main__":', 'args = sys.argv[1:]', 'main(args[0], args[1], args[2], args[3], int(args[4]), args[5], args[6], int(args[7]), args[8], int(args[9]), int(args[10]), args[11], bool(eval(args[12])))' at the end of this programme to make it runnable.
- Tips: DO NOT include other text output. Make sure the output is runnable. Code and think step by step.

### F.1.2 CIRCUIT RULE INFERENCE (CRI)

**Prompt for Coding LLM**
# Boolean Circuit Family Implementation Task
## Definitions and Background
In our task: - A **boolean circuit** is an acyclic computational structure with fixed 0/1 bit inputs, composed of AND/OR/NOT logic gates. Each gate receives inputs either from input wires or from outputs of other gates. - The task is to write the code of the game platform. The player will use the platform to gain information with the circuit, and try to guess the function of the circuit.
## Task Overview
Your objective is to formalize a player-blackbox interaction process into runnable Python code. Focus on implementing three key components:
1. **Circuit Function**: Implement a function that constructs the circuit according to the language description, given the input wires and returns the output of each gate.
2. **Main Function**: Implement the core interaction loop between the LLM player and the blackbox.
## Detailed Coding Instructions

### Required Imports Begin by importing these packages: '''python import os import sys current_path = os.path.abspath(__file__) oracle_path = os.path.dirname(os.path.dirname(os.path.dirname(os.path.dirname(current_path)))) if oracle_path not in sys.path: sys.path.insert(0, oracle_path) from ckpt import simulate_circuit from eva_models import ReasoningLLM import re '''

### Circuit Generator Function Implement 'blackbox(circuit_input)' where: - 'circuit_input' are the 0/1 bits of the input wires - Returns 'gate_output' describing the 0/1 bits of each gate output - The circuit's purpose is algorithm - Construction details: description - In 'blackbox', you should construct 'gates', and then use 'simulate_circuit' to generate the response (see Interface Details section) - **Important**: Verify that your gate construction achieves the stated goal and follows the description exactly

### Main Function Implement 'main(model_family, model_name, task, eva_mode, n_runs, difficulty, task_id, failure_num, output_dir, max_turns, version, mode, thinking_mode)' where: 1. Instantiate the 'ReasoningLLM' class: '''python player = ReasoningLLM(model_family, model_name, task, eva_mode, n_runs, difficulty, task_id, thinking_mode, mode) '''

2. Create an interaction loop for 'max_turns' iterations: - Call 'player_output = player.normal_output(blackbox_output)' to send the message 'blackbox_output' to the player, and get the player's output. - 'blackbox_output' should append the remain rounds for interaction. - For the first iteration, 'blackbox_output' should be a prompt telling player the game begin. - For every 'player_output', try to parse it into a 0/1 list as 'circuit_input' - If the parsing succeeds, 'blackbox_output = platform(circuit_input)' in the next iteration, otherwise 'blackbox_output' should remind the player follows the format.

3. After the loop completes: - 'player_output = player.normal_output(blackbox_output)' to give the last answer of the player. - Call 'player.evaluate(failure_num, version)' - Call 'player.save_history(output_dir, version)'

4. Add an entry point at the end of the program: '''python if __name__ == "__main__": args = sys.argv[1:] main(args[0], args[1], args[2], args[3], int(args[4]), args[5], args[6], int(args[7]), args[8], int(args[9]), int(args[10]), args[11], bool(eval(args[12]))) '''

## Circuit Details
- 'AND' and 'OR' gates always require exactly two inputs - 'NOT' gates always require exactly one input - A gate's output or an input wire can be used as input to multiple other gates without restriction - Circuits must be acyclic: for the i-th gate, all inputs must come from either input wires or outputs of gates with indices smaller than i - The index of the gate is from 1 to $m$, where $m$ is the number of gates.

## Interface Details
Use 'simulate_circuit(n, m, input, gates)' as follows:
'''python ''' Returns a list of the gates' outputs, or a string containing an error message Parameters: n: number of input wires m: number of gates input: list of input values (0 or 1) gates: list of tuples, each tuple contains gate type and its inputs - gate type: 'AND', 'OR', 'NOT' - inputs: (0, i) means the input wire i (1, j) means the output of gate j Note that every index begins from 1 (i, j>=1)
Examples: - ('AND', (0, 1), (0, 2)) means AND gate taking input wires 1 and 2 - ('NOT', (1, 3)) means NOT gate taking the output of gate 3
Example usage:
gates = [ ('AND', (0, 1), (0, 2)), ('AND', (0, 3), (1, 1)), ('OR', (0, 4), (1, 2)), ('NOT', (1, 3)), ('NOT', (0, 5)) ]
input = [1, 1, 0, 1, 0]
simulate_circuit(5, 5, input, gates) # Returns [1, 0, 1, 0, 1] ''' def simulate_circuit(n, m, input_wires, gates) -> str | list: # Implementation details omitted '''

## Development Tips - Focus only on producing runnable code without additional text output - Verify your implementation step-by-step - Ensure the circuit generator correctly implements the specified algorithm - Validate that the platform function properly handles all input formats

### F.1.3 PHYSICS SYSTEM INFERENCE (PSI)

**Prompt for Coding LLM**
- Task overview: You should formalize a player-blackbox interaction process into runnable python code. Generally speaking, you need to focus on two things: - Generate the function of a blackbox, which implements an encryption algorithm. - Generate the main function which is responsible for the interaction between player and blackbox.
- Detailed Coding Instructions: - Before starting, we list all the related packages. You need to import them at the beginning of programme: - 'import os' - 'import sys' - 'import math' - 'current_path = os.path.abspath(__file__)' - 'oracle_path = os.path.dirname(os.path.dirname(os.path.dirname(os.path.dirname(current_path))))' - 'if oracle_path

not in sys.path:' - ' sys.path.insert(0, oracle_path)' - 'from eva_models import ReasoningLLM' - 'from scipy.integrate import solve_ivp' - 'import numpy as np'

- Generate the code of a blackbox: - Write a function named 'blackbox'. It implements algorithm. The detailed description is that description. - You need to first build a 3-dimensional coordinate for this mechanical system. However you only consider algorithm as 2-dimensional, but return 3-dimensional coordinate. - You need to figure out how the objects' location changes over time, namely calculating x(t), y(t), z(t) based on physics knowledge. If analytical solution does not exist, use 'solve_ivp' to calculate numerical solution. - The function tasks only one variable as input, which is 't: float', and return only one variable 'object_coordinate', which is a dict that contains 3-dimensional coordinate of all the objects in the black-box. The objects are named 'object1', 'object2', etc., and 'object_coordinate' is formatted in '"object1": (x, y, z), "object2": (x, y, z), ...'. All coordinates are approximated to two decimal places.

- Generate the main code: - The main function takes some input variables 'main(model_family, model_name, task, eva_mode, n_runs, difficulty, task_id, failure_num, output_dir, max_turns, version, mode, thinking_mode)'. - First, the main code need to instantiate 'ReasoningLLM' class through 'player = ReasoningLLM(model_family, model_name, task, eva_mode, n_runs, difficulty, task_id, thinking_mode, mode)'. - Then, call 'player_output = player.normal_output(blackbox_output)' and 'blackbox_output = black(player_output)' iteratively in a 'for' loop with 'max_turns+1' iterations. When 'player_output = player.normal_output(blackbox_output)' is first called, set 'blackbox_output' as 'blackbox_output = f'You have max_turns interaction turns to understand the black-box. Now the interaction starts. Only output the value and DO NOT contain any unrelated text.'' first. Besides the situation that 'blackbox_output' is first called, add string 'f'<Current Turn: i+1, max_turns-(i+1) Turns Remaining>'' before each 'blackbox_output'. 'i' is the index in the loop. In the last iteration of the loop, after 'player_output = player.normal_output(blackbox_output)' is called, add 'continue' subsequently to exit.

- When the loop exits, call 'player.evaluate(failure_num, version)' and 'player.save_history(output_dir, version)'. - Finish the main function.

- Add 'if __name__ == "__main__":', 'args = sys.argv[1:]', 'main(args[0], args[1], args[2], args[3], int(args[4]), args[5], args[6], int(args[7]), args[8], int(args[9]), int(args[10]), args[11], bool(eval(args[12])))' at the end of this programme to make it runnable.

- Tips: DO NOT include other text output. Make sure the output is runnable. Code and think step by step.

### F.1.4 ENCRYPTION RULE INFERENCE (ERI)

**Prompt for Coding LLM**

- Task overview: You should formalize a player-blackbox interaction process into runnable python code. Generally speaking, you need to focus on two things: - Generate the function of a blackbox, which implements an encryption algorithm. - Generate the main function which is responsible for the interaction between player and blackbox.

- Detailed Coding Instructions: - Before starting, we list all the related packages. You need to import them at the beginning of programme: - 'import os' - 'import sys' - 'current_path = os.path.abspath(__file__)' - 'oracle_path = os.path.dirname(os.path.dirname(os.path.dirname(os.path.dirname(current_path))))' - 'if oracle_path not in sys.path:' - ' sys.path.insert(0, oracle_path)' - 'from eva_models import ReasoningLLM'

- Generate the code of a blackbox: - Write a function named 'blackbox'. It implements algorithm. The description of this algorithm is that description. - The function tasks only one variable as input, which is 'plaintext', and return only one variable 'ciphertext'. Both 'plaintext' and 'ciphertext' only contains English letters.

- Generate the main code: - The main function takes some input variables 'main(model_family, model_name, task, eva_mode, n_runs, difficulty, task_id, failure_num, output_dir, max_turns, version, mode, thinking_mode)'. - First, the main code need to instantiate 'ReasoningLLM' class through 'player = ReasoningLLM(model_family, model_name, task, eva_mode, n_runs, difficulty, task_id, thinking_mode, mode)'. - 'blackbox_output' is first as 'blackbox_output = f'You have max_turns interaction turns to understand the black-box. Now the interaction starts. Only output the value and DO NOT contain any unrelated text.''. - Then, call 'player_output = player.normal_output(blackbox_output)', 'blackbox_output = f'<Current Turn: i+1, max_turns-(i+1) Turns Remaining> ' + blackbox(player_output)', iteratively in a 'for' loop with 'max_turns+1' iterations. 'i' is the index in the loop. - When the loop exits, call 'player.evaluate(failure_num, version)' and 'player.save_history(output_dir, version)'. - Finish the main function.

- Add 'if __name__ == "__main__":', 'args = sys.argv[1:]', 'main(args[0], args[1], args[2], args[3], int(args[4]), args[5], args[6], int(args[7]), args[8], int(args[9]), int(args[10]), args[11], bool(eval(args[12])))' at the end of this programme to make it runnable.

- Tips: DO NOT include other text output. Make sure the output is runnable. Code and think step by step.

### F.1.5 INTERACTIVE PUZZLE INFERENCE (IPI)

**Prompt for Coding LLM**
- Task overview: You should formalize a player-blackbox interaction process into runnable python code. Generally speaking, you need to focus on 3 things: - Generate the function of a blackbox, which implements a interactive puzzle. You need to check the validity of the player's queries and give feedback according to the truth. - Generate a boolean format checking function, which checks the answers format when the player tries to give answers. - Generate the main function which calls some essential functions.
- Detailed Coding Instructions: - Before starting, we list all the related packages. You need to import them at the beginning of programme: - 'import os' - 'import sys' - 'current_path = os.path.abspath(__file__)' - 'oracle_path = os.path.dirname(os.path.dirname(os.path.dirname(os.path.dirname(current_path))))' - 'if oracle_path not in sys.path:' - ' sys.path.insert(0, oracle_path)' - 'from eva_models import ReasoningLLM'
- Generate the code of a blackbox: - Write a function named 'blackbox'. It implements algorithm. The description of this algorithm is that description. - Note That 'blackbox' just answers every query and does not include other functions. - The function takes 2 variables as input, which is 'truth' and 'query', and return some feedback according to above description. Note that 'truth' is a string that represents the correct answer (i.e. the player will answer it finally) and 'query' is a string that represents the player's query. - If 'query' doesn't follow correct format, please give some correcting advice as a 'str' using 'return'.
- Generate the boolean format checking function - Write a function named 'check_answer_format'. It is used to check whether answer format matches 'truth' according to above description. - The function takes only 1 variable as input, which is 'answer', and return True / False. - Note that answer will be given directly as the description above. Do not assume 'answer' including any other text.
- Generate the main code: - The main function takes some input variables 'main(model_family, model_name, task, eva_mode, n_runs, difficulty, task_id, failure_num, output_dir, max_turns, version, mode, thinking_mode)'. - First, the main code need to instantiate 'ReasoningLLM' class through 'player = ReasoningLLM(model_family, model_name, task, eva_mode, n_runs, difficulty, task_id, thinking_mode, mode)'. - Then, call 'player.evaluate(failure_num, version, max_turns)' and 'player.save_history(output_dir, version)' to evaluate and save logs. - Finish the main function. - Only contain above code in the main function, do not include any other code.
- Add 'if __name__ == "__main__":', 'args = sys.argv[1:]', 'main(args[0], args[1], args[2], args[3], int(args[4]), args[5], args[6], int(args[7]), args[8], int(args[9]), int(args[10]), args[11], bool(eval(args[12])))' at the end of this programme to make it runnable.
- Tips: DO NOT include other text output. Make sure the output is runnable. Code and think step by step.

### F.1.6 GAME STRATEGY INFERENCE (GSI)

**Prompt for Coding LLM**
- Task overview: You should formalize a player-blackbox interaction process into runnable python code. Generally speaking, you need to focus on 3 things: - Generate the function of a blackbox, which implements a interactive puzzle. You need to check the validity of the player's queries and give feedback according to the truth. - Generate a boolean format checking function, which checks the answers format when the player tries to give answers. - Generate the main function which calls some essential functions.
- Detailed Coding Instructions: - Before starting, we list all the related packages. You need to import them at the beginning of programme: - 'import os' - 'import sys' - 'current_path = os.path.abspath(__file__)' - 'oracle_path = os.path.dirname(os.path.dirname(os.path.dirname(os.path.dirname(current_path))))' - 'if oracle_path not in sys.path:' - ' sys.path.insert(0, oracle_path)' - 'from eva_models import ReasoningLLM'
- Generate the code of a blackbox: - Write a function named 'blackbox'. It implements {algorithm}. The description of this algorithm is that {description}. - Note That 'blackbox' just answers every query and does not include other functions. - The function takes 2 variables as input, which is 'truth' and 'query', and return some feedback according to above description. Note that 'truth' is a string that represents the correct answer (i.e. the player will answer it finally) and 'query' is a string that represents the player's

query. - If 'query' doesn't follow correct format, please give some correcting advice as a 'str' using 'return'.
- Generate the boolean format checking function - Write a function named 'check_answer_format'. It is used to check whether answer format matches 'truth' according to above description. - The function takes only 1 variable as input, which is 'answer', and return True / False. - Note that answer will be given directly as the description above. Do not assume 'answer' including any other text.
- Generate the main code: - The main function takes some input variables 'main(model_family, model_name, task, eva_mode, n_runs, difficulty, task_id, failure_num, output_dir, max_turns, version, mode, thinking_mode)'. - First, the main code need to instantiate 'ReasoningLLM' class through 'player = ReasoningLLM(model_family, model_name, task, eva_mode, n_runs, difficulty, task_id, thinking_mode, mode)'. - Then, call 'player.evaluate(failure_num, version, max_turns)' and 'player.save_history(output_dir, version)' to evaluate and save logs. - Finish the main function. - Only contain above code in the main function, do not include any other code.
- Add 'if __name__ == "__main__":', 'args = sys.argv[1:]', 'main(args[0], args[1], args[2], args[3], int(args[4]), args[5], args[6], int(args[7]), args[8], int(args[9]), int(args[10]), args[11], bool(eval(args[12])))' at the end of this programme to make it runnable.
- Tips: DO NOT include other text output. Make sure the output is runnable. Code and think step by step.

## F.2 PROMPT FOR BLACK-BOX INTERACTION

We report initial instructions (i.e., task introduction) for benchmarked LLMs in different black-box tasks. Prompts are written in markdown style.

### F.2.1 CODE INTENT INFERENCE (CII)

1. Task overview: - The user plays the role of a code function, but you don't know what the function is. You need to understand the detailed working principle of this function by interacting with user in multiple turns.
2. Goals: - By interacting with the user within given interaction turns, you need to understand how the code function operates in every checkpoint.
3. User property: - The code function takes some variables as input. - The code function has some checkpoints. You can specify '(idx, iter)' to get the value of accessible intermediate variables: - 'idx': int is the index of checkpoint (i.e. the idx-th checkpoint). - 'iter': int is the number of visited times (i.e. checkpoint will be visited multiple times in loop). - Example: '(2, 3)' means getting the value of accessible variables when the 2-nd checkpoint is visited for the 3-rd time.
4. Interaction rules: - Rule 0: The user will first tell you the *type* and *name* of all the function input variables, the *total number of checkpoints*. In each turn, the user will tell *current turn* and *remaining turns*. - Rule 1: You can assign or re-assign any values to the input variables freely, but they must match the variable types (e.g. if num is int, you can't assign a float number to it). DO NOT assign 'idx' and 'iter' in this stage. - Rule 2: You can ask for the value of accessible variables at 'idx' checkpoint in 'iter' iteration. The user will return the variable names and values. - The interaction order matters: You are free to implement Interaction Rule 1, 2 at any time, **but you must make sure Rule 1 is implemented before Rule 2.**
5. Output format: - **You must strictly obey the output format rules, DO NOT output any unrelated text!**: - According to Interaction Rule 1, when you want to assign values to input variables, output "variable_name_1 = value_1; variable_name_2 = value_2; ..." (e.g. arr = [1, 5, 2, 9]; k = 3) - According to Interaction Rule 2, when you want to ask for accessible variables in 'idx' checkpoint in 'iter' iteration, output "('idx', 'iter')" (e.g. (1, 2)). **Note the minimal number for 'idx' and 'iter' is 1.**
6. Evaluation: - When the given number of interactions is reached, several questions on the variable value at certain checkpoint will be presented. **You MUST ONLY output the value, DO NOT contain any other text.**

### F.2.2 CIRCUIT RULE INFERENCE (CRI)

0. Definitions: A 'circuit' means a boolean circuit with no circle, whose input wires are a fixed number of 0/1 bits, and consisted of 'AND'/'OR'/'NOT' gates. For each gates, the input are input wires or the output of other gates.
1. Task overview: You are presented with a platform containing a circuit. Your task is to discover the structure of this circuit by interacting with the platform over multiple turns.
2. Goals: You have one final goal: You need to know output of circuit gates given every input.
3. Interaction Process: To finish the goal, you need to interact with the platform. The interaction process is as follows: - The platform will first provide you with the input sizes $n$ - In each turn, you can ask the platform for the outputs of every circuit gates with a given input size $x = (x_1, x_2, ..., x_n), x_i \in \{0, 1\}$. - Each turn after you ask the platform, you will receive the circuit's gate output in the following format: $[y_1, y_2, ..., y_m], y_i \in \{0, 1\}$, where $m$ is the number of gates, $y_i$ is the output of the $i$-th gate.
4. Output format in Interaction: **You must strictly obey the output format rules, DO NOT output any unrelated text!**: Format your queries as: (x_1, x_2, ... , x_n). For example, if you what to now the output of circuit given input $x = (0, 1, 1)$, you should output: (0, 1, 1).
5. Evaluation: After reaching the maximum number of allowed interactions, you will be tested on your understanding of the circuit family: - You will be given input $x = (x_1, x_2, ..., x_n)$ and you should answer the output of each gate $[y_1, y_2, ..., y_m]$. - Your construction should specify the exact connections and gate types in $C_n$.
6. Circuit Details - 'AND' and 'OR' gates always have exactly two inputs, and 'NOT' gatea always have one input. - There are no restrictions on how many times a gate's output or an input wire can be used as input to other gates. - Every circuit in the circuit family contains no loops, For the $i$-th gate, all inputs must come from either input wires or outputs of gates with indices smaller than $i$.

### F.2.3 PHYSICS SYSTEM INFERENCE (PSI)

1. Task overview: - The user plays the role of a classical mechanical system, but you don't know what it is. You need to understand how this classical mechanical system operates by interacting with the user in multiple turns.
2. Goal: - By interacting with the user within given interaction turns, you need to understand how the mechanical system operates.
3. User property: - The user will remind you the remaining number of turns in each turn. - The user takes time 't: float' as input. - The user return the 3-dimensional coordinate '(x, y, z)' of each object in time 't'.
4. Interaction rules: - Rule 1: You need to assign a value of 't'. You can only assign one 't' in each turn. Make sure the assigned 't' is a one-digit decimal. - Rule 2: You will receive the user response, which is 3-dimensional coordinate '(x, y, z)' of each object in time 't', after you assign specific 't'.
5. Output format: - You must strictly obey the output format rules: When you want to assign value for 't', **only output the value**. DO NOT output any unrelated text or symbols like "Let's input", "I'll try". - If you understand the mechanical system before reaching given interaction turns, keep interacting with the user to make sure you don't miss any details. DO NOT output text like "I understand the pattern".
6. Evaluation: - When the given number of interactions is reached, you need to calculate the 3-dimensional coordinate '(x, y, z)' of each object at time 't' with the format of '"object1": (x, y, z), "object2": (x, y, z), ...'. All coordinates are approximated to two decimal places.

### F.2.4 ENCRYPTION RULE INFERENCE (ERI)

1. Task overview: - The user transforms one string into another based on a fixed rule, but you don't know what the fixed rule is. You need to figure out this rule by interacting with the user in multiple turns.
2. Goal: - By interacting with the user within given interaction turns, you need to understand the fixed rule of transforming one string into another.
3. User property: - The user will remind you the remaining number of turns in each turn. - The user will output transformed string in each turn.
4. Interaction rules: - Rule 1: You must only assign one string in each turn. You can assign any string freely, but make sure the string **only contains uppercase or lowercase English letters (A-Z, a-z) and blank space**. Then you will receive corresponding transformed string from the user.

5. Output format: - You must strictly obey the output format rules: **Only output the string**. DO NOT output any unrelated text or symbols like "Let's input", "I'll try", "/ n". - If you understand the transforming rule before reaching given interaction turns, keep interacting with the user to make sure you don't miss any details. DO NOT output text like "I understand the pattern".

6. Evaluation: - When the given number of interactions is reached, you will be given several test strings and you need to output corresponding transformed string. **You MUST ONLY output the value, DO NOT contain any other text.**

### F.2.5 INTERACTIVE PUZZLE INFERENCE (IPI)

1. Task overview: - The user plays the role of a puzzle, and you don't know what the hidden answer is. You need to guess the hidden answer by interacting with the user in multiple turns.

2. Goals: - You need to guess the answer to the puzzle within given interaction turns.

3. User property: - The user hides the answer which you need to figure out.

4. Interaction rules: - Rule 0: The user will first tell you the rule of the puzzle, and the interaction format that must be followed when querying. In each turn, the user will tell *current turn* and *remaining turns*. - Rule 1: You can ask questions according to the rules of the game and receive corresponding feedback. If your ask is unavailable, the user will tell you. - Rule 2: After a series of interactions, you should answer the puzzle in the format specified in the description.

5. Output format: - When you ask a question, you should strictly follow query format in the **Description**. - When you answer the puzzle, you should strictly follow the answer format in the **Description**. - Refer to the examples in the **Description**. for the correct format. - If you figure out the right answer before given turns, keep interacting with the puzzle to make sure your answer is correct.

6. Evaluation: - When the given number of interactions is reached, you need to give your answer of the puzzle. **You MUST ONLY output the answer itself in the format mentioned in the description, DO NOT contain more text.**

Now Let's Solve the Puzzle algorithm.

**Description**: description.

### F.2.6 GAME STRATEGY INFERENCE (GSI)

1. Task overview: - The user plays the role of an opponent who takes a fixed strategy in a game. But you don't know what the stategy is. You need to guess the hidden stategy of your opponent by interacting (playing game) with him in multiple turns.

2. Goals: - You have 1 final goal: You need to guess your opponent's strategy and try to maximize your score in the game. The score might depend on winning rate, or minimal cost, etc.

3. User property: - The user hides his game strategy which you need to figure out to win the game.

4. Interaction rules: - To finish the goal, you need to interact with the user. The interaction rules are as follows: - Interaction Rule 0: The user will first tell you the rule of the game, and the interaction format that must be followed when playing. In each turn, the user will tell *current turn* and *remaining turns*. - Interaction Rule 1: You can take actions according to the rules of the game and receive corresponding feedback, such as current game states. If your action is unavailable, the user will tell you. - Interaction Rule 2: We will first play a few times of the game to familiarize you with the rules and the behavior of your opponent. In this phase, your actions will not be recorded, and your score does not matter. You can make use of this phase to explore the game and understand the opponent's strategy. - Interaction Rule 3: After the *exploration phase*, you will enter the *evaluation phase*. We will only play the game for 1 final time, and your actions will be recorded. Your score will be calculated based on your actions in this final game.

5. Output format: - **You must strictly obey the output format rules, DO NOT output any unrelated text!**:

6. Evaluation: - When the given number of interactions is reached, the game ends and we'll calculate your **score**

Now Let's Play the Game {algorithm}, the Description Is that {description}.

## G DETAILS OF BLACK-BOXES IN THE ORACLE BENCHMARK

We introduce the detailed implementation of black-boxes in each task. The definition of easy and hard black-boxes are based on two factors: human understanding of black-box difficulty and the

black-box's inherent difficulty. The designers ranked the black-box difficulty of each task based on a partial order relationship. They ultimately discussed and collectively determined the sets for easy and hard tasks. In the beginning of each subsection, we detail the basis for the difficulty division.

### G.1 CODE INTENT INFERENCE (CII)

In the context of CII tasks, easy black-boxes represent simple algorithms, while hard black-boxes encompass more complex algorithms and problem-specific programs. We instruct LLM to use meaningless letters as variable names (e.g., $a, b$), so the variable name will not result in the leakage of code intent.

**(Easy 1) Quicksort Recursion**   This code implements the quick sort algorithm using recursion to sort an array of integers. Always choose the last number in the list as pivot.

**(Easy 2) Most Frequent Char**   Given a string, find the character that appears most frequently. If there are multiple characters with the same frequency, return the one that appears first in the string. The input string can contain letters, digits, and punctuation marks. The output should be a single character that is the most frequent in the input string.

**(Easy 3) KMP**   The Knuth-Morris-Pratt (KMP) algorithm is used for substring search. It preprocesses the pattern to create a longest prefix-suffix (LPS) array, which is then used to skip unnecessary comparisons in the text. This allows for efficient searching of a pattern within a text string.

**(Easy 4) High Precision Divide**   Simulate high-precision division $a/b$ (including floating-point results), without using any floating-point numbers in the code. Keep to 4 decimal places.

**(Easy 5) High Precision Add**   This algorithm performs addition of very large integers. By simulating the columnar addition, it adds digits from the least significant (rightmost) position, handling carries.

**(Easy 6) Fib Recursion**   Give $n$, it returns the $n$-th number in the Fibonacci sequence using recursion.

**(Easy 7) Factorial Recursion**   Given an integer $n$, return the factorial of $n$ using recursion. The factorial of $n$ (denoted as $n!$) is the product of all positive integers less than or equal to $n$.

**(Easy 8) Exgcd**   The Extended Euclidean Algorithm computes the greatest common divisor (GCD) of two integers and also finds the coefficients $(x, y)$ such that $ax + by = gcd(a, b)$.

**(Easy 9) Coins for Fowls**   You have m coins and you want to buy exactly n fowls. Each rooster costs 5 coin, each hen costs 3 coins, and each chick costs 1 coin. This code solves how many Roosters, Hens, and Chicks can you buy.

**(Easy 10) Bubble Sort**   Bubble sort is a sorting algorithm that repeatedly steps through the list, compares adjacent elements and swaps them if they are in the wrong order. The pass through the list is repeated until the list is sorted.

**(Easy 11) Random Algebraic Operations**   Perform basic algebraic operations on three input int numbers $a, b, c$. First, calculate $d = a + b + c$. Then, calculate $e = a \times b - c$. Finally, calculate $f = (e + d - 10)/2$.

**(Hard 1) Sieve of Eratosthenes**   Implement the Sieve of Eratosthenes algorithm to find all prime numbers up to a given limit.

**(Hard 2) Mergesort Recursion**   Merge sort is a divide-and-conquer algorithm that sorts an array by recursively splitting it into halves, sorting each half, and then merging the sorted halves back together.

**(Hard 3) Manacher** Use Manacher's algorithm to find the longest palindromic substring in a given string. The algorithm works by transforming the input string to handle even-length palindromes and then expanding around potential centers to find the longest palindrome efficiently.

**(Hard 4) Heapsort** Heap sort is a comparison-based sorting algorithm that uses a binary heap data structure. It first builds a max heap from the input data, then repeatedly extracts the maximum element from the heap and rebuilds the heap until all elements are sorted.

**(Hard 5) Complex Random Algebraic Operations** Perform basic algebraic operations on four input int numbers $a, b, c, d$. First, calculate $e = a \times b + c \times d$. If this value is greater than 50, $e = e - 10$, else $e = e + 10$. Then, calculate $f = e^2 - a - b - c - d$. If f is odd, $f = (f - 1)/2$, else $f = f/2$. Finally, calculate $g = f - a$.

**(Hard 6) Arithmetic Slices** A sequence is defined as an arithmetic progression if it contains at least three elements and the difference between any two consecutive elements is constant. Given an integer array, return the number of all arithmetic subsequences within the array.

## G.2 CIRCUIT RULE INFERENCE (CRI)

Black-boxes in CRI task represents a boolean circuit without circle, whose input wires are a fixed number of 0/1 bits, and consisted of "AND"/"OR"/"NOT" gates. Easy black-boxes are typically formatted in sequence structure and implement basic function (e.g., xor), while some hard black-boxes are formatted in tree structure and implement advanced function (e.g., add).

**(Easy 1) Swap** For input size $n = 9$, let the first 4 output gates are the last 4 input wires, and last 5 output gates are first 5 input wires. Notice that we use b[i]=a[j] and a[j] to implement copy.

**(Easy 2) Random Small** For input size $n = 4$, we construct a circuit of 8 gates without nothing restriction. The circuit is not for some well-known purpose.

**(Easy 3) Consequence** For input size $n = 8$, we construct a circuit output whether there are two consecutive input wires are both 1. Let input wires are a[i], b[i] = a[i-1] and a[i], s[i] = b[i] or s[i-1].

**(Easy 4) Xor Sequence** For input size $n = 8$, we construct a circuit calculate prefix XOR. Notice that a xor b = (a and (not b)) or ((not a) and b), s[i] = s[i-1] xor a[i].

**(Easy 5) Palindrome** For input size $n = 8$, let a[i] be the input wires, and we construct a circuit check whether the input wires form a palindrome and a[1..4] = a[5..8]. b[i] = a[i] and a[n-i+1], c[i] = a[i] or a[n-i+1], d[i] = b[i] or (not c[i]), e[i] = a[i] and a[i+n/2], f[i] = a[i] or a[i+n/2], g[i] = e[i] or (not f[i]), s[i] = s[i-1] and d[i] and g[i].

**(Hard 1) Random Big** For input size $n = 7$, we construct a circuit of 16 gates without nothing restriction. The circuit is not for some well-known purpose.

**(Hard 2) Greater** For input size $n = 6$, we construct a circuit to check whether the number of input 1 wires is greater than or equal to the number of 0 wires. If number of 1 is greater or equal, there must be three input wires are 1. Let the input be a[1..6], c[1..20] be whether there are three 1 wires. For example, c[1] = a[1] and a[2] and a[3], c[2] = a[1] and a[2] and a[4], ..., c[20] = a[4] and a[5] and a[6]. The result is c[1] or ... or c[20].

**(Hard 3) Path** For input size $n = 9$, it can be treated as a $4\times4$ adjacent matrix G of a 3 nodes directed graph, where the diagnol is not important. We should construct a circuit that compute whether there are paths between every pair of nodes. Because every possible path has length less than 2, G'[i][j] = OR ( G[i][k] and G[k][j] ). Then G' is the result. Notice that G[i][i]=1 no matter what the input is.

**(Hard 4) Matrixmul** For input size $n = 9$, it can be treated as a 3×3 matrix M in field $Z_2$. Calculate the matrix square $M^2$. Notice that $M^2[i][j]$ = XOR (M[i][k] and M[k][j]). a xor b = (a and (not b)) or (b and (not a)).

**(Hard 5) Count** For input size $n = 7$, we construct a circuit to count how many input wires are 1. Construct a subcircuit of add 0/1: x'[0] = x[0] xor input, y[0] = x[0] and input, x'[1] = x[1] xor y[0], y'[1] = x[1] and y[0], x'[2] = x[2] xor y[1]. Copy the subcircuit n times. Notice that a xor b = (not a and b) or (not b and a).

**(Hard 6) Arbitrary** For input size $n = 7$ and an arbitrary function $f : Z_2^7 \rightarrow Z_2$. We can construct a circuit to compute $f$. Let $f(x_1) = 1$, $f(x_2) = 1$, ... $f(x_k) = 1$. $f(x) = (x = x_1)$ or $(x = x_2)$ or ... or $(x = x_k)$, $(x = y)$ can be construct by ((x[1] and y[1]) or not(x[1] and y[1])) and ((x[2] and y[2]) or not(x[2] and y[2])) and ... We choose $x_1, x_2, ..., x_8$ without any particular pattern and construct the circuit according to that.

**(Hard 7) And Tree** For an input size of $n = 8$, we construct a circuit with $n - 1$ gates. The circuit forms a tree structure, where each gate has two children and the outuput is the AND of its two children. The last gate outputs the AND of all input wires. The second-to-last gate outputs the AND of input wires with indices [1, n/2], while the third-to-last gate outputs the AND of input wires with indices [n/2+1, n]. The last gate outputs the AND of the results from the second-to-last and third-to-last gates...

**(Hard 8) Add** For input size $n = 10$, let the input wires be a[i], we construct a circuit to compute a[2]a[3]a[4]... + a[1], which means add the first bit to the binary number of the last n-1 bits. result[i]=a[n-i+1] xor b[i-1], b[i]=a[n-i+1] and b[i-1], b[0]=1.

**(Hard 9) Consequence k** For input size $n = 10$, we construct a circuit output whether there four two consecutive input wires are both 1 (we should treat the input wires as a circle, which means x[8] x[9] x[10] x[1] are also consecutive. Let s[i] = x[i] and x[next(i)], s'[i] = s[i] and s[next(next(i))], where next(i) = i mod 10 +1. s'[i] checks whether there is consecutive 1 begins at i.

**(Hard) Compare 10** For input size $n = 10$ $(x_1, ...x_5, y_1, ..., y_5)$, we construct a circuit for compare $x$ and $y$. That is, we compare $(x_1, y_1)(x_2, y_2) \ldots (x_5, y_5)$ in order. $a > b$ can be construct by $(a$ and $(not\ b))$.

G.3 PHYSICS SYSTEM INFERENCE (PSI)

Easy black-boxes contain the most basic classical mechanics laws governing the fundamental motion of one or two objects, while hard black-boxes contain two or more objects with more complicated mechanics laws. Although the motion trajectories of most objects in the implemented black box are one-dimensional or two-dimensional, for the sake of universality, we still performed three-dimensional modeling for all object trajectories. In some cases (Double Pendulum, Harmonic with Friction, Ball Air Resistance), equations of motion do not have analytical solutions. In these situations, we instruct LLMs to output code containing numerical solutions for the differential equations. Then we compare black-box output with code output on test samples.

**(Easy 1) Two Balls Collision Type 1** Two balls, A and B, are on an infinitely long, smooth plane. Ball A has a mass of 5 kg, and Ball B has a mass of 2 kg. Ball A starts from the origin and moves uniformly along the x-axis at a speed of 4 m/s. Ball B is placed at rest 10 m from the origin. When the two balls collide, they bounce off each other completely elastically.

**(Easy 2) Two Balls Collision Type 2** Two balls, A and B, are on an infinitely long, smooth plane. Ball A has a mass of 4 kg, and Ball B has a mass of 2 kg. Ball A starts from the origin and moves uniformly along the x-axis at a speed of 3 m/s. Ball B is placed at 12 m from the origin and come to Ball A at a speed of 5 m/s. When the two balls collide, they bounce off each other completely elastically.

**(Easy 3) Simple Harmonic Type 1**  A simple harmonic oscillator is a system that experiences a restoring force proportional to the displacement from its equilibrium position. The system contains one object. The mass of the object is 1 kg. The spring constant is 100 N/m. The initial displacement from the equilibrium position is 0.2 m (spring in contracted state). The system is horizontal and the object is support by smooth ground without friction. The system is released from rest and do not consider any air resistance. The coordinate origin is set at the equilibrium position of the oscillator.

**(Easy 4) Simple Harmonic Type 2**  A simple harmonic oscillator is a system that experiences a restoring force proportional to the displacement from its equilibrium position. The system contains one object. The mass of the object is 1 kg. The spring constant is 100 N/m. The initial displacement from the equilibrium position is 0.2 m (spring in contracted state). The system is vertical and the gravity is 10 m/s$^2$. The system is released from rest and do not consider any air resistance. The coordinate origin is set at the initial position of the oscillator.

**(Easy 5) Oblique Projectile Motion**  An oblique projectile motion system is a scenario where an object is projected at an angle from a very high point and falls under the influence of gravity. The system contains one object. The mass of the object is 2 kg. The initial height from which the object is projected is infinite, meaning it starts at a very high altitude. The initial horizontal velocity of the object is 10 m/s, and the initial vertical velocity is 10 m/s. The simulation does not consider air resistance. The coordinate origin is set at the initial position. Gravity g is 10 m/s$^2$.

**(Easy 6) Pendulum**  A pendulum is a weight suspended from a pivot so that it can swing freely. The system contains one object. The initial angle of the pendulum is 60 degree. The length of the pendulum is 2 meters. The acceleration due to gravity is 10 m/s$^2$. Do not take air resistance into account. The pendulum is released from rest. The coordinate origin is set at the pivot point of the pendulum.

**(Easy 7) Elliptical Planetary Orbits**  An elliptical planetary orbit is a path followed by a planet around a star, where the orbit is an ellipse. The system contains one object which is the planet. The mass of the planet is $9 \times 10^{24}$ kg and the mass of the star is $8 \times 10^{29}$ kg. The periapsis is $6 \times 10^{10}$ m and the apoapsis is $9 \times 10^{10}$ m. The simulation does not consider any other forces except for the gravitational force between the planet and the star. The coordinate origin is set at the center of the star. G equals to $6.7 \times 10^{-11}$ Nm$^2$/kg$^2$.

**(Easy 8) Cycloid**  A cycloid is the curve traced by a fixed point on the circumference of a circle as it rolls along a straight line without slipping. The system contains one object which is the fixed point. The radius of the circle is 1 m. The circle moves in a uniform speed of 1 m/s. The coordinate origin is set at the fixed point where the circle starts rolling.

**(Easy 9) Conical Pendulum**  A conical pendulum is a pendulum that moves in a circular path while swinging at an angle. The system contains one object which is the pendulum bob. The mass of the pendulum bob is 2 kg. The length of the pendulum is 5 m. The angle between the string and the vertical is 30 degrees. The simulation does not consider air resistance. The coordinate origin is set at the point where the string is attached to the ceiling. Gravity g is 10 m/s$^2$.

**(Easy 10) Free Fall to the Ground Type 1**  A free fall system is a scenario where an object falls under the influence of gravity until it collides with the ground, and then bounces. The system contains one object. The mass of the object is 5 kg. The initial height from which the object is dropped is 10 m. The ground is modeled as a surface that can completely deform elastically upon impact, meaning it will return to its original shape after the collision. The simulation does not consider air resistance. The coordinate origin is set at the ground level. Gravity g is 10 m/s$^2$.

**(Easy 11) Free Fall Infinite Height**  An object is dropped from an infinite height. The acceleration due to gravity is 10 m/s$^2$. The object falls freely under the influence of gravity. Do not take air resistance into account. The coordinate origin is set at the point where the object is dropped.

**(Easy 12) Horizontal Projectile Motion**  A horizontal projectile motion system is a scenario where an object is projected horizontally from a very high point and falls under the influence of

gravity. The system contains one object. The mass of the object is 2 kg. The initial height from which the object is projected is infinite, meaning it starts at a very high altitude. The initial horizontal velocity of the object is 10 m/s. The simulation does not consider air resistance. The coordinate origin is set at the initial position. Gravity g is 10 m/s$^2$.

**(Easy 13) Free Fall to the Ground Type 2**   A free fall system is a scenario where an object falls under the influence of gravity until it collides with the ground, and then bounces. The system contains one object. The mass of the object is 5 kg. The initial height from which the object is dropped is 20 m. The ground is modeled as a surface that deform inelastically upon impact. The coefficient of restitution is 0.6. The simulation does not consider air resistance. The coordinate origin is set at the ground level. Gravity g is 10 m/s$^2$.

**(Hard 1) Boat in River**   A rectangular river is 2000 m long and 20 m wide. The river's flow velocity is proportional to the distance from the river banks, with the velocity being 0 at both banks and 10 m/s at the center of the river. A boat travels from one bank to the other perpendicular to the current at a constant relative speed of 2 m/s. At a point 5 m from the bank, due to a malfunction, the boat immediately turns around and heads back towards the original bank perpendicular to the current at a speed of 1 m/s.

**(Hard 2) Block Released from Incline Plane**   An inclined plane with an angle of 30 degrees and a mass of 10 kg is placed on a smooth horizontal surface. A wooden block with a mass of 5 kg is placed at a perpendicular height of 8 m on the inclined plane. There is no friction between the wooden block and the inclined plane. In the initial state, the wooden block's coordinates are (0, 0, 8) and the inclined plane's coordinates are (0, 0, 0). The system is released from rest. After the wooden block separates from the inclined plane, both the wooden block and the inclined plane will move along the smooth horizontal surface. g = 10 m/s$^2$.

**(Hard 3) Three Balls Collision**   There are three balls, A, B, and C, with masses of 4 kg, 3 kg, and 2 kg, respectively. Their initial positions are at 0 m, 10 m, and 20 m, and their initial velocities are 3 m/s, 3 m/s, and -4 m/s. The plane is 30 m long with walls at both ends. The collisions between the balls and between the balls and the walls are all perfectly elastic. The plane is perfectly smooth.

**(Hard 4) Harmonic with Friction**   A harmonic oscillator with friction is a system that experiences a restoring force proportional to the displacement from its equilibrium position, along with a damping force proportional to the velocity. The system contains one object. The mass of the object is 1 kg. The spring constant is 100 N/m. The initial displacement from the equilibrium position is 1 m (spring in contracted state). The system is horizontal and the object is supported by ground with dynamic friction coefficient 0.1, gravity equals to 10. The system is released from rest and do not consider any air resistance. The coordinate origin is set at the equilibrium position of the oscillator. The simulation time begins from 0 s to 100 s with a time step of 0.1 s.

**(Hard 5) Double Pendulum**   A double pendulum is a mechanical system that consists of two pendulums attached end to end. The first pendulum is attached to a fixed pivot, and the second pendulum is attached to the free end of the first pendulum. The system contains two objects. The first object weighs 1 kg and attached to a fixed pivot with a 1 meter rigid rod. The first object weighs 1 kg and attached to the first object with a 1 meter rigid rod. The initial angle of the first pendulum is 45 degrees, and the initial angle of the second pendulum is 45 degrees. The acceleration due to gravity is 10 m/s$^2$. Do not take air resistance into account. The coordinate origin is set at the pivot point of the first pendulum. The simulation time begins from 0 s to 100 s with a time step of 0.1 s. The system is released from rest.

**(Hard 6) Ball Air Resistance**   A 2 kg ball is thrown vertically upward from a horizontal surface with an initial velocity of 15 m/s. The air resistance is given by f=0.1v$^2$, where v is the ball's velocity. After reaching its highest point, the ball keeps falling as there's no ground. g = 10 m/s$^2$.

**(Hard 7) Two Balls Collision Type 3**   Two balls, A and B, are on a 20 m, smooth plane with walls on each side. Ball A has a mass of 5 kg, and Ball B has a mass of 3 kg. Ball A starts from one side of the plane and moves uniformly along the plane at a speed of 4 m/s. Ball B starts from the other

side of the plane and moves towards Ball A at a speed of 6 m/s. When the two balls collide, the collision is inelastic and the coefficient of restitution e = 0.8. When the balls collide with the walls, they bounce off elastically.

**(Hard 8) Bullet** A 2000 g object, 60m in length, slides on a horizontal surface with a coefficient of friction of 0.1. Its initial speed is 15 m/s. A 50 g bullet is flying horizontally at 400 m/s. After 1.5 seconds, the bullet horizontally embeds into the object. During the bullet's passage through the object, it experiences a resistive force f = 0.2v, where v is the bullet's instantaneous velocity. When the bullet go through the object, it continues horizontal uniform flight. Neglect the bullet's vertical drop. Use g = 10 m/s$^2$.

## G.4 Encryption Rule Inference (ERI)

For every letter, its plaintext and ciphertext correspond in all cases for most of the easy black-boxes. Some easy black-boxes (e.g. Zigzag Cipher and Curve Cipher) change the order of plaintext via specific algorithm. For hard black-boxes, the encrypted letters barely retain their original frequencies, which adds difficulty in decryption.

**(Easy 1) 01248 Encryption Type 2** For characters A-Z or a-z (case-insensitive), each letter is assigned a value from 1 to 26 (a/A=1, b/B=2, etc.). This value is then broken down into the sum of combinations of 1, 2, 4, 8. Use larger numbers first to sum. Then calculate the amount of 1, 2, 4, 8 separately and convert it to a four-digits number where the ones, tens, hundreds, and thousands places represent the amounts of 1, 2, 4, 8, respectively. For example, C=1+2=0011, j=2+8=1010, Z=2+8+8+8=3010. Black space is replaced by 0.

**(Easy 2) Backpack Encryption** It uses a superincreasing sequence as private key. The plaintext is a string of letters. The encryption process is as follows: Convert each letter to a binary value: a/A=00000, b/B=00001, ..., z/Z=11001. The public key is set as $\{34, 51, 58, 11, 35\}$. The ciphertext is obtained by summing the products of each public key element and its corresponding digit in the binary representation. For example, the letter 'c' is encrypted to $(0 \times 34) + (0 \times 51) + (0 \times 58) + (1 \times 11) + (0 \times 35) = 11$. Output the ciphertext as a string of numbers, separated by commas.

**(Easy 3) Caesar Cipher** Caesar cipher is a type of substitution cipher in which each letter in the plaintext is shifted 8 places down the alphabet. For example, A(a) would be replaced by I(i), B(b) would become J(j), and so on. For letters at the end of the alphabet, the shift wraps around to the beginning. For example, Y(y) would become G(g). Only take letters A-Z and a-z (case-sensitive) into account. Blank spaces are ignored.

**(Easy 4) ADHXZ Encryption** ADHXZ is a method of encryption that uses a 5×5 grid to encrypt. The encryption is case-insensitive and follows this replacement rule: 'a/A=AH, b/B=AA, c/C=XH, d/D=DA, e/E=ZH, f/F=HD, g/G=XA, h/H=DD, i/I=XD, j/J=XD, k/K=DZ, l/L=AX, m/M=ZA, n/N=HZ, o/O=DH, p/P=AZ, q/Q=HA, r/R=ZD, s/S=HX, t/T=AD, u/U=XX, v/V=HH, w/W=ZX, x/X=XZ, y/Y=ZZ, z/Z=DX'. Black spaces are kept.

|   | A | D | H | X | Z |
|---|---|---|---|---|---|
| A | b/B | t/T | a/A | l/L | p/P |
| D | d/D | h/H | o/O | z/Z | k/K |
| H | q/Q | f/F | v/V | s/S | n/N |
| X | g/G | i/I/j/J | c/C | u/U | x/X |
| Z | m/M | r/R | e/E | w/W | y/Y |

**(Easy 5) Bacon Cipher** This variant of Bacon's cipher is a method of encoding text using a binary system. Each letter of the alphabet is represented by a unique combination of five letters, either 'X'('x') or 'Y'('y'). 'X'('x') indicates 0 and 'Y'('y') indicates 1. The encryption is case-sensitive, meaning uppercase letters will be replaced by 'X', 'Y', and lowercase letters will be replaced by 'x', 'y'. The first thirteen letters of the alphabet, namely 'A' to 'M' will be replaced by 'XXXXX' to 'XYYXX', while the last thirteen letters, namely 'N' to 'Z' will be replaced by 'YXXYY' to 'YYYYY'. Blank spaces are ignored.

**(Easy 6) Zigzag Cipher**   Write the letters of plaintext in a zigzag path, moving downwards and then upwards across the 3 rows. Imagine writing on 3 parallel lines: when you reach the bottom row, move back up, and when you reach the top row, move back down, repeating this pattern until all plaintext characters are written. Spaces are ignored. After writing, read the characters from left to right, row by row, to form the ciphertext. For example, the zigzag path for 'HELLO WORLD' is

$$
\begin{array}{ccccccccc}
\text{H} & & & & \text{O} & & & & \text{L} \\
& \text{E} & & \text{L} & & \text{W} & & \text{R} & & \text{D} \\
& & \text{L} & & & & \text{O} & &
\end{array}
$$

So it's encrypted as 'HOLELWRDLO'.

**(Easy 7) Fibonacci Encryption**   Fibonacci cipher is a method of encryption that uses the Fibonacci sequence to determine the shift of each letter in the plaintext. Suppose each letter is represented by a=0, b=1, ..., z=25, A=26, B=27, ..., Z=51. The value of each encrypted letter is its value in plaintext plus a shift number from the Fibonacci sequence and then modulo 52. The Fibonacci sequence starts with 1 and 1, and each subsequent number is the sum of the two preceding ones. The first few numbers in the sequence are 1, 1, 2, 3, 5, 8, 13, ... For example, if the plaintext is 'HELLO', the first letter 'H' is shifted by 1 (the first Fibonacci number), resulting in 'I'. The second letter 'E' is shifted by 1 (the second Fibonacci number), resulting in 'F'. The third letter 'L' is shifted by 2 (the third Fibonacci number), resulting in 'N'. The fourth letter 'L' is shifted by 3 (the fourth Fibonacci number), resulting in 'O'. The fifth letter 'O' is shifted by 5 (the fifth Fibonacci number), resulting in 'T'. Therefore, the ciphertext is 'IFNOT'. Keep the blank space in the ciphertext.

**(Easy 8) Index Shift Encryption**   Index shift encryption is a method of encryption that uses the index of each letter in the plaintext to determine the shift for each letter. The index of first letter is 0. Suppose the letters are represented by a=0, b=1, ..., z=25, A=26, B=27, ..., Z=51. The value of each letter in ciphertext equals its value in plaintext plus a shift number (i.e. the number of index) and then modulo 52. Then convert the value to the corresponding letter. Keep the blank spaces in the ciphertext.

**(Easy 9) Curve Cipher**   There is a table with 999 rows and 6 columns. Each letter in plaintext is sequentially filled into the table row by row. Blank spaces are ignored. Uppercase and lowercase are kept. When all the plaintext letters are filled into the table, the ciphertext is generated by reading letters in the table in a zigzag manner, beginning from the last letter in the last available column (e.g. begin from 't' in the 2-th column for 'at' and begin from 'i' in the 6-th column for 'beautiful'). When reaching the top row, move back down, and when reaching the bottom row, move back up, repeating this pattern until all the letters are read. Letters in the same column are written together and letters in different columns are separated by blank space. For example, if the plaintext is 'Hello World', the table will be like

| H | e | l | l | o | W |
|---|---|---|---|---|---|
| o | r | l | d |   |   |

And the ciphertext is supposed to be 'W o dl ll re Ho'.

**(Easy 10) 01248 Encryption Type 1**   For characters A-Z or a-z (case-insensitive), the encryption is done by first replacing each character with a number based on its position in the alphabet, namely, a/A=1, b/B=2, ..., z/Z=26. Each number in 1 to 26 is replaced by the sum of combinations of 1, 2, 4, 8. Black space is replaced by 0. For example, 3 is replaced by 12, 10 is replaced by 28, 20 is replaced by 488. The replacement follows two rule: (1). Use larger numbers first to make sure the encrypted number has the fewest digits (e.g. use 18 for 9 instead of 1224). (2). Put small number in the front. (e.g. use 18 instead of 81). According to the above rules, string 'Hello World' is encrypted to '814484812480124881248288484'.

**(Easy 11) Substitution Encryption**   Simple substitution cipher is a method of encryption where each letter in the plaintext is replaced by a letter with a fixed relationship to it. The detailed corresponding relationship is as follows: Each letter in 'abcdefghijklmnopqrstuvwxyz' will be replaced by 'phqgiumeaylnofdxjkrcvstzwb'. The process is case-sensitive, meaning uppercase will keep as

uppercase and lowercase will keep as lowercase. If the plaintext contains blank spaces, keep them in the ciphertext.

**(Hard 1) Sequential Feedback Cipher**     Sequential feedback cipher encrypts letter in the plaintext one by one in a sequential order. Each letter in the alphabet is assigned a value A=0, B=1, ..., Z=25, a=26, b=27, ..., z=51. The initial hidden letter is set as 'b'. For each letter in the plaintext, the shift number is the value of previous encrypted letter. And for the first letter, the shift number is the value of initial hidden letter. The ciphertext for each letter is its original value plus the shift number and then modulo 52. For example, the letter 'A' in plaintext 'At' will be encrypted to 'b' because $(27 + 0) \bmod 52 = 27$. The letter 't' will be encrypted to 'U' because $(45 + 27) \bmod 52 = 20$. Blank spaces are kept in the ciphertext.

**(Hard 2) RSA Encryption**     The public key for RSA is set as $(e, n) = (13, 713)$, and the private key is set as $(d, n) = (1117, 713)$. The plaintext is a string of letters. The encryption process is as follows: First, convert each letter to its corresponding integer value: a/A=1, b/B=2, ..., z/Z=26. This procedure is case-insensitive. Then, split the plaintext into blocks with a size of 2. Calculate the value of each block in hexavigesimal. For example, the value of 'HE' is $8 \times 26 + 5 = 213$. Finally, perform traditional RSA algorithm to each block. Blank spaces in plaintext are ignored. Output the ciphertext as a string of integers, separated by commas.

**(Hard 3) Frequency Encryption**     For each letter in the plaintext, its ciphertext is affected by the frequency of letters before it in the plaintext. Suppose we have plaintext $P = p_1 p_2 ... p_n$ and the initial hidden keyword $p_0$ is set as 'H'. A-Z and a-z are represented by numerical values 0-25 and 26-51 respectively. For letters $p_i (i \geq 1)$, the value for encrypted letter $c_i$ is calculated as $c_i = (p_i + \text{shift}_i) \bmod 52$, where $\text{shift}_i$ is the value of the letter that has the highest frequency in $p_0 p_1 ... p_{i-1}$. If several letters have the same frequency, choose the letter according to the following preference order: 'A'≻'B'≻...≻'Z'≻'a'≻'b'≻...≻'z'. Keep the blank spaces in the ciphertext.

**(Hard 4) Dynamic Curve Cipher**     There is a table with 999 rows and $k$ columns, where $k = (l \bmod 3) + 3$, and $l$ is the number of letters in the plaintext. Each letter in plaintext is sequentially filled into the table row by row. Blank spaces are ignored. Uppercase and lowercase are kept. When all the plaintext letters are filled into the table, the ciphertext is generated by reading letters in the table in a zigzag manner, beginning from the last letter in the last available column (e.g. begin from 't' in the 2-th column for 'at' and begin from 'l' in the 3-th column for 'beautiful'). When reaching the top row, move back down, and when reaching the bottom row, move back up, repeating this pattern until all the letters are read. Letters in the same column are written together and letters in different columns are separated by blank space. For example, if the plaintext is 'Hello World', the ciphertext is supposed to be 'rl lo dWe Hol'.

**(Hard 5) Vigenere Encryption**     Vigenere cipher is a method of encryption that uses a keyword 'MEMORY' to encrypt the plaintext. Each letter in the plaintext is shifted by a number of positions determined by the corresponding letter in the keyword. The keyword is repeated to match the length of the plaintext. For example, if the plaintext is 'HELLOWORLD', then the keyword is repeated as 'MEMORYMEMO'. The first letter 'H' is shifted by 'M' (12 positions), resulting in 'T'. The second letter 'E' is shifted by 'E' (4 positions), resulting in 'I'. The encryption is case-insensitive. Both uppercase and lowercase letters will be treated as uppercase. Keep the blank space in the ciphertext.

**(Hard 6) Hill Cipher**     The hill cipher is a polygraphic substitution cipher based on linear algebra. Each letter is first mapped into a number (a/A=0, b/B=1, ..., z/Z=25). The hill cipher uses matrix multiplication to encrypt blocks of text. The key is a $2 \times 2$ square matrix $K = \begin{pmatrix} 3 & 5 \\ 1 & 2 \end{pmatrix}$, and the plaintext is divided into blocks with a length of 2. If the last block is fewer than 2 letters, use 'x' to fill. Each block is represented as a 2-D vector, and the encryption is performed by multiplying the key matrix with the plaintext vector modulo 26. The resulting vector is then converted back to letters. The encryption process is case-insensitive. Output the ciphertext in lowercase style. For example, 'Hi' will be encrypted to 'jx'. Keep the blank spaces in the ciphertext.

**(Hard 7) Positional Keyword Cipher**   This method uses a keyword 'Jackal' to determine how each letter in the plaintext is shifted. Each letter of the alphabet is assigned a value A=0, B=1, ..., Z=25, a=26, b=27, ..., z=51. Write the keyword repeatedly above the plaintext, matching each letter of the plaintext with a letter from the keyword. Black spaces are kept. For example, if plaintext is 'Hello World', then the corresponding keyword for matching is 'Jacka lJack'. For each letter, calculate the sum of the value of plaintext and keyword, then modulo 52 to get the value of the ciphertext letter. Finally, translate the value to corresponding letter. Since the keyword is repeated, the same letter in the plaintext will be paired with different letters of the keyword at different positions, leading to different ciphertext letters.

**(Hard 8) Compositional Encryption**   It's an encryption method that undergoes polyalphabetic substitution, affine transformation, and fixed permutation. The encryption process follows: For each English letter $P_i$ in the plaintext: (1). Letter to Number Conversion: Convert $P_i$ to its corresponding number $p_i$ (a/A=0, ..., z/Z=25). (2). Get Keyword Character: Let the keyword be $K$ with length $L_k$. Take the $j = (i \bmod L_k)$-th character $K_j$ from the keyword and convert it to its numerical value $k_j$. The keyword $K$ is set as 'LOVE'. (3). Polyalphabetic Substitution: Calculate $s_{1,i} = (p_i + k_j) \bmod 26$. (4). Affine Transformation: Calculate $s_{2,i} = (3 \times s_{1,i} + 10) \bmod 26$. (5). Fixed Permutation: Generate a permutation table $P_{table}$ and use it to find $s_{3,i} = P_{table}[s_{2,i}]$. (6). Number to Letter Conversion: Convert $s_{3,i}$ back to its corresponding letter $C_i$. (7). Keep Blank Spaces: Keep the blank spaces in the ciphertext.

**(Hard 9) Playfair Cipher**   Playfair is a method of encryption. It encrypts pairs of letters using a $5 \times 5$ grid of letters constructed from a keyword 'SECURITY'. The keyword is used to fill the grid, and the remaining letters of the alphabet are filled in order, skipping any letters already in the keyword. The letter 'J' is combined with 'I'. To encrypt, locate each letter pair in the grid: If they are in the same row, replace them with letters to their right; If they are in the same column, replace them with letters below; If they form a rectangle, replace them with letters on the same row but at the opposite corners. If a pair consists of two identical letters, insert a filler letter 'X' between them. If the plaintext has an odd number of letters, append a filler letter 'X' at the end. The ciphertext is case-sensitive and blank spaces are considered.

G.5   INTERACTIVE PUZZLE INFERENCE (IPI)

Easy black-boxes in IPI task are puzzles with simple rules and limited action space, while hard black-boxes have more complex rules and a much larger range of actions to choose from.

**(Easy 1) Three Arms Bandit**   In the 3-Arm Bandit problem, there are 3 slot machines named 'Bandit A', 'Bandit B', and 'Bandit C'. Each machine has a probability of winning, which is unknown to you. The player's goal is to discover the better bandit by choosing which machine to play over a series of rounds. You can choose either 'Bandit A', 'Bandit B', or 'Bandit C' in each round, and you will receive a reward (such as 'Reward: 0' or 'Reward: 1') based on the machine's winning probability. After a series of rounds, player need to answer 'Bandit A', 'Bandit B' or 'Bandit C' to indicate which bandit they believe has the higher winning probability.

**(Easy 2) Single Battleship**   Single Battleship is a simplified version of the classic Battleship game. In this game, the player has to guess the location of a single $1 \times 5$ battleship on a $9 \times 9$ grid. The player will make guesses by specifying coordinates on the grid, in the form of '(x, y)', from '(1, 1)' to '(9, 9)', and the game will provide feedback on whether the guess 'hits' or 'misses' the battleship. After a series of guesses, the player will provide the coordinates of the battleship in the format 'Row X' or 'Column Y', where 'X' and 'Y' are integers from 1 to 9, since a battleship must be placed either horizontally or vertically and occupies a whole row or column.

**(Easy 3) Wordle**   Wordle is a word-guessing game where players attempt to deduce a hidden 8-letter word (all uppercase). Each guess provides feedback (1).Correct letter in the correct position, represented by 'A'. (2).Correct letter but misplaced, represented by 'M'. (3). Letter not in the word, represented by 'X'. Players iteratively refine guesses using feedback until solving the word, typically within limited attempts. After a few times of querying, the player will give an 8-letter uppercase word answer, and he'll win if he answers correctly. Output the 8-letter uppercase word directly.

**(Easy 4) Heavy Coin** In the Heavy Coin puzzle, there are 100 coins, named 'Coin 1', 'Coin 2', ..., 'Coin 100'. Among these coins, one is heavier than the others. The task is to identify the heavy coin using a balance scale. The balance scale has two properties: (1).It can compare the weight of two groups of coins at a time. But it only returns balance or imbalance, without giving which side is heavier. (2).It will lie once in a random turn in every 10 interactions. When it lies, it will return the opposite of the truth. Player must make queries with the same format of this example: 'Left: Coin 1, Coin 2, Coin 3; Right: Coin 4, Coin 5, Coin 6'. The balance scale will return 'Balance' if both sides are balanced or 'Imbalance' if one side is heavier. After a series of queries, the player need to identify the heavy coin among the 100 coins in the form of 'Heavy Coin X', where 'X' is the number of the heavy coin.

**(Easy 5) Number Guessing** In the Number Guessing game, the player will try to guess a secret integer that is randomly chosen from [0, 100]. The player will make guesses in the form of 'Number X' (such as 'Number 10'). And after each guess, the player will receive feedback indicating whether the absolute value of guess minus ground truth number is 'Greater than 15' or 'Less than or equal to 15'. If the absolute value is 'Greater than 15', the player will receive 'Far', else if the absolute value is 'Less than or equal to 15', the player will receive 'Close'. After a series of guesses, the player needs to answer with the correct number they believe is the secret integer in the form of 'Number X'.

**(Hard 1) Heavy Light Coins** In the Heavy Light Coins puzzle, there are 152 coins, named 'Coin 1', 'Coin 2', ..., 'Coin 152'. Among these coins, one is heavier than the others, and another one is lighter than the others. If the normal coin weighs 1, then the heavy coin weighs 1.1 and the light coin weighs 0.9. The task is to identify the heavy coin and the light coin using a balance scale. The balance scale can compare the weight of two groups of coins at a time. For example, player can make queries like 'Left: Coin 1, Coin 2, Coin 3; Right: Coin 4, Coin 5, Coin 6'. The balance scale will return 'Left' if the left side is heavier, 'Right' if the right side is heavier, or 'Equal' if both sides are balanced. After a series of queries, the player need to identify the heavy coin and the light coin among the 152 coins in the form of 'Heavy Coin X; Light Coin Y', where 'X' and 'Y' is the number of the heavy coin and the light coin.

**(Hard 2) Wordle Hard** Wordle is a word-guessing game where players attempt to deduce a hidden 11-letter word (all uppercase). Each guess provides feedback (1).Correct letter in the correct position, represented by 'A'. (2).Correct letter but misplaced, represented by 'M'. (3).Letter not in the word, represented by 'X'. Players iteratively refine guesses using feedback until solving the word, typically within limited attempts. After a few times of querying, the player will give an 11-letter uppercase word answer, and he'll win if he answers the correct word. Output the 11-letter uppercase word directly.

**(Hard 3) Nerdle** Nerdle is a math-based puzzle game where players guess a hidden equation consisting of numbers and operators. Here we guarantee that the length of the equation is 15, including digits (0-9) and operators (+, -, *, /, =). For example, truth can be '10002*3+6=30012'. In each turn, player can guess a length 15 valid equation (such as '1+1+1+1=50' is invalid since this is not correct, and '1+1+1+1=4' is invalid since length is only 9). Each guess provides feedback: (1) Correct digit/operator in the correct position, represented by 'A'; (2) Correct digit/operator but misplaced, represented by 'M'; (3) Digit/operator not in the equation, represented by 'X'. Players iteratively refine guesses using feedback until solving the equation, within limited attempts. After a few turns of querying, the player will give a 15-character valid equation answer.

**(Hard 4) Nuts and Bolts** There are 8 bolts and 8 nuts of different sizes, named 'Bolt 1', ..., 'Bolt' and 'Nut 1' ... 'Bolt 8', where each bolt exactly matches one nut. We know that 'Bolt 1' is smaller than 'Bolt 2', 'Bolt 2' is smaller than 'Bolt 3', and so on. The goal is to find the matching nut for each bolt. In each turn, the player can make a query of 3 bolt-but pairs in the form of 'Bolt X1, Nut Y1; Bolt X2, Nut Y2; Bolt X3, Nut Y3', which returns 'small', 'large' or 'equal' to represent whether the nut is too small, too large, or a match for the bolt, for example 'small; small; large'. After a series of queries, the player must answer a permutation of the nuts, such as 'Nut 1, Nut 3, Nut 2, Nut 5, Nut 4, Nut 6, Nut 7, Nut 8', which represents the order of nuts that match the bolts

from 'Bolt 1' to 'Bolt 8'. The player will win if they correctly identify the matching nuts for all bolts.

**(Hard 5) Battleship**   Battleship is a strategic guessing game where the player attempts to figure out the location of 3 hidden ships on a $7 \times 7$ grid. 3 ships are $1 \times 3$, $1 \times 4$ and $1 \times 5$ respectively, and we use 'AAA', 'BBBB' and 'CCCCC' to represent them. The player will make guesses by specifying coordinates on the grid, in the form of '(x, y)', from '(1, 1)' to '(7, 7)', and the game will provide feedback on whether the guess 'hits' or 'misses' any ships. Besides the ships, there are two 1x1 bombs denoted as 'O'. When the coordinate is bomb, the game will also provide 'hits'. Note that the player can only know whether a ship or a bomb is attacked, but not which ship or bomb it is. After a series of guesses, the player need to distinguish the bombs and provide the coordinates of the ships by drawing a grid with 'A', 'B' and 'C' representing the ships, and '.' representing empty water cells. The grid must be 7x7 in size, and the ships must be placed either horizontally or vertically. For example:

```
..AAA..
.......
......C
.BBBB.C
......C
......C
......C
```

is a valid answer. The player must ensure that the ships do not overlap and that they fit within the grid boundaries. The player will win if they correctly identify the locations of all three ships within the allowed number of turns.

**(Hard 6) Quordle**   Quordle is a word-guessing game where players attempt to deduce four hidden 8-letter words (all uppercase) simultaneously. Each guess provides feedback for each word: (1).Correct letter in the correct position, represented by 'A'. (2).Correct letter but misplaced, represented by 'M'. (3).Letter not in the word, represented by 'X'. Players iteratively refine guesses using feedback until solving all four words, within limited attempts. In each turn, players ask a single 8-letter uppercase word, and will receive corresponding answer made up of 'AMX' for the four words. After a few times of querying, the player will give four 8-letter uppercase words answers, separated by commas. He will win if all four words are correct.

G.6   GAME STRATEGY INFERENCE (GSI)

Easy black-boxes in GSI are typically with fixed game strategy or games with easy rules, while hard black-boxes are with dynamic game strategy or games with difficult rules. Winning a round of game will gain one point. A tie will gain zero point and a loss will cost one point. Such design of score eliminates gain from randomness in a game. The expectation of a trivial strategy is supposed to achieve zero point. In real-world scenarios with a finite number of game rounds, an LLM's strategy can sometimes result in a negative score. When this happens, we consider it a trivial strategy and assign it a score of zero.

**(Easy 1) Load Shoot Defend Defender**   In each turn, a player can choose 'load': gain one bullet, 'scout': see the opponent's bullet count, 'shoot x': spend 'x' bullets to attack, or 'defend y': spend 'y' bullets to defend. Then, actions and the gaining points of both players are revealed, but the specific numerical values 'x' and 'y' are kept hidden. Each player starts with 0 bullets and can hold a maximum of 8. 'x' and 'y' cannot exceed the number of bullets they have currently. If one player 'shoot x' and the other 'defend y', the defense succeeds and the player wins 1 point if $y \geq x$, and the shooter wins -1 point. But if $y < x$, the attacker wins 1 point and the defender wins -1 point. A Shoot action will always succeed against an opponent who chooses to Load or Scout. When a simultaneous shoot happens, player with more bullets will win 1 point and the other win -1 point. The same bullets will result in a tie where both players win 0 point. When a simultaneous defend or scout or load happens, both players win 0 point. If player 'scout', he will receive opponent's bullet number. The game lasts for [total_turns] turns, both players try to maximize their scores. The

black-box plays the role of an opponent with the following strategy: Cycle through 'load', 'load', 'defend 2', 'load', 'defend 1',...

**(Easy 2) RPS7 Cycle**   Rock, paper, scissors, fire, water, air, sponge is a hand game where players simultaneously choose between rock, paper, scissors, fire, water, air, sponge. Rock triumphs over three opponents: it smashes Scissors, crushes Sponge, and puts out Fire. Paper also defeats three: it covers Rock, floats on Water, and fans Air. Scissors have three victories: they cut Paper, shred Sponge, and can create a spark to start a Fire. Fire is victorious against three as well: it melts Scissors, burns Paper, and scorches Sponge. Water overcomes three challengers: it erodes Rock, extinguishes Fire, and rusts Scissors. Air has three wins: it blows out Fire, erodes Rock, and evaporates Water. Finally, Sponge defeats its three foes: it soaks up Water, cleans Paper (rendering it useless), and utilizes its pockets to contain Air. In the game, two players repeat [total_turns] times, each time the winner gets 1 point, the loser gets -1 point, and if they are tied, 0 point each. The goal is to maximize total scores. The black-box plays the role of an opponent with the following strategy: cycling through 'rock', 'paper', 'scissors', 'fire', 'water', 'air', 'sponge' in order, starting with 'rock'.

**(Easy 3) RPS7 Random**   Rock, paper, scissors, fire, water, air, sponge is a hand game where players simultaneously choose between rock, paper, scissors, fire, water, air, sponge. Rock triumphs over three opponents: it smashes Scissors, crushes Sponge, and puts out Fire. Paper also defeats three: it covers Rock, floats on Water, and fans Air. Scissors have three victories: they cut Paper, shred Sponge, and can create a spark to start a Fire. Fire is victorious against three as well: it melts Scissors, burns Paper, and scorches Sponge. Water overcomes three challengers: it erodes Rock, extinguishes Fire, and rusts Scissors. Air has three wins: it blows out Fire, erodes Rock, and evaporates Water. Finally, Sponge defeats its three foes: it soaks up Water, cleans Paper (rendering it useless), and utilizes its pockets to contain Air. In the game, two players repeat [total_turns] times, each time the winner gets 1 point, the loser gets -1 point, and if they are tied, 0 point each. The goal is to maximize total scores. The black-box plays the role of an opponent with the following strategy: choose 'rock', 'paper', 'air' with the same probability.

**(Easy 4) Anti RPS Random**   Here we focus on a reversed version of Rock, Paper, Scissors. Rock, Paper, Scissors is a hand game where players simultaneously choose between rock, paper, or scissors, with rock crushing scissors, scissors cutting paper, and paper covering rock to determine the winner. In the Anti Rock, Paper, Scissors game, the winning rules are exactly opposite to the original ones. So scissors beat rock, paper beats scissors, and rock beats paper. In the game, two players repeat [total_turns] times, each time the winner gets 1 point, the loser gets -1 point, and if they are tied, 0 point each. The goal is to maximize total scores. The black-box plays the role of an opponent with the following strategy: choose 'rock' and 'scissors' with the same probability.

**(Easy 5) RPS7 Beat Last**   Rock, paper, scissors, fire, water, air, sponge is a hand game where players simultaneously choose between rock, paper, scissors, fire, water, air, sponge. Rock triumphs over three opponents: it smashes Scissors, crushes Sponge, and puts out Fire. Paper also defeats three: it covers Rock, floats on Water, and fans Air. Scissors have three victories: they cut Paper, shred Sponge, and can create a spark to start a Fire. Fire is victorious against three as well: it melts Scissors, burns Paper, and scorches Sponge. Water overcomes three challengers: it erodes Rock, extinguishes Fire, and rusts Scissors. Air has three wins: it blows out Fire, erodes Rock, and evaporates Water. Finally, Sponge defeats its three foes: it soaks up Water, cleans Paper (rendering it useless), and utilizes its pockets to contain Air. In the game, two players repeat [total_turns] times, each time the winner gets 1 point, the loser gets -1 point, and if they are tied, 0 point each. The goal is to maximize total scores. The black-box plays the role of an opponent with the following strategy: The first time you play a stone. And then each time you choose an action that can beat your opponent's last round action.

**(Easy 6) RPS7 Imitate Last**   Rock, paper, scissors, fire, water, air, sponge is a hand game where players simultaneously choose between rock, paper, scissors, fire, water, air, sponge. Rock triumphs over three opponents: it smashes Scissors, crushes Sponge, and puts out Fire. Paper also defeats three: it covers Rock, floats on Water, and fans Air. Scissors have three victories: they cut Paper, shred Sponge, and can create a spark to start a Fire. Fire is victorious against three as well: it

melts Scissors, burns Paper, and scorches Sponge. Water overcomes three challengers: it erodes Rock, extinguishes Fire, and rusts Scissors. Air has three wins: it blows out Fire, erodes Rock, and evaporates Water. Finally, Sponge defeats its three foes: it soaks up Water, cleans Paper (rendering it useless), and utilizes its pockets to contain Air. In the game, two players repeat [total_turns] times, each time the winner gets 1 point, the loser gets -1 point, and if they are tied, 0 point each. The goal is to maximize total scores. The black-box plays the role of an opponent with the following strategy: first choose 'fire', second choose 'air', and then imitate your opponent's behavior the time before last.

**(Hard 1) Comparing Cards Smart**    Comparing Cards is a game where two players play a card at the same time and compare the number. Initially, both players have [total_cards] cards, numbered from 1 to [total_cards]. In each turn, both players choose a card on his hand to play at the same time, in the form of 'card x', where 'x' is the number on the card between 1 and [total_cards], and this card must be available. The player with the higher number wins the turn and earns 1 point, while the other player earns -1 point. If both players play the same number, then both players earn 0 points. The game lasts for [total_cards] turns, both players try to maximize their scores. The black-box plays the role of an opponent with the following strategy: First choose the card with median value. If the opponent plays a card smaller than that, play accordingly to the order of 'median-1, median-2, ..., 1, median+1, median+2, ..., the highest value'. Else if the opponent plays a card larger than that, play accordingly to the order of 'median+1, median+2, ..., the highest value, median-1, median-2, ..., 1'.

**(Hard 2) Comparing Cards Slice**    Comparing Cards is a game where two players play a card at the same time and compare the number. Initially, both players have [total_cards] cards, numbered from 1 to [total_cards]. In each turn, both players choose a card on his hand to play at the same time, in the form of 'card x', where 'x' is the number on the card between 1 and [total_cards], and this card must be available. The player with the higher number wins the turn and earns 1 point, while the other player earns -1 point. If both players play the same number, then both players earn 0 points. The game lasts for [total_cards] turns, both players try to maximize their scores. The black-box plays the role of an opponent with the following strategy: All the cards are arranged in ascending order and divided equally into four piles, named a, b, c, and d, from smallest to largest. The cards are played in the following sequence: Randomly select and play cards from pile b, one at a time, until pile b is empty. Randomly select and play cards from pile d, one at a time, until pile d is empty. Randomly select and play cards from pile a, one at a time, until pile a is empty. Randomly select and play cards from pile c, one at a time, until pile c is empty.

**(Hard 3) Load Shoot Defend Smart**    In each turn, a player can choose 'load': gain one bullet, 'scout': see the opponent's bullet count, 'shoot x': spend 'x' bullets to attack, or 'defend y': spend 'y' bullets to defend. Then, actions and the gaining points of both players are revealed, but the specific numerical values 'x' and 'y' are kept hidden. Each player starts with 0 bullets and can hold a maximum of 8. 'x' and 'y' cannot exceed the number of bullets they have currently. If one player 'shoot x' and the other 'defend y', the defense succeeds and the player wins 1 point if y>=x, and the shooter wins -1 point. But if y<x, the attacker wins 1 point and the defender wins -1 point. A Shoot action will always succeed against an opponent who chooses to Load or Scout. When a simultaneous shoot happens, player with more bullets will win 1 point and the other win -1 point. The same bullets will result in a tie where both players win 0 point. When a simultaneous defend or scout or load happens, both players win 0 point. If player 'scout', he will receive opponent's bullet number. The game lasts for [total_turns] turns, both players try to maximize their scores. The black-box plays the role of an opponent with the following strategy: if the max turn is lower than 10, 'load' in the first 3 turns. Else, 'load' in the first 6 turns. If the opponent does not shoot in your 'load' turns, keep 'load' until your opponent shoot first or you have 8 bullets . Then you keep 'shoot 2' until you don't have enough bullets. Then 'load' until your opponent shoot first again. Then you keep 'shoot 2' until you don't have enough bullets, keep cycling this process.

**(Hard 4) RPS7 Mapping**    Rock, paper, scissors, fire, water, air, sponge is a hand game where players simultaneously choose between rock, paper, scissors, fire, water, air, sponge. Rock triumphs over three opponents: it smashes Scissors, crushes Sponge, and puts out Fire. Paper also defeats three: it covers Rock, floats on Water, and fans Air. Scissors have three victories: they cut Paper,

shred Sponge, and can create a spark to start a Fire. Fire is victorious against three as well: it melts Scissors, burns Paper, and scorches Sponge. Water overcomes three challengers: it erodes Rock, extinguishes Fire, and rusts Scissors. Air has three wins: it blows out Fire, erodes Rock, and evaporates Water. Finally, Sponge defeats its three foes: it soaks up Water, cleans Paper (rendering it useless), and utilizes its pockets to contain Air. In the game, two players repeat [total_turns] times, each time the winner gets 1 point, the loser gets -1 point, and if they are tied, 0 point each. The goal is to maximize total scores. The black-box plays the role of an opponent with the following strategy: When last turn your opponent played 'rock', play 'fire'. When last turn your opponent played 'paper', play 'water'. When last turn your opponent played 'scissors', play 'air'. When last turn your opponent played 'fire', play 'sponge'. When last turn your opponent played 'water', play 'rock'. When last turn your opponent played 'air', play 'paper'. When last turn your opponent played 'sponge', play 'scissors'.

**(Hard 5) Load Shoot Defend Attacker**    In each turn, a player can choose 'load': gain one bullet, 'scout': see the opponent's bullet count, 'shoot x': spend 'x' bullets to attack, or 'defend y': spend 'y' bullets to defend. Then, actions and the gaining points of both players are revealed, but the specific numerical values 'x' and 'y' are kept hidden. Each player starts with 0 bullets and can hold a maximum of 8. 'x' and 'y' cannot exceed the number of bullets they have currently. If one player 'shoot x' and the other 'defend y', the defense succeeds and the player wins 1 point if y>=x, and the shooter wins -1 point. But if y<x, the attacker wins 1 point and the defender wins -1 point. A Shoot action will always succeed against an opponent who chooses to Load or Scout. When a simultaneous shoot happens, player with more bullets will win 1 point and the other win -1 point. The same bullets will result in a tie where both players win 0 point. When a simultaneous defend or scout or load happens, both players win 0 point. If player 'scout', he will receive opponent's bullet number. The game lasts for [total_turns] turns, both players try to maximize their scores. The black-box plays the role of an opponent with the following strategy: Cycle through 'load', 'load', 'shoot 2' and 'load', 'shoot 1'. i.e. repeat 'load', 'load', 'shoot 2', 'load', 'shoot 1', 'load', 'load', 'shoot 2', ...

**(Hard 6) Load Shoot Defend Balance**    In each turn, a player can choose 'load': gain one bullet, 'scout': see the opponent's bullet count, 'shoot x': spend 'x' bullets to attack, or 'defend y': spend 'y' bullets to defend. Then, actions and the gaining points of both players are revealed, but the specific numerical values 'x' and 'y' are kept hidden. Each player starts with 0 bullets and can hold a maximum of 8. 'x' and 'y' cannot exceed the number of bullets they have currently. If one player 'shoot x' and the other 'defend y', the defense succeeds and the player wins 1 point if y>=x, and the shooter wins -1 point. But if y<x, the attacker wins 1 point and the defender wins -1 point. A Shoot action will always succeed against an opponent who chooses to Load or Scout. When a simultaneous shoot happens, player with more bullets will win 1 point and the other win -1 point. The same bullets will result in a tie where both players win 0 point. When a simultaneous defend or scout or load happens, both players win 0 point. If player 'scout', he will receive opponent's bullet number. The game lasts for [total_turns] turns, both players try to maximize their scores. The black-box plays the role of an opponent with the following strategy: keep cycling 'load', 'load', 'shoot 1', 'defend 1',...

# H    DETAILED EXPERIMENTAL RESULTS

We report detailed model performance of 19 benchmarked LLMs in 6 tasks in table form separately. The order of black-boxes remains the same as Section G.

Table 10: Detailed model performance in the ORACLE benchmark 10@1&1@1.

| Models | CII | | CRI | | PSI | | ERI | | IPI | | GSI | |
|---|---|---|---|---|---|---|---|---|---|---|---|---|
| | Easy | Hard | Easy | Hard | Easy | Hard | Easy | Hard | Easy | Hard | Easy | Hard |
| qwq-plus | 37.45% | 18.65% | 30.00% | 11.00% | 48.72% | 12.50% | 3.41% | 0.00% | 43.33% | 0.00% | 28.18% | 23.24% |
| qwen-plus* | 44.20% | 30.08% | 48.00% | 14.00% | 50.00% | 12.50% | 3.41% | 0.00% | 70.00% | 16.67% | 31.06% | 16.76% |
| qwen3-32b* | 51.56% | 16.03% | 34.00% | 12.00% | 34.62% | 6.25% | 3.41% | 0.00% | 50.00% | 5.56% | 35.97% | 26.02% |
| qwen3-235b* | 49.05% | 26.51% | 32.00% | 9.00% | 47.44% | 16.67% | 5.68% | 0.00% | 46.67% | 13.89% | 58.76% | 32.50% |
| o4-mini | **77.23%** | 36.35% | 48.00% | 21.00% | **61.54%** | 25.00% | **63.64%** | **15.28%** | 80.00% | 33.33% | 62.68% | 32.92% |
| o3-mini | 56.71% | 18.89% | 36.00% | 10.00% | 32.05% | 25.00% | 38.64% | 9.72% | 46.67% | 13.89% | 54.73% | 31.78% |
| o3 | 74.11% | **52.14%** | **64.00%** | **24.00%** | 60.26% | **43.75%** | 26.14% | **15.28%** | **83.33%** | **36.11%** | **79.43%** | 43.95% |
| o1 | 36.02% | 24.76% | 44.00% | 18.00% | 34.62% | 20.83% | 21.59% | 6.94% | 40.00% | 22.22% | 51.14% | 25.43% |
| gemini-2.5-pro | 72.77% | 39.29% | 54.00% | 23.00% | 53.85% | 41.67% | 17.05% | 1.39% | 73.33% | 33.33% | 61.60% | **46.76%** |
| gemini-2.5-flash* | 54.98% | 21.83% | 20.00% | 13.00% | 41.03% | 31.25% | 3.41% | 0.00% | 30.00% | 19.44% | 42.19% | 36.46% |
| gemini-2.5-flash | 57.27% | 27.22% | 20.00% | 10.00% | 28.21% | 18.75% | 3.41% | 0.00% | 13.33% | 2.78% | 14.08% | 22.18% |
| gemini-2.0-flash | 38.18% | 14.52% | 18.00% | 18.00% | 6.41% | 6.25% | 2.27% | 0.00% | 10.00% | 2.78% | 16.42% | 19.17% |
| deepseek-r1 | 45.71% | 39.05% | 44.00% | 17.00% | 53.85% | 16.67% | 1.14% | 0.00% | 53.33% | 19.44% | 52.60% | 25.33% |
| deepseek-v3 | 58.74% | 19.76% | 24.00% | 12.00% | 19.23% | 2.08% | 1.14% | 0.00% | 16.67% | 2.78% | 17.58% | 10.13% |
| claude-4-sonnet* | 64.03% | 26.35% | 44.00% | 19.00% | 43.59% | 12.50% | 5.68% | 0.00% | 53.33% | 30.56% | 47.59% | 27.84% |
| claude-4-sonnet | 53.33% | 23.17% | 32.00% | 13.00% | 15.38% | 2.08% | 4.55% | 0.00% | 43.33% | 13.89% | 32.27% | 17.33% |
| claude-3.7-sonnet* | 68.66% | 27.22% | 52.00% | 12.00% | 48.72% | 35.42% | 22.73% | 5.56% | 40.00% | 27.78% | 39.96% | 25.47% |
| claude-3.7-sonnet | 46.93% | 22.54% | 18.00% | 12.00% | 33.33% | 10.42% | 9.09% | 0.00% | 30.00% | 5.56% | 41.52% | 14.75% |
| claude-3.5-sonnet | 51.73% | 16.51% | 26.00% | 17.00% | 24.36% | 12.50% | 6.82% | 0.00% | 26.67% | 5.56% | 38.51% | 20.98% |

Table 11: Detailed model performance in the ORACLE benchmark 20@2&2@1.

| Models | CII | | CRI | | PSI | | ERI | | IPI | | GSI | |
|---|---|---|---|---|---|---|---|---|---|---|---|---|
| | Easy | Hard | Easy | Hard | Easy | Hard | Easy | Hard | Easy | Hard | Easy | Hard |
| qwq-plus | 62.60% | 38.89% | 34.00% | 12.00% | 48.72% | 14.58% | 3.41% | 0.00% | 60.00% | 2.78% | 25.93% | 18.30% |
| qwen-plus* | 61.13% | 42.30% | 70.00% | 30.00% | 43.59% | 18.75% | 3.41% | 0.00% | 76.67% | 22.22% | 38.99% | 16.29% |
| qwen3-32b* | 67.88% | 37.62% | 50.00% | 10.00% | 29.49% | 16.67% | 6.82% | 0.00% | 83.33% | 13.89% | 38.46% | 38.36% |
| qwen3-235b* | 77.10% | 34.76% | 56.00% | 16.00% | 47.44% | 22.92% | 4.55% | 0.00% | 76.67% | 16.67% | 62.82% | 51.40% |
| o4-mini | **92.60%** | 58.02% | 76.00% | 35.00% | 64.10% | 35.42% | **61.36%** | **16.67%** | 86.67% | 44.44% | 55.11% | 23.21% |
| o3-mini | 75.11% | 34.76% | 52.00% | 19.00% | 41.03% | 29.17% | 43.18% | 11.11% | 76.67% | 16.67% | 49.50% | 34.22% |
| o3 | 90.52% | **74.84%** | **84.00%** | 47.00% | **65.38%** | 37.50% | 51.14% | 5.56% | **96.67%** | **66.67%** | **82.51%** | **54.79%** |
| o1 | 45.28% | 24.84% | 62.00% | 17.00% | 42.31% | 20.83% | 12.50% | 0.00% | 83.33% | 30.56% | 49.92% | 34.11% |
| gemini-2.5-pro | 88.05% | 57.30% | **84.00%** | 52.00% | **65.38%** | 31.25% | 26.14% | **16.67%** | **96.67%** | 44.44% | 60.77% | 41.02% |
| gemini-2.5-flash* | 71.90% | 35.08% | 44.00% | 31.00% | 35.90% | 16.67% | 6.82% | 0.00% | 80.00% | 36.11% | 52.83% | 26.89% |
| gemini-2.5-flash | 60.48% | 23.81% | 34.00% | 11.00% | 35.90% | 22.92% | 6.82% | 0.00% | 20.00% | 5.56% | 31.48% | 19.51% |
| gemini-2.0-flash | 39.61% | 5.87% | 44.00% | 21.00% | 0.00% | 2.08% | 4.55% | 1.39% | 13.33% | 2.78% | 17.96% | 16.39% |
| deepseek-r1 | 73.77% | 43.73% | 70.00% | 27.00% | 50.00% | 31.25% | 3.41% | 0.00% | 63.33% | 41.67% | 45.71% | 24.58% |
| deepseek-v3 | 70.48% | 23.17% | 44.00% | 21.00% | 20.51% | 6.25% | 3.41% | 0.00% | 40.00% | 0.00% | 20.54% | 14.34% |
| claude-4-sonnet* | 84.03% | 48.33% | 56.00% | 41.00% | 55.13% | 27.08% | 23.86% | 1.39% | 83.33% | 41.67% | 62.97% | 36.27% |
| claude-4-sonnet | 71.99% | 34.44% | 42.00% | 22.00% | 28.21% | 4.17% | 4.55% | 0.00% | 60.00% | 16.67% | 44.31% | 20.28% |
| claude-3.7-sonnet* | 78.66% | 46.59% | 64.00% | 34.00% | 56.41% | **37.50%** | 18.18% | 0.00% | 80.00% | 38.89% | 49.54% | 32.36% |
| claude-3.7-sonnet | 69.18% | 37.30% | 52.00% | 26.00% | 34.62% | 12.50% | 11.36% | 0.00% | 56.67% | 16.67% | 41.61% | 17.78% |
| claude-3.5-sonnet | 64.42% | 33.65% | 36.00% | 20.00% | 34.62% | 10.42% | 7.95% | 0.00% | 60.00% | 8.33% | 32.69% | 8.73% |

Table 12: Detailed model performance on CII task in the ORACLE benchmark 10@1.

| Models | Easy Black-Box in CII | | | | | | | | | | | Hard Black-Box in CII | | | | | |
|---|---|---|---|---|---|---|---|---|---|---|---|---|---|---|---|---|---|
| | 1 | 2 | 3 | 4 | 5 | 6 | 7 | 8 | 9 | 10 | 11 | 1 | 2 | 3 | 4 | 5 | 6 |
| qwq-plus | 0.23 | 0.54 | 0 | 0.3 | 0.33 | 0.11 | 0.2 | 0.9 | 0.47 | 0.43 | 0.6 | 0 | 0.14 | 0.2 | 0.26 | 0.09 | 0.43 |
| qwen-plus* | 0.53 | 0.43 | 0.37 | 0.43 | 0.27 | 0.11 | 0.37 | 0.97 | 0.4 | 0.89 | 0.63 | 0.4 | 0.4 | 0 | 0.17 | 0.29 | 0.33 |
| qwen3-32b* | 0.37 | 0.63 | 0 | 0.43 | 0.3 | 0.09 | 0.3 | 0.87 | 0.57 | 0.66 | 0.66 | 0.2 | 0.31 | 0.3 | 0.37 | 0.29 | 0.33 |
| qwen3-235b* | 0.5 | 0.46 | 0.54 | 0.53 | 0.5 | 0.43 | 0.07 | 0.97 | 0.3 | 0.63 | 0.74 | 0.06 | 0.23 | 0.1 | 0.11 | 0.23 | 0.23 |
| o4-mini | 0.97 | 0.91 | 0.54 | 0.87 | 0.8 | 0.66 | 0.83 | 0.97 | 0.63 | 0.6 | 0.71 | 0.37 | 0.4 | 0.23 | 0.23 | 0.31 | 0.63 |
| o3-mini | 0.57 | 0.77 | 0.69 | 0.33 | 0.23 | 0.37 | 0.9 | 0.53 | 0.5 | 0.89 | 0.46 | 0.2 | 0.09 | 0.17 | 0.03 | 0.29 | 0.37 |
| o3 | 0.93 | 0.8 | 0.29 | 0.77 | 0.87 | 0.54 | 1 | 0.93 | 0.57 | 0.74 | 0.71 | 0.54 | 0.49 | 0.5 | 0.51 | 0.29 | 0.8 |
| o1 | 0.8 | 0.11 | 0.26 | 0 | 0.03 | 0.49 | 0.3 | 0.5 | 0.1 | 0.77 | 0.6 | 0.4 | 0.34 | 0.13 | 0.2 | 0.14 | 0.27 |
| gemini-2.5-pro | 0.73 | 0.77 | 0.63 | 0.57 | 0.77 | 0.4 | 0.97 | 0.87 | 0.73 | 0.86 | 0.71 | 0.63 | 0.31 | 0.23 | 0.26 | 0.26 | 0.67 |
| gemini-2.5-flash* | 0.27 | 0.66 | 0.23 | 0.47 | 0.47 | 0.49 | 1 | 0.6 | 0.53 | 0.66 | 0.69 | 0 | 0.31 | 0.2 | 0.23 | 0.2 | 0.37 |
| gemini-2.5-flash | 0.57 | 0.8 | 0.29 | 0.57 | 0.6 | 0 | 1 | 0.63 | 0.73 | 0.46 | 0.66 | 0.23 | 0.37 | 0.2 | 0.26 | 0.14 | 0.43 |
| gemini-2.0-flash | 0.13 | 0.34 | 0.46 | 0.53 | 0.3 | 0.57 | 0 | 0.53 | 0.3 | 0.57 | 0.46 | 0.09 | 0.54 | 0.07 | 0 | 0.14 | 0.03 |
| deepseek-r1 | 0.07 | 0.11 | 0.43 | 0.33 | 0.53 | 0.49 | 0.43 | 0.73 | 0.5 | 0.86 | 0.54 | 0.46 | 0.29 | 0.27 | 0.37 | 0.23 | 0.73 |
| deepseek-v3 | 0.7 | 0.57 | 0.34 | 0.6 | 0.6 | 0.66 | 0.8 | 0.67 | 0.47 | 0.6 | 0.46 | 0.11 | 0.29 | 0.03 | 0.34 | 0.14 | 0.27 |
| claude-4-sonnet* | 0.67 | 0.91 | 0.69 | 0.53 | 0 | 0 | 1 | 1 | 0.5 | 0.94 | 0.8 | 0.34 | 0.37 | 0.23 | 0.37 | 0.03 | 0.23 |
| claude-4-sonnet | 0.43 | 0.6 | 0.26 | 0.5 | 0.43 | 0.51 | 1 | 0.53 | 0.57 | 0.57 | 0.46 | 0.23 | 0.46 | 0.27 | 0.17 | 0 | 0.27 |
| claude-3.7-sonnet* | 0.37 | 0.77 | 0.57 | 0.57 | 0.67 | 0.51 | 0.93 | 0.97 | 0.77 | 0.89 | 0.54 | 0.09 | 0.34 | 0.1 | 0.26 | 0.51 | 0.33 |
| claude-3.7-sonnet | 0.13 | 0.57 | 0.57 | 0.53 | 0.5 | 0.57 | 0.33 | 0.53 | 0.5 | 0.46 | 0.46 | 0.31 | 0.23 | 0.1 | 0.2 | 0.14 | 0.37 |
| claude-3.5-sonnet | 0.3 | 0.34 | 0.34 | 0.5 | 0.33 | 0.54 | 0.97 | 0.5 | 0.63 | 0.57 | 0.66 | 0.23 | 0.23 | 0.13 | 0.26 | 0.14 | 0 |

Table 13: Detailed model performance on CII task in the ORACLE benchmark 20@2.

| Models | Easy Black-Box in CII | | | | | | | | | | | Hard Black-Box in CII | | | | | |
|---|---|---|---|---|---|---|---|---|---|---|---|---|---|---|---|---|---|
| | 1 | 2 | 3 | 4 | 5 | 6 | 7 | 8 | 9 | 10 | 11 | 1 | 2 | 3 | 4 | 5 | 6 |
| qwq-plus | 0.34 | 0.94 | 0.83 | 0.53 | 0.37 | 0.23 | 0.43 | 0.97 | 0.5 | 0.97 | 0.77 | 0.37 | 0.6 | 0.2 | 0.34 | 0.29 | 0.53 |
| qwen-plus* | 0.2 | 0.91 | 0.43 | 0.53 | 0.8 | 0.34 | 0.53 | 1 | 0.4 | 0.8 | 0.77 | 0.51 | 0.63 | 0.3 | 0.37 | 0.29 | 0.43 |
| qwen3-32b* | 0.43 | 0.94 | 0.86 | 0.57 | 0.67 | 0.51 | 0.27 | 0.97 | 0.6 | 0.86 | 0.8 | 0.34 | 0.53 | 0.27 | 0.29 | 0.43 | 0.4 |
| qwen3-235b* | 0.6 | 0.46 | 0.89 | 0.63 | 0.87 | 0.6 | 0.9 | 1 | 0.77 | 1 | 0.77 | 0.34 | 0.5 | 0.27 | 0.23 | 0.31 | 0.43 |
| o4-mini | 0.77 | 0.94 | 1 | 0.9 | 1 | 0.91 | 1 | 1 | 0.8 | 0.91 | 0.94 | 0.31 | 0.93 | 0.3 | 0.54 | 0.46 | 0.93 |
| o3-mini | 0.37 | 0.86 | 0.83 | 0.63 | 0.7 | 0.66 | 1 | 0.9 | 0.6 | 0.91 | 0.8 | 0.6 | 0.43 | 0.37 | 0.2 | 0.29 | 0.2 |
| o3 | 0.66 | 1 | 1 | 0.9 | 1 | 0.77 | 1 | 1 | 0.8 | 0.83 | 1 | 0.74 | 1 | 0.67 | 0.77 | 0.34 | 0.97 |
| o1 | 0.63 | 0.69 | 1 | 0.2 | 0 | 0 | 0.13 | 0.87 | 0.07 | 0.83 | 0.57 | 0.43 | 0.57 | 0.03 | 0.11 | 0.31 | 0.03 |
| gemini-2.5-pro | 0.94 | 0.94 | 0.94 | 0.63 | 0.97 | 0.57 | 1 | 0.93 | 0.87 | 0.94 | 0.94 | 0.69 | 0.77 | 0.3 | 0.46 | 0.43 | 0.8 |
| gemini-2.5-flash* | 0.46 | 0.97 | 0.43 | 0.63 | 0.83 | 0.43 | 1 | 0.8 | 0.9 | 0.74 | 0.71 | 0 | 0.5 | 0.3 | 0.34 | 0.43 | 0.53 |
| gemini-2.5-flash | 0.46 | 0 | 0.49 | 0.6 | 0.9 | 0.43 | 1 | 0.73 | 0.73 | 0.63 | 0.69 | 0 | 0.63 | 0.17 | 0.46 | 0.17 | 0 |
| gemini-2.0-flash | 0.14 | 0.71 | 0 | 0.53 | 0.4 | 0.43 | 0.13 | 0.63 | 0.4 | 0.51 | 0.46 | 0.09 | 0.03 | 0.03 | 0 | 0.2 | 0 |
| deepseek-r1 | 0.74 | 0.91 | 0.54 | 0.83 | 0.97 | 0.37 | 0.8 | 0.73 | 0.67 | 0.8 | 0.74 | 0.49 | 0.77 | 0.37 | 0.6 | 0.17 | 0.23 |
| deepseek-v3 | 0.6 | 0.77 | 0.54 | 0.7 | 0.8 | 0.66 | 1 | 0.67 | 0.7 | 0.71 | 0.6 | 0.26 | 0.5 | 0.23 | 0.26 | 0.14 | 0 |
| claude-4-sonnet* | 0.74 | 0.94 | 0.91 | 0.67 | 0.9 | 0.63 | 1 | 0.9 | 0.83 | 0.91 | 0.8 | 0.51 | 0.77 | 0.33 | 0.34 | 0.14 | 0.8 |
| claude-4-sonnet | 0.63 | 0.91 | 0.86 | 0.63 | 0.6 | 0.51 | 1 | 0.67 | 0.73 | 0.71 | 0.66 | 0.37 | 0.53 | 0.2 | 0.29 | 0.14 | 0.53 |
| claude-3.7-sonnet* | 0.43 | 0.94 | 0.89 | 0.6 | 0.9 | 0.57 | 1 | 0.97 | 0.9 | 0.66 | 0.8 | 0.49 | 0.5 | 0.2 | 0.37 | 0.77 | 0.47 |
| claude-3.7-sonnet | 0.54 | 0.69 | 0.71 | 0.63 | 0.8 | 0.66 | 1 | 0.7 | 0.73 | 0.46 | 0.69 | 0.51 | 0.63 | 0.17 | 0.29 | 0.17 | 0.47 |
| claude-3.5-sonnet | 0.6 | 0.66 | 0.6 | 0.7 | 0.6 | 0.71 | 1 | 0.63 | 0.47 | 0.46 | 0.66 | 0.43 | 0.57 | 0.13 | 0.31 | 0.14 | 0.43 |

Table 14: Detailed model performance on CRI task in the ORACLE benchmark 10@1.

| Models | Easy Black-Box in CRI | | | | | Hard Black-Box in CRI | | | | | | | | | |
|---|---|---|---|---|---|---|---|---|---|---|---|---|---|---|---|
| | 1 | 2 | 3 | 4 | 5 | 1 | 2 | 3 | 4 | 5 | 6 | 7 | 8 | 9 | 10 |
| qwq-plus | 0.5 | 0.7 | 0.3 | 0 | 0 | 0.2 | 0 | 0.3 | 0 | 0 | 0 | 0.4 | 0 | 0.2 | 0 |
| qwen-plus* | 1 | 0.6 | 0.7 | 0.1 | 0 | 0.1 | 0.1 | 0.3 | 0.1 | 0.1 | 0 | 0.7 | 0 | 0 | 0 |
| qwen3-32b* | 0.8 | 0.6 | 0.1 | 0 | 0.2 | 0 | 0.1 | 0.5 | 0.1 | 0.1 | 0 | 0.4 | 0 | 0 | 0 |
| qwen3-235b* | 0.8 | 0.7 | 0.1 | 0 | 0 | 0 | 0 | 0.2 | 0.1 | 0 | 0 | 0.6 | 0 | 0 | 0 |
| o4-mini | 0.6 | 0.8 | 0.8 | 0 | 0.2 | 0 | 0.1 | 0.4 | 0 | 0.1 | 0 | 1 | 0 | 0.5 | 0 |
| o3-mini | 0.9 | 0.6 | 0.1 | 0 | 0.2 | 0.4 | 0 | 0.3 | 0 | 0 | 0 | 0.2 | 0 | 0.1 | 0 |
| o3 | 0.9 | 0.9 | 0.6 | 0.6 | 0.2 | 0.2 | 0.1 | 0.5 | 0 | 0.1 | 0 | 0.7 | 0.2 | 0.6 | 0 |
| o1 | 0.8 | 0.8 | 0.4 | 0 | 0.2 | 0.2 | 0 | 0.5 | 0.1 | 0.2 | 0 | 0.8 | 0 | 0 | 0 |
| gemini-2.5-pro | 0 | 0.9 | 1 | 0.5 | 0.3 | 0.1 | 0 | 0.4 | 0.1 | 0.2 | 0 | 1 | 0.2 | 0.3 | 0 |
| gemini-2.5-flash* | 0 | 0.8 | 0.1 | 0.1 | 0 | 0.2 | 0 | 0.5 | 0 | 0.1 | 0 | 0.4 | 0.1 | 0 | 0 |
| gemini-2.5-flash | 0 | 0.8 | 0.2 | 0 | 0 | 0.1 | 0 | 0.3 | 0 | 0.2 | 0 | 0.4 | 0 | 0 | 0 |
| gemini-2.0-flash | 0 | 0 | 0.2 | 0.1 | 0.6 | 0.1 | 0.3 | 0.5 | 0 | 0.4 | 0.2 | 0.2 | 0 | 0.1 | 0 |
| deepseek-r1 | 0.9 | 0.7 | 0.3 | 0.1 | 0.2 | 0 | 0.1 | 0.4 | 0.2 | 0.2 | 0 | 0.4 | 0 | 0.4 | 0 |
| deepseek-v3 | 0 | 0.1 | 0.5 | 0.2 | 0.4 | 0.2 | 0.1 | 0.6 | 0.2 | 0 | 0 | 0.1 | 0 | 0 | 0 |
| claude-4-sonnet* | 0.9 | 0.9 | 0.1 | 0.2 | 0.1 | 0.2 | 0.1 | 0.4 | 0.1 | 0.2 | 0 | 0.5 | 0.3 | 0.1 | 0 |
| claude-4-sonnet | 0.9 | 0.5 | 0 | 0 | 0.2 | 0.1 | 0.2 | 0.4 | 0.1 | 0.2 | 0 | 0.3 | 0 | 0 | 0 |
| claude-3.7-sonnet* | 0.7 | 0.9 | 0.4 | 0 | 0.6 | 0.1 | 0 | 0.5 | 0 | 0.2 | 0 | 0.3 | 0.1 | 0 | 0 |
| claude-3.7-sonnet | 0 | 0.5 | 0.2 | 0 | 0.2 | 0.1 | 0.1 | 0.5 | 0.2 | 0.3 | 0 | 0 | 0 | 0 | 0 |
| claude-3.5-sonnet | 0 | 0.7 | 0.3 | 0.1 | 0.2 | 0.1 | 0.1 | 0.5 | 0.2 | 0.3 | 0 | 0.5 | 0 | 0 | 0 |

Table 15: Detailed model performance on CRI task in the ORACLE benchmark 20@2.

| Models | Easy Black-Box in CRI | | | | | Hard Black-Box in CRI | | | | | | | | | |
|---|---|---|---|---|---|---|---|---|---|---|---|---|---|---|---|
| | 1 | 2 | 3 | 4 | 5 | 1 | 2 | 3 | 4 | 5 | 6 | 7 | 8 | 9 | 10 |
| qwq-plus | 0.9 | 0.7 | 0.1 | 0 | 0 | 0 | 0 | 0.4 | 0 | 0 | 0 | 0.8 | 0 | 0 | 0 |
| qwen-plus* | 1 | 0.9 | 1 | 0.1 | 0.5 | 0.7 | 0.3 | 0.7 | 0 | 0.2 | 0 | 1 | 0 | 0.1 | 0 |
| qwen3-32b* | 0.9 | 0.6 | 0.7 | 0.1 | 0.2 | 0.1 | 0 | 0.4 | 0 | 0 | 0 | 0.5 | 0 | 0 | 0 |
| qwen3-235b* | 1 | 0.9 | 0.6 | 0 | 0.3 | 0 | 0 | 0.5 | 0.1 | 0.1 | 0 | 0.8 | 0 | 0.1 | 0 |
| o4-mini | 1 | 1 | 1 | 0.3 | 0.5 | 0.8 | 0.1 | 0.5 | 0 | 0.3 | 0 | 1 | 0.1 | 0.7 | 0 |
| o3-mini | 1 | 1 | 0.5 | 0 | 0.1 | 0.6 | 0 | 0.4 | 0 | 0 | 0 | 0.9 | 0 | 0 | 0 |
| o3 | 1 | 1 | 1 | 0.8 | 0.4 | 0.8 | 0.3 | 0.8 | 0.2 | 0.3 | 0 | 1 | 0.3 | 0.9 | 0.1 |
| o1 | 1 | 1 | 0.6 | 0 | 0.5 | 0.4 | 0 | 0.6 | 0.1 | 0.1 | 0 | 0.4 | 0 | 0.1 | 0 |
| gemini-2.5-pro | 1 | 1 | 1 | 1 | 0.2 | 0.9 | 0.5 | 0.5 | 0.2 | 0.3 | 0 | 1 | 0.7 | 1 | 0.1 |
| gemini-2.5-flash* | 0.2 | 0.9 | 0.9 | 0 | 0.2 | 0.6 | 0.3 | 0.2 | 0.2 | 0.2 | 0 | 0.6 | 0.7 | 0.3 | 0 |
| gemini-2.5-flash | 0 | 1 | 0.7 | 0 | 0 | 0 | 0.3 | 0.3 | 0 | 0.2 | 0.1 | 0.2 | 0 | 0 | 0 |
| gemini-2.0-flash | 0.1 | 1 | 0.3 | 0.2 | 0.6 | 0.3 | 0.5 | 0.5 | 0 | 0.3 | 0.3 | 0.1 | 0 | 0 | 0.1 |
| deepseek-r1 | 1 | 1 | 1 | 0.1 | 0.4 | 0.1 | 0.1 | 0.5 | 0.2 | 0.3 | 0 | 1 | 0 | 0.5 | 0 |
| deepseek-v3 | 1 | 0.4 | 0.2 | 0.2 | 0.4 | 0.4 | 0.1 | 0.6 | 0.2 | 0.1 | 0 | 0.4 | 0 | 0.3 | 0 |
| claude-4-sonnet* | 1 | 1 | 0.4 | 0.1 | 0.3 | 0.6 | 0.2 | 0.5 | 0.2 | 0.2 | 0.1 | 0.9 | 0.7 | 0.7 | 0 |
| claude-4-sonnet | 0.1 | 1 | 0.4 | 0.4 | 0.2 | 0.4 | 0.1 | 0.5 | 0.1 | 0.4 | 0.1 | 0.6 | 0 | 0 | 0 |
| claude-3.7-sonnet* | 1 | 1 | 0.7 | 0.1 | 0.4 | 0.7 | 0.2 | 0.5 | 0 | 0.5 | 0.3 | 1 | 0.2 | 0 | 0 |
| claude-3.7-sonnet | 0.6 | 1 | 0.4 | 0.4 | 0.2 | 0.4 | 0.2 | 0.5 | 0.2 | 0.7 | 0.1 | 0.5 | 0 | 0 | 0 |
| claude-3.5-sonnet | 0 | 1 | 0.4 | 0.2 | 0.2 | 0.3 | 0.3 | 0.5 | 0 | 0.4 | 0.2 | 0.3 | 0 | 0 | 0 |

Table 16: Detailed model performance on PSI task in the ORACLE benchmark 10@1.

| Models | Easy Black-Box in PSI | | | | | | | | | | | | | Hard Black-Box in PSI | | | | | | | |
|---|---|---|---|---|---|---|---|---|---|---|---|---|---|---|---|---|---|---|---|---|---|
| | 1 | 2 | 3 | 4 | 5 | 6 | 7 | 8 | 9 | 10 | 11 | 12 | 13 | 1 | 2 | 3 | 4 | 5 | 6 | 7 | 8 |
| qwq-plus | 1 | 1 | 0.33 | 0 | 1 | 0 | 0.17 | 0 | 0 | 0.17 | 1 | 1 | 0.67 | 0 | 0 | 0.33 | 0 | 0 | 0 | 0.67 | 0 |
| qwen-plus* | 1 | 0.83 | 0 | 0 | 1 | 0 | 0.33 | 0 | 0.17 | 0.17 | 1 | 1 | 1 | 0.17 | 0.17 | 0 | 0.33 | 0 | 0 | 0 | 0.33 |
| qwen3-32b* | 1 | 0 | 0.17 | 0 | 1 | 0 | 0 | 0 | 0 | 0 | 1 | 1 | 0.33 | 0 | 0.17 | 0 | 0.33 | 0 | 0 | 0 | 0 |
| qwen3-235b* | 1 | 0.67 | 0.33 | 0 | 1 | 0 | 0 | 0.17 | 0 | 0 | 1 | 1 | 1 | 0 | 0.5 | 0 | 0.5 | 0 | 0 | 0 | 0.33 |
| o4-mini | 1 | 0.33 | 0.33 | 1 | 1 | 0.17 | 0.33 | 0 | 0.83 | 1 | 0.83 | 1 | 0.17 | 0 | 0.5 | 0.67 | 0.17 | 0 | 0 | 0.67 | 0 |
| o3-mini | 0.17 | 0 | 0.17 | 0 | 1 | 0.17 | 0.33 | 0 | 0 | 0.17 | 1 | 1 | 0.17 | 0 | 0.17 | 0.5 | 0.5 | 0 | 0.17 | 0 | 0.67 |
| o3 | 1 | 0.83 | 0.33 | 0 | 1 | 0.17 | 0.5 | 0.67 | 0.33 | 0.17 | 1 | 1 | 0.83 | 0.5 | 0.67 | 0.5 | 0.67 | 0 | 0.17 | 0.17 | 0.83 |
| o1 | 0.17 | 0.5 | 0.17 | 0.17 | 1 | 0.17 | 0.33 | 0 | 0 | 0 | 0.83 | 1 | 0.17 | 0 | 0.5 | 0.5 | 0.33 | 0 | 0.17 | 0 | 0.17 |
| gemini-2.5-pro | 1 | 1 | 0.33 | 0 | 1 | 0.17 | 0.17 | 0.17 | 0.17 | 0.17 | 1 | 1 | 0.83 | 0.33 | 0.83 | 0.5 | 0.67 | 0 | 0.33 | 0 | 0.67 |
| gemini-2.5-flash* | 0.33 | 0.67 | 0.17 | 0 | 0.83 | 0.17 | 0.17 | 0 | 0 | 0.17 | 1 | 1 | 0.83 | 0.17 | 0.67 | 0.33 | 0.67 | 0 | 0.17 | 0 | 0.5 |
| gemini-2.5-flash | 0 | 0.17 | 0 | 0.17 | 0.33 | 0 | 0.17 | 0.17 | 0 | 0.17 | 1 | 1 | 0.5 | 0.17 | 0.67 | 0.33 | 0.17 | 0 | 0 | 0 | 0.17 |
| gemini-2.0-flash | 0.33 | 0 | 0.17 | 0 | 0.17 | 0 | 0.17 | 0 | 0 | 0 | 0 | 0 | 0 | 0 | 0 | 0 | 0.5 | 0 | 0 | 0 | 0 |
| deepseek-r1 | 0.83 | 1 | 0 | 0 | 1 | 0 | 0 | 0.33 | 0.67 | 0.17 | 1 | 1 | 1 | 0 | 0 | 0.5 | 0.17 | 0 | 0 | 0.33 | 0.33 |
| deepseek-v3 | 0.17 | 0 | 0 | 0.33 | 0 | 0 | 0 | 0 | 0 | 0 | 1 | 1 | 0 | 0 | 0.17 | 0 | 0 | 0 | 0 | 0 | 0 |
| claude-4-sonnet* | 0.83 | 0.33 | 0 | 0 | 0 | 1 | 0.33 | 0.17 | 0.17 | 0.17 | 1 | 1 | 0.67 | 0 | 0.17 | 0.33 | 0.33 | 0 | 0 | 0 | 0.17 |
| claude-4-sonnet | 0.17 | 0 | 0 | 0 | 0.17 | 0 | 0 | 0 | 0 | 0 | 0.83 | 0.83 | 0 | 0 | 0 | 0.17 | 0 | 0 | 0 | 0 | 0 |
| claude-3.7-sonnet* | 0.83 | 0.33 | 0.17 | 0 | 1 | 0.17 | 0.33 | 0.33 | 0.17 | 0.17 | 1 | 1 | 0.83 | 0.17 | 0.67 | 0.5 | 0.67 | 0 | 0.17 | 0 | 0.67 |
| claude-3.7-sonnet | 0.33 | 0.17 | 0 | 0.17 | 0.33 | 0.17 | 0.17 | 0.17 | 0.17 | 0.17 | 1 | 1 | 0.5 | 0 | 0.17 | 0 | 0.33 | 0 | 0 | 0 | 0.33 |
| claude-3.5-sonnet | 0.33 | 0.17 | 0 | 0.17 | 0.33 | 0 | 0 | 0 | 0 | 0 | 0.83 | 1 | 0.33 | 0 | 0.33 | 0 | 0.33 | 0 | 0 | 0 | 0.33 |

Table 17: Detailed model performance on PSI task in the ORACLE benchmark 20@2.

| Models | Easy Black-Box in PSI | | | | | | | | | | | | | Hard Black-Box in PSI | | | | | | | |
|---|---|---|---|---|---|---|---|---|---|---|---|---|---|---|---|---|---|---|---|---|---|
| | 1 | 2 | 3 | 4 | 5 | 6 | 7 | 8 | 9 | 10 | 11 | 12 | 13 | 1 | 2 | 3 | 4 | 5 | 6 | 7 | 8 |
| qwq-plus | 0.5 | 1 | 1 | 0 | 0.83 | 0 | 0.33 | 0 | 0 | 0.17 | 0.83 | 1 | 0.67 | 0 | 0 | 0 | 0.67 | 0 | 0 | 0.17 | 0.33 |
| qwen-plus* | 0.17 | 0 | 0 | 0.17 | 1 | 0 | 0.67 | 0.67 | 0 | 0.5 | 1 | 1 | 0.5 | 0.17 | 0.5 | 0.33 | 0.33 | 0 | 0 | 0 | 0.17 |
| qwen3-32b* | 0.67 | 0.5 | 0 | 0.17 | 1 | 0 | 0.17 | 0 | 0 | 0 | 1 | 1 | 0.33 | 0 | 0 | 0.33 | 0.33 | 0 | 0 | 0 | 0.67 |
| qwen3-235b* | 0.83 | 0.17 | 1 | 0.17 | 1 | 0 | 0.17 | 0 | 0 | 0.5 | 1 | 1 | 0.33 | 0.17 | 0.17 | 0.5 | 0.67 | 0 | 0 | 0 | 0.33 |
| o4-mini | 0.83 | 1 | 0.17 | 0 | 1 | 0 | 0.5 | 0.83 | 0.17 | 0.83 | 1 | 1 | 1 | 0 | 0.83 | 0.83 | 0.67 | 0 | 0 | 0 | 0.5 |
| o3-mini | 0.67 | 0 | 0.17 | 0.17 | 1 | 0 | 0 | 0.5 | 0.33 | 0.33 | 1 | 1 | 0.17 | 0 | 1 | 0.17 | 0.67 | 0 | 0 | 0 | 0.5 |
| o3 | 1 | 1 | 0.17 | 0.17 | 1 | 0 | 0.5 | 1 | 0.5 | 0.33 | 1 | 1 | 0.83 | 0 | 0.83 | 0.67 | 0.67 | 0 | 0 | 0 | 0.83 |
| o1 | 0.33 | 0.83 | 0.17 | 0.17 | 1 | 0.17 | 0.5 | 0.17 | 0.17 | 0 | 1 | 1 | 0 | 0 | 0.17 | 0.5 | 0.33 | 0 | 0 | 0 | 0.67 |
| gemini-2.5-pro | 1 | 1 | 0.17 | 0.17 | 1 | 0 | 0.33 | 0.5 | 0.33 | 1 | 1 | 1 | 1 | 0 | 0 | 0.5 | 0.83 | 0 | 0.33 | 0 | 0.83 |
| gemini-2.5-flash* | 0.83 | 0.17 | 0.17 | 0.17 | 0.5 | 0.17 | 0.17 | 0.17 | 0 | 0.17 | 1 | 1 | 0.17 | 0 | 0 | 0.33 | 0.83 | 0 | 0 | 0 | 0.17 |
| gemini-2.5-flash | 0.5 | 0.33 | 0.17 | 0.5 | 0.17 | 0.17 | 0.17 | 0 | 0 | 0.17 | 1 | 1 | 0.5 | 0.17 | 0.5 | 0.33 | 0.67 | 0 | 0 | 0 | 0.17 |
| gemini-2.0-flash | 0 | 0 | 0 | 0 | 0 | 0 | 0 | 0 | 0 | 0 | 0 | 0 | 0 | 0 | 0 | 0 | 0.17 | 0 | 0 | 0 | 0 |
| deepseek-r1 | 1 | 1 | 0.17 | 0 | 1 | 0 | 0.17 | 0 | 0.33 | 0 | 0.83 | 1 | 1 | 0.5 | 0.67 | 0 | 0.67 | 0 | 0 | 0 | 0.67 |
| deepseek-v3 | 0.17 | 0 | 0 | 0 | 0.17 | 0 | 0.17 | 0.17 | 0 | 0.17 | 0.83 | 1 | 0 | 0 | 0.17 | 0.17 | 0.17 | 0 | 0 | 0 | 0 |
| claude-4-sonnet* | 1 | 0.5 | 0.33 | 0.17 | 1 | 0 | 0.33 | 0.33 | 0.17 | 0.33 | 1 | 1 | 0.83 | 0 | 0.5 | 0.33 | 0.67 | 0 | 0 | 0 | 0.67 |
| claude-4-sonnet | 0.33 | 0 | 0.17 | 0.17 | 0.17 | 0.17 | 0.17 | 0 | 0 | 0.17 | 1 | 1 | 0.33 | 0 | 0 | 0 | 0.17 | 0 | 0 | 0 | 0.17 |
| claude-3.7-sonnet* | 1 | 0.67 | 0.17 | 0.17 | 1 | 0.17 | 0.33 | 0.5 | 0.17 | 0.17 | 1 | 1 | 1 | 0.33 | 0.67 | 0.5 | 0.83 | 0 | 0 | 0 | 0.67 |
| claude-3.7-sonnet | 0.33 | 0.17 | 0.17 | 0.17 | 0.5 | 0.17 | 0.17 | 0.17 | 0.17 | 0.17 | 0.83 | 1 | 0.5 | 0.33 | 0 | 0.17 | 0.33 | 0 | 0 | 0 | 0.17 |
| claude-3.5-sonnet | 0.33 | 0.33 | 0.17 | 0.17 | 0.5 | 0.17 | 0.33 | 0 | 0.17 | 0 | 0.83 | 1 | 0.5 | 0.17 | 0 | 0 | 0.5 | 0 | 0 | 0 | 0.17 |

Table 18: Detailed model performance on ERI task in the ORACLE benchmark 10@1.

| Models | Easy Black-Box in ERI | | | | | | | | | | | Hard Black-Box in ERI | | | | | | | | |
|---|---|---|---|---|---|---|---|---|---|---|---|---|---|---|---|---|---|---|---|---|
| | 1 | 2 | 3 | 4 | 5 | 6 | 7 | 8 | 9 | 10 | 11 | 1 | 2 | 3 | 4 | 5 | 6 | 7 | 8 | 9 |
| qwq-plus | 0.13 | 0 | 0.13 | 0 | 0 | 0 | 0 | 0 | 0 | 0 | 0.13 | 0 | 0 | 0 | 0 | 0 | 0 | 0 | 0 | 0 |
| qwen-plus* | 0.13 | 0 | 0.13 | 0 | 0 | 0 | 0 | 0 | 0.13 | 0 | 0 | 0 | 0 | 0 | 0 | 0 | 0 | 0 | 0 | 0 |
| qwen3-32b* | 0 | 0.13 | 0.38 | 0 | 0 | 0 | 0 | 0 | 0 | 0.13 | 0.13 | 0 | 0 | 0 | 0 | 0 | 0 | 0 | 0 | 0 |
| qwen3-235b* | 0.25 | 0 | 0 | 0 | 0 | 0 | 0.13 | 0.13 | 0 | 0 | 0 | 0 | 0 | 0 | 0 | 0 | 0 | 0 | 0 | 0 |
| o4-mini | 0.25 | 0.25 | 0.75 | 0 | 0 | 1 | 0.75 | 1 | 0.75 | 1 | 1 | 0 | 0 | 0 | 0 | 0 | 0 | 1 | 0.5 | 0 |
| o3-mini | 0.25 | 0 | 0.25 | 0.38 | 0 | 0.88 | 0.13 | 0.13 | 0.75 | 1 | 1 | 0 | 0 | 0 | 0 | 0 | 0 | 1 | 0 | 0 |
| o3 | 0.13 | 0 | 1 | 0.38 | 0 | 0 | 0.38 | 1 | 1 | 0.75 | 1 | 0 | 0 | 0 | 0 | 0 | 0 | 0 | 0.5 | 0 |
| o1 | 0 | 0.38 | 0 | 0 | 0 | 0.75 | 0.25 | 0 | 0 | 0 | 0 | 0 | 0 | 0 | 0 | 0 | 0 | 0 | 0 | 0 |
| gemini-2.5-pro | 0.875 | 1 | 0.25 | 0.13 | 0 | 0 | 0.13 | 0.5 | 0.13 | 1 | 0.38 | 0 | 0 | 0.13 | 0.13 | 0.5 | 0 | 0 | 0 | 0 |
| gemini-2.5-flash* | 0.25 | 0 | 0 | 0 | 0 | 0 | 0 | 0 | 0.38 | 0.13 | 0 | 0 | 0 | 0 | 0 | 0 | 0 | 0 | 0 | 0 |
| gemini-2.5-flash | 0.13 | 0 | 0 | 0 | 0 | 0 | 0.13 | 0.13 | 0 | 0.38 | 0 | 0 | 0 | 0 | 0 | 0 | 0 | 0 | 0 | 0 |
| gemini-2.0-flash | 0.25 | 0 | 0 | 0 | 0 | 0 | 0 | 0 | 0 | 0.25 | 0 | 0 | 0 | 0 | 0 | 0.13 | 0 | 0 | 0 | 0 |
| deepseek-r1 | 0 | 0 | 0 | 0 | 0 | 0.25 | 0.13 | 0 | 0 | 0 | 0 | 0 | 0 | 0 | 0 | 0 | 0 | 0 | 0 | 0 |
| deepseek-v3 | 0.25 | 0 | 0 | 0 | 0 | 0 | 0.13 | 0 | 0 | 0 | 0 | 0 | 0 | 0 | 0 | 0 | 0 | 0 | 0 | 0 |
| claude-4-sonnet* | 0.25 | 0.13 | 0.38 | 0.75 | 0 | 0 | 0.13 | 0 | 0.13 | 0.13 | 0.75 | 0 | 0 | 0 | 0.13 | 0 | 0 | 0 | 0 | 0 |
| claude-4-sonnet | 0.25 | 0 | 0 | 0 | 0 | 0 | 0.13 | 0.13 | 0 | 0 | 0 | 0 | 0 | 0 | 0 | 0 | 0 | 0 | 0 | 0 |
| claude-3.7-sonnet* | 0.25 | 0 | 0.38 | 0.63 | 0 | 0 | 0.13 | 0 | 0.38 | 0.13 | 0.13 | 0 | 0 | 0 | 0 | 0 | 0 | 0 | 0 | 0 |
| claude-3.7-sonnet | 0.25 | 0 | 0 | 0 | 0 | 0 | 0.13 | 0 | 0.25 | 0.63 | 0 | 0 | 0 | 0 | 0 | 0 | 0 | 0 | 0 | 0 |
| claude-3.5-sonnet | 0.25 | 0 | 0 | 0 | 0 | 0 | 0.13 | 0 | 0.38 | 0.13 | 0 | 0 | 0 | 0 | 0 | 0 | 0 | 0 | 0 | 0 |

Table 19: Detailed model performance on ERI task in the ORACLE benchmark 20@2.

| Models | Easy Black-Box in ERI | | | | | | | | | | | Hard Black-Box in ERI | | | | | | | | |
|---|---|---|---|---|---|---|---|---|---|---|---|---|---|---|---|---|---|---|---|---|
| | 1 | 2 | 3 | 4 | 5 | 6 | 7 | 8 | 9 | 10 | 11 | 1 | 2 | 3 | 4 | 5 | 6 | 7 | 8 | 9 |
| qwq-plus | 0.13 | 0 | 0.13 | 0 | 0 | 0 | 0 | 0 | 0 | 0 | 0.13 | 0 | 0 | 0 | 0 | 0 | 0 | 0 | 0 | 0 |
| qwen-plus* | 0.13 | 0 | 0.13 | 0 | 0 | 0 | 0 | 0 | 0.13 | 0 | 0 | 0 | 0 | 0 | 0 | 0 | 0 | 0 | 0 | 0 |
| qwen3-32b* | 0.13 | 0 | 0.13 | 0 | 0 | 0 | 0 | 0 | 0 | 0.13 | 0 | 0 | 0 | 0 | 0 | 0 | 0 | 0 | 0 | 0 |
| qwen3-235b* | 0.25 | 0 | 0.13 | 0 | 0 | 0 | 0.13 | 0.13 | 0 | 0 | 0 | 0 | 0 | 0 | 0 | 0 | 0 | 0 | 0 | 0 |
| o4-mini | 0.75 | 0.5 | 0.25 | 1 | 0 | 1 | 0.75 | 0.63 | 0.88 | 0.25 | 1 | 0.13 | 0 | 0.13 | 0.13 | 1 | 0 | 0 | 0 | 0 |
| o3-mini | 0 | 0.38 | 0.25 | 0.88 | 0 | 0.25 | 0.75 | 0.75 | 0 | 0 | 1 | 0 | 0 | 0.13 | 0 | 0.75 | 0 | 0 | 0 | 0 |
| o3 | 0.5 | 0.13 | 0.25 | 0 | 0 | 0.38 | 0.75 | 0.88 | 0 | 0 | 0 | 0 | 0 | 0 | 0.38 | 1 | 0 | 0 | 0 | 0 |
| o1 | 0 | 0 | 0 | 0.5 | 0 | 0 | 0.13 | 0 | 0 | 1 | 0.75 | 0 | 0 | 0 | 0 | 0 | 0 | 0.63 | 0 | 0 |
| gemini-2.5-pro | 0.88 | 0 | 0.13 | 0.13 | 0 | 0.38 | 0.25 | 0.13 | 0 | 0 | 0 | 0 | 0 | 0.13 | 0 | 0 | 0 | 0 | 0 | 0 |
| gemini-2.5-flash* | 0.13 | 0 | 0.25 | 0 | 0 | 0 | 0 | 0 | 0 | 0 | 0 | 0 | 0 | 0 | 0 | 0 | 0 | 0 | 0 | 0 |
| gemini-2.5-flash | 0 | 0 | 0.25 | 0 | 0 | 0 | 0 | 0 | 0 | 0 | 0.13 | 0 | 0 | 0 | 0 | 0 | 0 | 0 | 0 | 0 |
| gemini-2.0-flash | 0 | 0 | 0 | 0 | 0 | 0 | 0.13 | 0 | 0 | 0.13 | 0 | 0 | 0 | 0 | 0 | 0 | 0 | 0 | 0 | 0 |
| deepseek-r1 | 0 | 0 | 0 | 0 | 0 | 0 | 0 | 0 | 0.13 | 0 | 0 | 0 | 0 | 0 | 0 | 0 | 0 | 0 | 0 | 0 |
| deepseek-v3 | 0 | 0 | 0 | 0 | 0 | 0 | 0.13 | 0 | 0 | 0 | 0 | 0 | 0 | 0 | 0 | 0 | 0 | 0 | 0 | 0 |
| claude-4-sonnet* | 0 | 0 | 0.13 | 0.13 | 0 | 0 | 0.25 | 0.13 | 0 | 0 | 0 | 0 | 0 | 0 | 0 | 0 | 0 | 0 | 0 | 0 |
| claude-4-sonnet | 0.13 | 0 | 0.25 | 0 | 0 | 0 | 0 | 0 | 0.13 | 0 | 0 | 0 | 0 | 0 | 0 | 0 | 0 | 0 | 0 | 0 |
| claude-3.7-sonnet* | 0.25 | 0 | 0.13 | 0.13 | 0 | 0 | 0.13 | 0.13 | 0.75 | 0.75 | 0.25 | 0.13 | 0.13 | 0 | 0 | 0 | 0 | 0.25 | 0 | 0 |
| claude-3.7-sonnet | 0 | 0 | 0.25 | 0 | 0 | 0 | 0 | 0 | 0.13 | 0 | 0.63 | 0 | 0 | 0 | 0 | 0 | 0 | 0 | 0 | 0 |
| claude-3.5-sonnet | 0 | 0 | 0.25 | 0 | 0 | 0 | 0 | 0 | 0.13 | 0 | 0.38 | 0 | 0 | 0 | 0 | 0 | 0 | 0 | 0 | 0 |

Table 20: Detailed model performance on IPI task in the ORACLE benchmark 10@1.

| Models | Easy Black-Box in IPI | | | | | Hard Black-Box in IPI | | | | | |
|---|---|---|---|---|---|---|---|---|---|---|---|
| | 1 | 2 | 3 | 4 | 5 | 1 | 2 | 3 | 4 | 5 | 6 |
| qwq-plus | 0.17 | 0.83 | 0.67 | 0 | 0.5 | 0 | 0 | 0 | 0 | 0 | 0 |
| qwen-plus* | 0.67 | 1 | 0.67 | 0.5 | 0.67 | 0 | 0.83 | 0 | 0.17 | 0 | 0 |
| qwen3-32b* | 0.17 | 1 | 0.83 | 0 | 0.5 | 0 | 0 | 0 | 0.33 | 0 | 0 |
| qwen3-235b* | 0.17 | 0.83 | 0.5 | 0.5 | 0.33 | 0.17 | 0.5 | 0 | 0.17 | 0 | 0 |
| o4-mini | 0.17 | 1 | 0.83 | 1 | 1 | 0 | 1 | 0.17 | 0.83 | 0 | 0 |
| o3-mini | 0 | 0.67 | 0.67 | 0.33 | 0.67 | 0 | 0.17 | 0 | 0.67 | 0 | 0 |
| o3 | 0.17 | 1 | 1 | 1 | 1 | 0.33 | 1 | 0.17 | 0.67 | 0 | 0 |
| o1 | 0.17 | 0.67 | 0.67 | 0.17 | 0.33 | 0 | 0.5 | 0 | 0.83 | 0 | 0 |
| gemini-2.5-pro | 0.33 | 0.83 | 0.83 | 0.83 | 0.83 | 0.5 | 1 | 0.17 | 0.33 | 0 | 0 |
| gemini-2.5-flash* | 0.17 | 0.5 | 0.5 | 0 | 0.33 | 0 | 0.67 | 0 | 0.5 | 0 | 0 |
| gemini-2.5-flash | 0.17 | 0.33 | 0.17 | 0 | 0 | 0 | 0 | 0 | 0.17 | 0 | 0 |
| gemini-2.0-flash | 0.17 | 0.33 | 0 | 0 | 0 | 0 | 0 | 0 | 0.17 | 0 | 0 |
| deepseek-r1 | 0.33 | 0.67 | 1 | 0.33 | 0.33 | 0 | 0.67 | 0.17 | 0.33 | 0 | 0 |
| deepseek-v3 | 0.17 | 0.67 | 0 | 0 | 0 | 0 | 0 | 0 | 0 | 0 | 0 |
| claude-4-sonnet* | 0.33 | 0.83 | 0.5 | 0.83 | 0.17 | 0 | 1 | 0 | 0.83 | 0 | 0 |
| claude-4-sonnet | 0.67 | 0.83 | 0.67 | 0 | 0 | 0 | 0 | 0 | 0.83 | 0 | 0 |
| claude-3.7-sonnet* | 0.33 | 0.67 | 0.17 | 0.83 | 0 | 0 | 0.67 | 0 | 1 | 0 | 0 |
| claude-3.7-sonnet | 0.5 | 0.83 | 0.17 | 0 | 0 | 0 | 0 | 0 | 0.33 | 0 | 0 |
| claude-3.5-sonnet | 0.17 | 0.83 | 0.33 | 0 | 0 | 0 | 0 | 0 | 0.33 | 0 | 0 |

Table 21: Detailed model performance on IPI task in the ORACLE benchmark 20@2.

| Models | Easy Black-Box in IPI | | | | | Hard Black-Box in IPI | | | | | |
|---|---|---|---|---|---|---|---|---|---|---|---|
| | 1 | 2 | 3 | 4 | 5 | 1 | 2 | 3 | 4 | 5 | 6 |
| qwq-plus | 1 | 1 | 0 | 0.67 | 0.33 | 0 | 0 | 0 | 0.17 | 0 | 0 |
| qwen-plus* | 1 | 1 | 0.5 | 0.83 | 0.5 | 0.5 | 0.67 | 0 | 0.17 | 0 | 0 |
| qwen3-32b* | 0.83 | 1 | 0.83 | 1 | 0.5 | 0 | 0.17 | 0 | 0.67 | 0 | 0 |
| qwen3-235b* | 1 | 1 | 0.5 | 1 | 0.33 | 0.5 | 0.33 | 0 | 0.17 | 0 | 0 |
| o4-mini | 1 | 0.83 | 0.83 | 1 | 0.67 | 0.5 | 1 | 0.33 | 0.83 | 0 | 0 |
| o3-mini | 1 | 1 | 0.67 | 1 | 0.17 | 0 | 0.5 | 0 | 0.5 | 0 | 0 |
| o3 | 1 | 1 | 1 | 1 | 0.83 | 0.83 | 1 | 1 | 1 | 0.17 | 0 |
| o1 | 1 | 1 | 0.5 | 0.83 | 0.83 | 0.5 | 0.5 | 0 | 0.83 | 0 | 0 |
| gemini-2.5-pro | 1 | 1 | 1 | 1 | 0.83 | 1 | 1 | 0.17 | 0.5 | 0 | 0 |
| gemini-2.5-flash* | 0.83 | 1 | 0.67 | 1 | 0.5 | 0.5 | 0.5 | 0.17 | 1 | 0 | 0 |
| gemini-2.5-flash | 0.67 | 0.17 | 0 | 0 | 0.17 | 0 | 0 | 0 | 0.33 | 0 | 0 |
| gemini-2.0-flash | 0.33 | 0.17 | 0 | 0 | 0.17 | 0 | 0 | 0 | 0.17 | 0 | 0 |
| deepseek-r1 | 1 | 1 | 0.33 | 0.5 | 0.33 | 0.83 | 0.67 | 0 | 0.83 | 0.17 | 0 |
| deepseek-v3 | 0.83 | 0.67 | 0 | 0 | 0.5 | 0 | 0 | 0 | 0 | 0 | 0 |
| claude-4-sonnet* | 1 | 1 | 0.83 | 0.33 | 1 | 0.5 | 1 | 0 | 1 | 0 | 0 |
| claude-4-sonnet | 1 | 1 | 0 | 0 | 1 | 0 | 0 | 0 | 1 | 0 | 0 |
| claude-3.7-sonnet* | 1 | 1 | 0.67 | 0.5 | 0.83 | 0.17 | 0.83 | 0 | 1 | 0.33 | 0 |
| claude-3.7-sonnet | 0.83 | 0.83 | 0 | 0.33 | 0.83 | 0 | 0.17 | 0 | 0.83 | 0 | 0 |
| claude-3.5-sonnet | 1 | 1 | 0.17 | 0.17 | 0.67 | 0 | 0 | 0 | 0.5 | 0 | 0 |

Table 22: Detailed model performance on GSI task in the ORACLE benchmark 1@1.

| Models | Easy Black-Box in GSI | | | | | | Hard Black-Box in GSI | | | | | |
|---|---|---|---|---|---|---|---|---|---|---|---|---|
| | 1 | 2 | 3 | 4 | 5 | 6 | 1 | 2 | 3 | 4 | 5 | 6 |
| qwq-plus | 0.27 | 0.65 | 0 | 0.39 | 0.11 | 0.27 | 0.33 | 0.68 | 0 | 0.39 | 0 | 0 |
| qwen-plus* | 0.75 | 0.85 | 0.82 | 0.20 | 0.74 | 0.16 | 0.34 | 0.58 | 0.25 | 0.42 | 0 | 0.36 |
| qwen3-32b* | 0.32 | 0.44 | 0.31 | 0.45 | 0.18 | 0.16 | 0.15 | 0.14 | 0 | 0.52 | 0 | 0.19 |
| qwen3-235b* | 0.08 | 0.69 | 0.50 | 0.48 | 0.28 | 0.14 | 0.47 | 0.29 | 0 | 0.47 | 0 | 0.34 |
| o4-mini | 0.76 | 0.84 | 1 | 0.58 | 0.26 | 0.32 | 0.41 | 0.45 | 0.42 | 0.38 | 0.1 | 0.22 |
| o3-mini | 0.82 | 0.82 | 0.83 | 0.36 | 0 | 0.46 | 0.33 | 0.54 | 0.23 | 0.7 | 0.11 | 0 |
| o3 | 0.94 | 0.91 | 0.93 | 0.71 | 0.79 | 0.49 | 0.49 | 0.33 | 0.70 | 0.4 | 0.21 | 0.51 |
| o1 | 0.41 | 0.70 | 0.71 | 0.97 | 0.09 | 0.19 | 0.44 | 0.33 | 0 | 0.46 | 0.20 | 0.11 |
| gemini-2.5-pro | 0.42 | 0.68 | 0.70 | 0.34 | 0.93 | 0.63 | 0.39 | 0.52 | 0.64 | 0.60 | 0.2 | 0.46 |
| gemini-2.5-flash* | 0.38 | 0.36 | 0.48 | 0.33 | 0.58 | 0.39 | 0.31 | 0.13 | 0.6 | 0.33 | 0.38 | 0.44 |
| gemini-2.5-flash | 0.17 | 0.35 | 0 | 0.24 | 0.07 | 0.02 | 0 | 0.25 | 0.42 | 0.66 | 0 | 0 |
| gemini-2.0-flash | 0 | 0.06 | 0.28 | 0.32 | 0.02 | 0.30 | 0 | 0.15 | 0 | 1 | 0 | 0 |
| deepseek-r1 | 0.49 | 0.58 | 0.79 | 0.42 | 0.50 | 0.36 | 0.47 | 0.25 | 0.19 | 0.48 | 0.05 | 0.08 |
| deepseek-v3 | 0.06 | 0.03 | 0.29 | 0.25 | 0.30 | 0.12 | 0.16 | 0.29 | 0.16 | 0 | 0 | 0 |
| claude-4-sonnet* | 0.24 | 0.28 | 0.40 | 0.55 | 0.58 | 0.43 | 0.31 | 0.21 | 0.38 | 0.13 | 0.22 | |
| claude-4-sonnet | 0.43 | 0.14 | 0.24 | 0.67 | 0.17 | 0.30 | 0.24 | 0.41 | 0 | 0.27 | 0 | 0.12 |
| claude-3.7-sonnet* | 0.38 | 0.46 | 0.47 | 0.53 | 0.30 | 0.26 | 0.41 | 0.29 | 0 | 0.3 | 0.09 | 0.44 |
| claude-3.7-sonnet | 0.35 | 0 | 0.73 | 0.88 | 0.02 | 0.52 | 0.04 | 0.33 | 0.08 | 0.14 | 0 | 0.3 |
| claude-3.5-sonnet | 0.22 | 0.17 | 1 | 0.83 | 0.06 | 0.03 | 0.19 | 0.29 | 0.13 | 0.25 | 0 | 0.41 |

Table 23: Detailed model performance on GSI task in the ORACLE benchmark 2@1.

| Models | Easy Black-Box in GSI | | | | | | Hard Black-Box in GSI | | | | | |
|---|---|---|---|---|---|---|---|---|---|---|---|---|
| | 1 | 2 | 3 | 4 | 5 | 6 | 1 | 2 | 3 | 4 | 5 | 6 |
| qwq-plus | 0.45 | 0.61 | 0 | 0.26 | 0.13 | 0.11 | 0.43 | 0.34 | 0 | 0.33 | 0 | 0 |
| qwen-plus* | 0.64 | 0.70 | 0.85 | 0.56 | 0.69 | 0.32 | 0.38 | 0.68 | 0.43 | 0.80 | 0.33 | 0.46 |
| qwen3-32b* | 0.63 | 0.68 | 0 | 0.53 | 0.30 | 0.20 | 0.25 | 0.42 | 0 | 0 | 0.13 | 0.19 |
| qwen3-235b* | 0.44 | 0.36 | 0.68 | 0.43 | 0.14 | 0.26 | 0.47 | 0.68 | 0.08 | 0.90 | 0.05 | 0.12 |
| o4-mini | 0.72 | 0.76 | 0.77 | 0.76 | 0.21 | 0.10 | 0.37 | 0.31 | 0.21 | 0.27 | 0 | 0.23 |
| o3-mini | 0.5 | 0.62 | 0.88 | 0.59 | 0.15 | 0.22 | 0.49 | 1 | 0.13 | 0.26 | 0.06 | 0.12 |
| o3 | 1 | 0.97 | 0.76 | 0.79 | 0.82 | 0.62 | 0.48 | 0.61 | 0.44 | 0.45 | 0.54 | 0.76 |
| o1 | 0.18 | 0.68 | 0.89 | 1 | 0.05 | 0.20 | 0.58 | 0.74 | 0 | 0.29 | 0 | 0.44 |
| gemini-2.5-pro | 0.69 | 0.5 | 0.58 | 0.5 | 0.85 | 0.53 | 0.82 | 0.62 | 0.13 | 0.52 | 0.05 | 0.33 |
| gemini-2.5-flash* | 0.71 | 0.5 | 0.82 | 0.75 | 0.10 | 0.29 | 0.14 | 0.46 | 0 | 0.75 | 0.19 | 0.07 |
| gemini-2.5-flash | 0.24 | 0.81 | 0 | 0.57 | 0.09 | 0.18 | 0 | 0.15 | 0.06 | 0.71 | 0.15 | 0.10 |
| gemini-2.0-flash | 0 | 0.27 | 0.10 | 0.39 | 0.04 | 0.27 | 0 | 0.2 | 0.17 | 0.62 | 0 | 0 |
| deepseek-r1 | 0.63 | 0.51 | 0.48 | 0.40 | 0.31 | 0.41 | 0.50 | 0.37 | 0.08 | 0.48 | 0 | 0.05 |
| deepseek-v3 | 0.04 | 0.5 | 0.48 | 0 | 0.15 | 0.07 | 0.26 | 0.23 | 0.13 | 0.25 | 0 | 0 |
| claude-4-sonnet* | 0.88 | 0.78 | 0.27 | 0.81 | 0.56 | 0.48 | 0.61 | 0.45 | 0.25 | 0.12 | 0.13 | 0.62 |
| claude-4-sonnet | 0.49 | 0.17 | 0.55 | 0.78 | 0.36 | 0.30 | 0.18 | 0.28 | 0.3 | 0.26 | 0 | 0.2 |
| claude-3.7-sonnet* | 0.35 | 0.56 | 0.46 | 0.79 | 0.42 | 0.39 | 0.56 | 0.5 | 0.19 | 0.29 | 0 | 0.4 |
| claude-3.7-sonnet | 0.33 | 0.28 | 0.73 | 0.78 | 0 | 0.37 | 0.14 | 0.18 | 0.1 | 0.14 | 0 | 0.51 |
| claude-3.5-sonnet | 0.13 | 0.15 | 0.63 | 0.92 | 0.08 | 0.04 | 0.14 | 0.18 | 0 | 0 | 0 | 0.21 |

