# OpenReview forum: "Investigating Advanced Reasoning of Large Language Models via Black-Box Interaction"
_ICLR.cc/2026/Conference — Submitted to ICLR 2026_

### Official Review · Reviewer_NwnN · 2025-10-24

**Soundness:** 3
**Presentation:** 3
**Contribution:** 3
**Rating:** 6
**Confidence:** 4

**Summary:**

The paper presents a new evaluation framework, Black-box Interaction, aimed at assessing advanced reasoning abilities of large language models (LLMs) in interactive and verifiable settings. The authors introduce the ORACLE benchmark, consisting of six types of black-box tasks (e.g., code, physics, encryption, puzzle-solving, game strategy) where LLMs must explore, hypothesize, and refine their understanding of hidden rules through multi-round interactions. Experiments on 19 major models (OpenAI o3, Claude, Gemini, DeepSeek, Qwen3, etc.) reveal that even the strongest LLMs perform well on simple environments but struggle with adaptive planning and hypothesis revision in complex ones.

**Strengths:**

1. The ORACLE benchmark is broad and well-designed, covering multiple reasoning domains and allowing automatic evaluation.
2. The automatic generation pipeline for black-box environments offers scalability and reproducibility.

**Weaknesses:**

Conceptually, this paper shares nearly identical motivations as IDEA (https://aclanthology.org/2025.findings-acl.698/)—both examine how LLMs infer hidden rules from interaction—yet this paper does not discuss their relationship or provide comparative experiments.

**Questions:**

See Weaknesses

---

> ### Author Response · Authors · 2025-11-20
> **Rebuttal**
>
> Thanks for reviewing our paper and providing valuable feedback. We will make clarifications point by point.
>
> ### W1: Conceptually, this paper shares nearly identical motivations as IDEA. Yet this paper does not discuss their relationship or provide comparative experiments.
>
> ### A1:
> Thank you for identifying the missed related work. RULEARN (the benchmark proposed in IDEA) shares a similar idea of building an interactive environment with unknown rules. However, there still exist three noticeable differences:
>
> 1. **Fundamental Differences in Experimental Setup**: Our work formalizes the black-box interaction task into *distinct exploration and evaluation stages*. We quantify the model's understanding of the hidden rule by cleverly constructing input-output test samples during the evaluation stage. This experimental design allows us to be the first (as far as we know) to identify that LLMs cannot devise efficient and adaptive exploration strategies for refining hypotheses. In contrast, RULEARN only involves *a one-stage multi-turn interaction*, thus lacking an analysis of the reasons behind model failure.
>
> 2. **Fundamental Differences in Task Formulation**:
> The three tasks in RULEARN each construct one hidden answer based on one specific hidden rule, and the model is required to infer this hidden answer through interaction. Therefore, the three tasks can be classified as Interactive Puzzle Inference (IPI) task within the Oracle benchmark. In other tasks within the Oracle benchmark (e.g., Encryption Rule Inference (ERI)), the model must infer the *hidden rule* (e.g., the encryption rule in ERI) rather than *a hidden answer*, which is a more fundamental and challenging task. Furthermore, the black-box interaction concept we propose is highly *flexible* and *generalizable*. Its goal is to investigate whether a model can understand an unknown environment through exploration, thus accommodating a wide range of task settings (beyond the six types in the Oracle benchmark). For instance, in the Code Intent Inference (CII) task setting, the model can access the values of intermediate variables, not just the final output. In contrast, the task settings in RULEARN are singular and difficult to generalize.
>
> 3. **Fundamental Differences in Research Focus**:
> Our work focuses on building a general and scalable benchmark. Our greatest contribution lies in proposing a highly applicable black-box interaction paradigm, creating six categories of black-box tasks, and introducing an efficient black-box generation framework for expanding the benchmark. The IDEA work primarily focuses on proposing a reasoning framework that combines induction, abduction, and deduction. The RULEARN benchmark is quite primitive, relying entirely on manual generation. This benchmark is mainly used to validate the effectiveness of the proposed IDEA framework and is difficult to serve as a general benchmark for evaluating the reasoning capabilities of LLMs.
>
> We will add the above discussion to Appendix C.2 Comparison with Previous Work. Thanks again for giving a positive review. If you find our clarification helpful, we sincerely hope you may consider raising the rating. We are also very glad for further discussion.

---

> ### Author Response · Authors · 2025-11-25
>
> Dear Reviewer NwnN:
>
> Thank you for your thoughtful review. We have addressed your concerns in detail in the rebuttal and would appreciate it if you could revisit our responses. Please let us know if any concerns remain. If the rebuttal satisfactorily addresses your points, we kindly ask you to consider reflecting this in your overall rating.
>
> Best
>
> Authors

---

> ### Comment · Reviewer_NwnN · 2025-11-25
>
> Thank you for the responses. Given the overlap in contributions with IDEA, I will keep my score at 6, which I believe is a fair and reasonable assessment.

---

### Official Review · Reviewer_HrU1 · 2025-10-27

**Soundness:** 2
**Presentation:** 2
**Contribution:** 2
**Rating:** 2
**Confidence:** 4

**Summary:**

This paper address the deficiency of existing LLM reasoning evaluations in assessing integrated reasoning within interactive, unknown environments, this paper introduces a novel "black-box interaction" paradigm. Based on this paradigm, the authors constructed the ORACLE benchmark, comprising 96 black-boxes across 6 task types, utilizing an innovative automated agentic framework for generation and scalability . Experiments benchmarking 19 LLMs revealed that even leading models struggle significantly in developing efficient and adaptive exploration strategies required to uncover complex hidden rules within the black-boxes . The research provides a valuable new tool for evaluation and highlights critical shortcomings in current LLMs concerning high-level planning and strategic adaptation based on feedback .

**Strengths:**

1. The study clearly identifies and provides compelling evidence for a critical limitation of current LLMs: their inability to devise efficient and adaptive exploration strategies for refining hypotheses.
2. The paper’s motivation—to evaluate an integrated reasoning process that combines abduction, deduction, and induction—aligns more closely with the complexity of human problem-solving than assessing these reasoning modes in isolation.
3. The paper introduces the novel concept of “black-box interaction,” which offers a fresh perspective distinct from existing static benchmarks, and presents an automated agentic framework developed specifically to generate the ORACLE benchmark.

**Weaknesses:**

1.  The ORACLE benchmark, relies on only 6 task types (CII, CRI, PSI, ERI, IPI, GSI) which are primarily formal or rule-based. This narrow focus raises concerns about whether performance on these "overly regularized" tasks adequately represents the broad spectrum of "advanced reasoning" required in diverse, real-world unknown environments. Furthermore, the criteria for distinguishing between "easy" and "hard" black-boxes are not explicitly defined and the adequacy of the baseline screening test (12 turns, 10 samples total) is questionable, especially given counter-intuitive results like GPT-4o failing while Gemini-2.0-flash passed potentially undermining the benchmark's ability to accurately stratify and evaluate model capabilities.
2. The paper relies heavily on an automated agentic framework using LLMs (Coding, Test, Refinement LLMs) for black-box creation, simulation, and debugging. While innovative for scalability, this introduces concerns about the consistency, uniqueness, and accuracy of the generated black-boxes. The framework itself might inherit biases from the LLMs used, potentially generating tasks that favor certain model architectures or knowledge domains. The paper lacks a rigorous analysis of this framework's reliability, potential biases, and how they might affect the benchmark's fairness and validity.
3.  The paper compellingly identifies a key weakness: LLMs lack efficient and adaptive exploration strategies. However, the analysis primarily demonstrates *that* this deficiency exists (e.g., through performance gains analysis and the comparative feedback experiment) rather than deeply investigating *why*. It lacks exploration into contributing factors such as limitations in long-context reasoning during extended interactions, ineffective processing or utilization of feedback signals, fundamental deficits in planning capabilities, or misunderstandings of the task objectives. The absence of metrics on token/turn efficiency also hinders understanding if better models achieve success through superior strategy or simply more computation/output. Moreover, the case studies don't explicitly show CoT reasoning during feedback analysis, raising questions about whether the experimental setup might inadvertently suppress models' analytical capabilities.

**Questions:**

1.  How does the agentic framework ensure the uniqueness, correctness, and lack of inherent bias in the generated black-boxes across different generation runs or different underlying LLMs? What steps were taken to validate the framework itself?
2.  What specific, objective criteria were used to classify black-boxes as "easy" versus "hard" across the 6 different task types? How was the consistency of this difficulty scaling validated?
3.  Is the baseline test setup (12 turns, 10 samples across 3 specific black-boxes) statistically sufficient to reliably filter models, especially given seemingly counter-intuitive outcomes like GPT-4o failing and Gemini-2.0-flash passing?
4.  What is the approximate performance level of humans (e.g., computer science students, domain experts) on the ORACLE benchmark tasks under similar constraints (turn limits)?
5.  Can the observed performance *decrease* for some models/tasks when increasing turns from 10 to 20 (as hinted in Figure 8 analysis  be attributed to random fluctuation, context window limitations, error accumulation, or other factors? What trends are observed if the number of turns is increased further (e.g., to 5, 15, 30 or 50)?
6.  Was the token usage or computational cost per turn analyzed? Do higher-performing models demonstrate greater efficiency in their exploration (i.e., achieving higher accuracy with fewer tokens or less verbose queries per turn)?
7.  In the comparative experiment on adaptive exploration, is there evidence (or lack thereof) that models *attempt* to analyze feedback even if their strategy doesn't change? Could the lack of explicit prompting for step-by-step reasoning *about* the feedback in the interaction loop negatively impact their ability to adapt their strategy?
8. Beyond identifying the lack of adaptive exploration strategies, what specific experiments or analyses could pinpoint the underlying causes (e.g., failure to integrate information over long histories, inability to form complex hypotheses, poor credit assignment from feedback, misunderstanding feedback signals)?
9. Recent works, such as KORGym, have demonstrated the lack of planning capabilities in models through multi-turn environmental interactions, yet relevant discussion is missing.

---

> ### Author Response · Authors · 2025-11-20
> **Rebuttal (Part 1)**
>
> Thanks for reviewing our paper and providing valuable feedback. We will make clarifications point by point.
>
> ### W1(1): The ORACLE benchmark, relies on only 6 task types which are primarily formal or rule-based. This narrow focus raises concerns about whether performance on these "overly regularized" tasks adequately represents the broad spectrum of "advanced reasoning" required in diverse, real-world unknown environments.
>
> ### A1(1):
> All six designed tasks are an abstraction of real-world tasks:
> - Code Intent Inference (CII) models code reverse engineering or deciphering legacy code via breakpoint debugging.
> - Circuit Rule Inference (CRI) models hardware reverse engineering or integrated circuit (IC) analysis.
> - Physics System Inference (PSI) models scientific discovery and the scientific method.
> - Encryption Rule Inference (ERI) models cryptanalysis under the Chosen-Plaintext Attack (CPA) setting.
> - Interactive Puzzle Inference (IPI) models interactive problem-solving.
> - Game Strategy Inference (GSI) models opponent modeling and devising counter-strategies within game-theoretic scenarios.
>
> As to diversity, it’s easy to combine two or three of the tasks to build a "diverse" environment. However, as a benchmark, we believe that task decomposition is a clearer approach for analyzing model performance.
>
> ---
>
> ### W1(2): The criteria for distinguishing between "easy" and "hard" black-boxes are not explicitly defined.
>
> ### A1(2):
> How to define easy and hard black-box is explicitly shown in each subsection of Appendix G. The division of tasks into easy and hard is based on two points:
>
> 1. **Human understanding of black-box difficulty**. Four designers ranked the black-box difficulty of each task based on a partial order relationship. They ultimately discussed and collectively determined the sets for easy and hard tasks.
>
> 2. **The black-box's inherent difficulty**.
>
>     - (Code Intent Inference) Easy black-boxes cover common, simple algorithms (e.g., bubble sort, Fibonacci). Hard black-boxes cover complex, less common algorithms (e.g., heapsort, Sieve of Eratosthenes).
>
>     - (Circuit Rule Inference) Easy black-boxes mainly feature circuits with a sequence structure and those that implement basic functions (e.g., XOR). Hard black-boxes mainly feature circuits with a tree structure and those that implement more complex functions (e.g., ADD).
>
>     - (Physics System Inference) Easy black-boxes focus on fundamental motion involving one or two objects (e.g., projectile motion, collision). Hard black-boxes involve two or more objects or incorporate more complex mechanical systems (e.g., three-ball collision, damped simple harmonic motion).
>
>     - (Encryption Rule Inference) Easy black-boxes feature ciphers that do not change the frequency of characters in the plaintext. For hard black-boxes, the encrypted letters barely retain their original frequencies.
>
>     - (Interactive Puzzle Inference) Easy black-boxes are puzzles with simple rules and limited action space, while hard black-boxes have more complex rules and a much larger range of actions to choose from.
>
>     - (Game Strategy Inference) Easy black-boxes are typically with fixed game strategy or games with easy rules, while hard black-boxes are with dynamic game strategy or games with difficult rules.
>
> From the experimental results, the current division of easy and hard is reasonable. Almost all models perform significantly better on easy setting than on hard setting.
>
> ---
>
> ### W1(3): The adequacy of the baseline test is questionable, especially given counter-intuitive results like GPT-4o failing while Gemini-2.0-flash passed.
>
> ### A1(3):
> The baseline test was primarily designed for cost reduction: we lacked sufficient resources to test all models, necessitating a preliminary filtration process. We acknowledge that the design of the baseline test is not entirely rigorous. A model that fails the baseline test may, in fact, achieve a higher score on the Oracle benchmark than a model that passes. Nonetheless, the baseline test still provides a rough indication of model performance on the Oracle benchmark, especially considering that it only incorporates very simple black-boxes and requires three independent runs. We have supplemented the performance for gpt-4o and gpt-4.1 on the Oracle benchmark 10@1, comparing them against gemini-2.0-flash. The results are presented below.
>
> | TASK | DIFFICULTY | gpt-4o | gpt-4.1 | gemini-2.0-flash |
> | :--- | :--- | :--- | :--- | :--- |
> | CII | Easy | 49.96% | 35.84% | 38.18% |
> | CII | Hard | 21.19% | 18.81% | 14.52% |
> | CRI | Easy | 26.00% | 26.00% | 18% |
> | CRI | Hard | 13.00% | 11.00% | 18% |
> | PSI | Easy | 24.36% | 26.92% | 6.41% |
> | PSI | Hard | 12.5% | 18.75% | 6.25% |
> | ERI | Easy | 1.14% | 0% | 2.27% |
> | ERI | Hard | 0% | 0% | 0% |
> | IPI | Easy | 23.33% | 33.33% | 10.00% |
> | IPI | Hard | 2.78% | 5.56% | 2.78% |
> | GSI | Easy | 19.91% | 32.48% | 16.42% |
> | GSI | Hard | 7.41% | 12.74% | 19.71% |
> **(See next page)**

---

> ### Author Response · Authors · 2025-11-20
> **Rebuttal (Part 2)**
>
> **(Continue to Reply to W1(3))**
>
> We observe that the performance of gpt-4o and gpt-4.1 is marginally superior to gemini-2.0-flash on most tasks, while being inferior to it in a few instances. Overall, however, they are still considered poorly performing models. This observation suggests that, because the Oracle benchmark is a test under an entirely new concept, we should remain unbiased. Empirically excellent models do not necessarily achieve outstanding results, and empirically average models do not necessarily perform poorly.
>
> **In the formal benchmarking procedure, we will remove the baseline test. Any model wishing to participate can proceed directly to the Oracle benchmark.**
>
> ---
>
> ### W2: The framework for generating black-box might inherit biases from the LLMs used, potentially generating tasks that favor certain model architectures or knowledge domains. The paper lacks a rigorous analysis of this framework's reliability, potential biases, and how they might affect the benchmark's fairness and validity.
>
> ### A2:
> We analyze the black-box generation framework from two perspectives: bias and correctness.
>
> - **Bias**: **We must clarify that the bias problem does not exist.** The framework is only responsible for generating code for a black-box (i.e., the f in f: x->y). As long as the generated black-box code is correct, the black box, during its interaction with the model under evaluation, will only return the value mapped by the hidden rule (i.e., y). This return value is independent of the black-box code's specific implementation or the model that generated it (since correct code will yield the same value regardless of its origin). If the generated black-box code is erroneous, our designed framework detects the error and regenerates the code until it is correct.
>
>     In practice, for the CRI and GSI tasks, the black box returns not only the mapped value from the hidden rule but also supplementary information, such as the remaining number of interaction turns or certain game data records (in GSI). Strictly speaking, this supplementary content could potentially introduce a bias (although empirically, it does not); therefore, we conducted related experiments to address this.
>
>     **Experimental Design**: To investigate whether this supplementary information impacts the performance of the evaluation model, we used different LLMs and a human as the generators within the black-box generation framework. We selected Gemini-2.5-Pro, GPT-4.1, Claude-4-Sonnet, and a human as the generators. The models under evaluation were o3, Gemini-2.5-Pro, Claude-4-Sonnet∗, Deepseek-v3, and Gemini-2.5-Flash. We randomly selected six black boxes from both CRI and GSI tasks for the experiment. The randomly selected black boxes were: CRI (easy 1, easy 2, easy 4, hard 3, hard 6, hard 8) and GSI (easy 2, easy 3, easy 4, easy 5, hard 1, hard 5).
>
>     **Experimental Conclusion**: No significant performance difference was observed. We thus conclude that the potential bias introduced by the supplementary information does not affect the performance of the models under evaluation.
>
>     | TASK 10@1 | GENERATOR | o3 | gemini-2.5-pro | claude-4-sonnet* | deepseek-v3 | gemini-2.5-flash |
>     | :--- | :--- | :--- | :--- | :--- | :--- | :--- |
>     | CRI | gemini-2.5-pro | 66.67% | 51.67% | 50% | 30% | 18.33% |
>     | CRI | gpt-4.1 | 60% | 58.33% | 50% | 33.33% | 18.33% |
>     | CRI | claude-4-sonnet | 66.67% | 55% | 53.33% | 30% | 21.67% |
>     | CRI | human | 63.33% | 58.33% | 53.33% | 33.33% | 20.00% |
>     | GSI | gemini-2.5-pro | 41.52% | 32.55% | 38.55% | 15.01% | 10.65% |
>     | GSI | gpt-4.1 | 36.79% | 40.11% | 32.91% | 19.91% | 8.14% |
>     | GSI | claude-4-sonnet | 38.81% | 35.99% | 36.12% | 17.24% | 9.41% |
>     | GSI | human | 39.95% | 37.80% | 40.17% | 18.78% | 9.02% |
>
> - **Correctness**: We do not have a strict count of the errors that occurred during the black-box generation process, but we can provide an approximate value: **In the process of generating approximately 110 black boxes, about 5 generation errors occurred. The accuracy of automated generation is greater than 95%.** Errors mainly concentrate in three areas: 1. Extremely complex encryption processes 2. Physical systems involving differential equations 3. Games with intricate rules. **Since our designed generation framework features the output of an interaction log, humans can easily detect the errors and intervene for correction when they happen. This ultimately ensures efficient and correct black-box expansion.**

---

> ### Author Response · Authors · 2025-11-20
> **Rebuttal (Part 3)**
>
> ### W3(1): The paper compellingly identifies a key weakness: LLMs lack efficient and adaptive exploration strategies. However, the analysis primarily demonstrates that this deficiency exists rather than deeply investigating why.
>
> ### A3(1):
> Since the models that performed well on the Oracle benchmark are all closed-source, we are unable to analyze them to determine why they fail to develop efficient and adaptive strategies in black-box interaction tasks. We can only infer this conclusion by observing a large number of cases.
>
> Despite this, we still have a compelling explanation for this phenomenon. **It is well-known that the capabilities of an LLM primarily depend on the base model's pre-training and the post-training. The former involves next token prediction given a context, while the latter is an RL training on CoT prompt-based QA pairs. Neither of these training paradigms involves placing the model in an unknown environment for exploration. Consequently, the model inherently lacks the ability for strategy optimization through multi-turn interaction with an environment.**
>
> Some of the factors you mentioned (e.g., long-context reasoning, ineffective processing or utilization of feedback signals, etc.) are other reasons leading to performance degradation, but these factors are unrelated to the model's inability to develop efficient and adaptive strategies. We summarize and discuss three other model weaknesses in Appendix C.4: 1. LLMs primarily rely on pattern matching to understand black-box. 2. LLMs struggle in reasoning over dense information and long context. 3. LLMs fall short in basic computing ability.
>
> ---
>
> ### W3(2): The absence of metrics on token/turn efficiency also hinders understanding if better models achieve success through superior strategy or simply more computation/output.
>
> ### A3(2):
> We add an experiment that shows average reasoning tokens per turn. The experiments encompassed three representative tasks: Circuit Rule Inference (CRI), Encryption Rule Inference (ERI), and Physics System Inference (PSI). These were evaluated under two distinct settings: 10@1 and 30@2, and the results are presented below:
>
> | TASK 10@1 | DIFFICULTY | gemini-2.5-pro / avg token per turn | gemini-2.5-flash* / avg token per turn | o4-mini / avg token per turn |
> |:---|:---|:---|:---|:---|
> | CRI | Easy | 62% / 11700.9 | 24% / 9010.3 | 48% / 6619.4 |
> | CRI | Hard | 37% / 11844.7 | 17% / 12072.3 | 22% / 6220.8 |
> | ERI | Easy | 30.68% / 6464.8 | 6.82% / 1856.1 | 56.81% / 7203.6 |
> | ERI | Hard | 8.33% / 8208.4 | 0% / 1873.7 | 12.5% / 7878.6 |
> | PSI | Easy | 57.69% / 6460.7 | 38.46% / 5025.6 | 58.97% / 2320.0 |
> | PSI | Hard | 38.58% / 7411.9 | 37.50% / 5316.0 | 27.08% / 3354.6 |
>
> | TASK 30@2 | DIFFICULTY | gemini-2.5-pro / avg token per turn | gemini-2.5-flash* / avg token per turn | o3 / avg token per turn |
> | :---: | :---: | :---: | :---: | :---: |
> | CRI | Easy | 88.00% / 7619.4 | 44.00% / 7423.8 | 72.00% / 7654.3 |
> | CRI | Hard | 58.00% / 10206.8 | 37.00% / 9839.2 | 44.00% / 8747.6 |
> | ERI | Easy | 55.68% / 3002.2 | 6.82% / 728.9 | 68.18% / 6001.1 |
> | ERI | Hard | 2.78% / 5513.5 | 0% / 475.5 | 19.44% / 7776.4 |
> | PSI | Easy | 70.51% / 5160.2 | 55.13% / 2531.3 | 67.95% / 3502.7 |
> | PSI | Hard | 50.00% / 6256.8 | 45.83% / 3055.6 | 47.91% / 5360.3 |
>
> Two conclusions can be drawn from the results:
> 1. **Strong models can achieve success through superior strategy. But developing this strategy will cost lots of reasoning tokens.**
> 2. **Weak models cannot develop a good strategy, even with lots of reasoning tokens.** This elaborates the fundamental gap of reasoning ability between strong and weak models.
>
> Besides, we have already conducted an ablation experiment of analyzing the reasoning (extended thinking) process in Appendix E.2 Table 3. In this experiment, **we find allowing more reasoning tokens will lead to significant improvement of model accuracy.**
>
> ---
>
> ### W3(3): Moreover, the case studies don't explicitly show CoT reasoning during feedback analysis, raising questions about whether the experimental setup might inadvertently suppress models' analytical capabilities.
>
> ### A3(3):
> We show the complete reasoning process of two cases in Appendix D.3. We also add an experiment of prompting models for step-by-step reasoning about the feedback. Please refer to our reply to Q7.

---

> ### Author Response · Authors · 2025-11-20
> **Rebuttal (Part 4)**
>
> ### Q1: How does the agentic framework ensure the uniqueness, correctness, and lack of inherent bias in the generated black-boxes across different generation runs or different underlying LLMs? What steps were taken to validate the framework itself?
>
> ### A4:
> Please refer to A2.
>
> ---
>
> ### Q2: What specific, objective criteria were used to classify black-boxes as "easy" versus "hard" across the 6 different task types? How was the consistency of this difficulty scaling validated?
>
> ### A5:
> Please refer to A1(2)
>
> ---
>
> ### Q3: Is the baseline test setup (12 turns, 10 samples across 3 specific black-boxes) statistically sufficient to reliably filter models, especially given seemingly counter-intuitive outcomes like GPT-4o failing and Gemini-2.0-flash passing?
>
> ### A6:
> Please refer to A1(3)
>
> ---
> ### Q4: What is the approximate performance level of humans on the ORACLE benchmark tasks under similar constraints?
>
> ### A7:
> We add human evaluation here. It is worth noting that many black-box tasks are both time-consuming and challenging for humans. Due to cost constraints, for certain tasks, we merely randomly selected a few black boxes for testing (e.g., CII (3)). The human participants involved in the testing were undergraduate and graduate students from the departments of Mathematics, Physics, and Computer Science at several top-tier universities. The results are as follows.
>
>
> | TASK | DIFFICULTY | Human | SOTA LLM |
> | :--- | :--- | :--- | :--- |
> | CII (3) | Easy | 96% | 78% (o4-mini) |
> | CII (2) | Hard | 3% | 53% (o3) |
> | CRI (3) | Easy | 67% | 70% (o3) |
> | CRI (3) | Hard | 67% | 47% (o3) |
> | PSI (all) | Easy | 100% | 62% (o4-mini) |
> | PSI (5) | Hard | 100% | 27% (o3) |
> | ERI (all)  | Easy | 100% | 64% (o4-mini) |
> | ERI (3) | Hard | 67% | 0% (o3) |
> | IPI (all) | Easy | 100% | 83% (o3) |
> | IPI (all) | Hard | 100% | 36% (o3) |
> | GSI (all) | Easy | 88% | 79% (o3) |
> | GSI (4) | Hard | 65% | 35% (gemini-2.5-pro) |
>
> | TASK | DIFFICULTY | Human | SOTA LLM |
> | :--- | :--- | :--- | :--- |
> | CII (3) | Easy | 100% | 98% (o4-mini) |
> | CII (2) | Hard | 21% | 76% (o3) |
> | CRI (3) | Easy | 100% | 93% (o3) |
> | CRI (3) | Hard | 67% | 83% (gemini-2.5-pro) |
> | PSI (all) | Easy | 100% | 65% (o3) |
> | PSI (5) | Hard | 100% | 13% (o3) |
> | ERI (all) | Easy | 100% | 61% (o4-mini) |
> | ERI (3) | Hard | 67% | 13% (o4-mini) |
> | IPI (all) | Easy | 100% | 98% (o3) |
> | IPI (all) | Hard | 100% | 67% (o3) |
> | GSI (all) | Easy | 98% | 83% (o3) |
> | GSI (4) | Hard | 100% | 52% (o3) |
>
> **In the process of human evaluation, we summarized the following findings:**
> 1. Except for the Code Intent Inference task, humans achieved performance significantly surpassing the best-performing models on the other five categories of tasks.
> 2. During the black-box interaction process, humans demonstrated the ability to formulate highly effective adaptive and efficient exploration strategies, which enabled them to solve some black-box problems in around five turns.

---

> ### Author Response · Authors · 2025-11-20
> **Rebuttal (Part 5)**
>
> ### Q5: Can the observed performance decrease for some models/tasks when increasing turns from 10 to 20 (as hinted in Figure 8 analysis) be attributed to random fluctuation, context window limitations, error accumulation, or other factors? What trends are observed if the number of turns is increased further?
>
> ### A8:
> Model performance can be affected by random fluctuation. In Figure 8, we calculated the average increase in accuracy across 19 LLMs (represented by the blue bars). The conclusion drawn in the paper is: "... shows negligible improvement in PSI, ERI, and GSI tasks (line 415)," rather than "performance decrease for some models/tasks." This is a compelling conclusion which indicates that, generally, on tasks that heavily test exploration strategies, the model performance shows minimal increase as the number of interaction turns grows.
>
> The issue of context window limitations is absent, as the 20-turn conversation history is significantly shorter than the maximum token length accepted by the models. The problem of error accumulation is also irrelevant, as we simply extended the number of exploration turns. Furthermore, during the evaluation phase, each test sample is independent, thus precluding error accumulation.
>
> We supplemented this by conducting experiments with both more and fewer interaction turns. Due to the high testing cost (these experiments alone exceeded $1200 USD), we explored the results of four models under the 5@1 and 30@2 settings.
>
> | TASK 5@1 | DIFFICULTY | o3 | gemini-2.5-pro | gemini-2.5-flash* | deepseek-v3 |
> | :--- | :--- | :--- | :--- | :--- | :--- |
> | CII | Easy | 66.15% | 60.82% | 38.18% | 48.18% |
> | CII | Hard | 52.22% | 28.02% | 19.60% | 16.34% |
> | CRI | Easy | 24.00% | 30.00% | 14.00% | 26.00% |
> | CRI | Hard | 14.00% | 17.00% | 4.00% | 10.00% |
> | PSI | Easy | 38.46% | 58.97% | 28.21% | 19.23% |
> | PSI | Hard | 20.83% | 20.83% | 16.67% | 4.17% |
> | ERI | Easy | 52.27% | 23.86% | 6.82% | 2.27% |
> | ERI | Hard | 11.11% | 11.11% | 1.39% | 0% |
> | IPI | Easy | 46.67% | 53.44% | 30.00% | 26.67% |
> | IPI | Hard | 33.33% | 19.44% | 13.89% | 2.78% |
>
> | TASK 30@2 | DIFFICULTY | o3 | gemini-2.5-pro | gemini-2.5-flash\* | deepseek-v3 |
> | :--- | :--- | :--- | :--- | :--- | :--- |
> | CII | Easy | 92.94% | 88.96% | 66.88% | 69.91% |
> | CII | Hard | 74.21% | 55.48% | 41.59% | 31.35% |
> | CRI | Easy | 72.00% | 88.00% | 44.00% | 46.00% |
> | CRI | Hard | 44.00% | 58.00% | 37.00% | 31.00% |
> | PSI | Easy | 67.95% | 70.51% | 55.13% | 17.95% |
> | PSI | Hard | 47.91% | 50.00% | 45.83% | 12.50% |
> | ERI | Easy | 68.18% | 55.68% | 6.82% | 6.82% |
> | ERI | Hard | 19.44% | 2.78% | 0% | 0% |
> | IPI | Easy | 100.00% | 100.00% | 90.00% | 50.00% |
> | IPI | Hard | 80.56% | 63.89% | 38.89% | 0% |
>
> Meanwhile, we present the performance comparison of these four models on the PSI and ERI tasks across different interaction turns:
>
> | TASK | DIFFICULTY | o3 | gemini-2.5-pro | gemini-2.5-flash* | deepseek-v3 |
> | :--- | :--- | :--- | :--- | :--- | :--- |
> | PSI | Easy | 38.46% / 60.26% / 65.38% / 67.95% | 58.97% / 53.85% / 65.38% / 70.51% | 28.21% / 41.03% / 35.90% / 55.13% | 19.23% / 19.23% / 20.51% / 17.95% |
> | PSI | Hard | 20.83% / 43.75% / 37.50% / 47.91% | 20.83% / 41.67% / 37.50% / 50.00% | 16.67% / 31.25% / 22.92% / 45.83% | 4.17% / 2.08% / 6.25% / 12.5% |
> | ERI | Easy | 52.27% / 26.14% / 51.14% / 68.18% | 23.86% / 17.05% / 26.13% / 55.68% | 6.82% / 3.41% / 6.82% / 6.82% | 2.27% / 1.14% / 3.41% / 6.82% |
> | ERI | Hard | 11.11% / 15.28% / 5.56% / 19.44% | 11.11% / 1.39% / 16.67% / 2.78% | 1.39% / 0% / 0% / 0% | 0% / 0% / 0% / 0% |
>
> **We can draw several conclusions from the additional experiment:**
> 1. Generally speaking, more turns will not lead to significant accuracy increase for LLMs in most situations.
> 2. Accuracy can be affected by random fluctuation.
> 3. Surprisingly, in some cases (e.g., gemini-2.5-pro in ERI Easy), model performance surges when interaction turns are extended to 30. It's similar to an "aha moment". We leave it to future work to analyze. But compared to human performance, it's still inefficient.

---

> ### Author Response · Authors · 2025-11-20
> **Rebuttal (Part 6)**
>
> ### Q6: Was the token usage or computational cost per turn analyzed? Do higher-performing models demonstrate greater efficiency in their exploration?
>
> ### A9:
> Please refer to A3(2)
>
> ---
>
> ### Q7: In the comparative experiment on adaptive exploration, is there evidence (or lack thereof) that models attempt to analyze feedback even if their strategy doesn't change? Could the lack of explicit prompting for step-by-step reasoning about the feedback in the interaction loop negatively impact their ability to adapt their strategy?
>
> ### A10:
> **For the first question, the answer is yes.** We check the reasoning content of gemini-2.5-pro, and confirm that LLM try to analyze feedback (despite failure in developing efficient and adaptive strategy).
>
> **For the second question, we add an experiment that explicitly instructing LLMs to reflect prior feedback and keep an efficient exploration strategy at the end of each turn, and see if their performance gets improved.** We pick gemini-2.5-pro, gemini-2.5-flash_thinking, o3 in Circuit Rule Inference and Encryption Rule Inference task under 10@1. The results are shown below.
>
> | Feedback Reflection | TASK 10@1 | DIFFICULTY | gemini-2.5-pro | gemini-2.5-flash* | o3 |
> | :--- | :--- | :--- | :--- | :--- | :--- |
> | with / without | CRI | Easy | 66.00% / 62.00% | 18.00% / 24.00% | 58.00% / 64.00% |
> | with / without | CRI | Hard | 34.00% / 37.00% | 12.00% / 17.00% | 21.00% / 24.00% |
> | with / without | ERI | Easy | 25.00% / 30.68% | 9.09% / 6.82% | 32.95% / 26.14% |
> | with / without | ERI | Hard | 13.89% / 8.33% | 0% / 0% | 2.78% / 15.28% |
>
> **The above results indicate that the LLM's lack of an efficient and adaptive exploration strategy is a fundamental flaw, which cannot be compensated for by CoT prompting.**
>
> ---
>
> ### Q8: Beyond identifying the lack of adaptive exploration strategies, what specific experiments or analyses could pinpoint the underlying causes?
>
> ### A11:
> Please refer to A3(1)
>
> ---
>
> ### Q9: Recent works, such as KORGym, have demonstrated the lack of planning capabilities in models through multi-turn environmental interactions, yet relevant discussion is missing.
>
> ### A12:
> We apologize for the oversight regarding the related work. We will include the necessary citations and discussion in the revised version.
>
> It is crucial to note that the lack of planning ability in LLMs, as proposed by prior work (e.g., Natural Plan: Benchmarking LLMs on Natural Language Planning), is fundamentally different from the lack of an efficient and adaptive exploration strategy in black-box interaction, which is the focus of our study. **The task setting for planning typically involves finding the optimal path among multiple possible routes to complete a specific task, which is distinct from our proposed task of exploring an unknown environment.** We focus on building an interactive, unknown environment with hidden rules and investigating LLMs' behavior and performance in this environment. Furthermore, we did not find experimental analysis or discussion regarding the deficiency in planning ability in the KORGym paper.
>
> In the current version of our paper, we have discussed the distinction between black-box interaction and game-based benchmarks (e.g., Minecraft) in the Introduction (lines 043-048):
> > Some researches (Costarelli et al., 2024; Hu et al., 2024) employ games to simulate interactive, unknown environments. This approach presents two key limitations. First, the extensive training data of LLMs raises the possibility that they are already familiar with the game strategy, compromising the validity of testing reasoning in unknown environment. Second, it conflates the evaluation of reasoning with other abilities like spatial understanding and long-context understanding, preventing it from serving as a pure reasoning benchmark.
>
> In Appendix C.2, we discuss the distinction between black-box interaction and text-game-like benchmarks:
> > Black-box interaction differs from these works in three aspects: First, we focus on building an interactive, unknown environment with hidden rules and investigating LLMs behavior in this environment, instead of testing LLMs’ performance in playing specific games. Second, some of the text games can be viewed as a subset of Interactive Puzzle Inference in the ORACLE benchmark, and our proposed black-box interaction approach can easily scale the Interactive Puzzle Inference task. Third, all chosen text games in previous works are well-known, which increases the possibility that LLMs are already familiar with the exploration strategy.
>
> The difference between KORGym as a game benchmark and our work is included in the discussion above.
>
> ---
>
> **We will add the above content to the revised version of paper. Thanks again for giving a positive review. If you find our clarification helpful, we sincerely hope you may consider raising the rating. We are also very glad for further discussion.**

---

> ### Author Response · Authors · 2025-11-25
>
> Dear Reviewer HrU1:
>
> Thank you for your thoughtful review. We have addressed your concerns in detail in the rebuttal and would appreciate it if you could revisit our responses. Please let us know if any concerns remain. If the rebuttal satisfactorily addresses your points, we kindly ask you to consider reflecting this in your overall rating.
>
> Best
>
> Authors

---

> ### Author Response · Authors · 2025-11-27
>
> Dear Reviewer HrU1:
>
> As the deadline of rebuttal is approaching, we sincerely hope you can participate in the discussion and read our latest clarification. Thank you.
>
> Best
>
> Authors

---

### Official Review · Reviewer_JC5B · 2025-10-31

**Soundness:** 3
**Presentation:** 4
**Contribution:** 3
**Rating:** 6
**Confidence:** 3

**Summary:**

This paper introduces a new task for LLMs, called black-box interaction, to assess the exploration planning and reasoning ability in an unknown environment. The process mirrors C·P's framework of human reasoning—abduction (hypothesis formation), deduction (testing), and induction (refinement)—thereby evaluating reasoning as a dynamic and adaptive cycle rather than as isolated tasks.

Building on this paradigm, the authors develop the ORACLE benchmark, which includes 96 black-boxes spanning six task types: Code Intent Inference, Circuit Rule Inference, Physics System Inference, Encryption Rule Inference, Interactive Puzzle Inference, and Game Strategy Inference. Then they evaluate 19 modern LLMs.

The paper’s contributions are:

1. introducing the black-box interaction paradigm,
2. constructing the scalable ORACLE benchmark,
3. proposing an automated black-box generation framework, and
4. providing comprehensive analyses that expose current LLMs’ limits in advanced reasoning.

**Strengths:**

1. The black-box interaction paradigm is intuitive and novel.

2. The paper is well written and easy to understand.

3. Experiment details are shown in the appendix.

4. The analysis is valuable to share with the community, giving readers a better understanding of LLM reasoning.

**Weaknesses:**

1. My major concern is that, since the 6 types are hand-designed, how isolate the reasoning ability and the prior knowledge (like Circuit Rule).

2. The difficulty of the task is hard to understand. I think it is necessary to use some metrics to depict the difficulty of the tasks. (e.g., the minimal exploration steps for an expert to determine the answer)

**Questions:**

1. What is the accuracy of Random for each task?

2. L472, Figure 10, "Model and human performance ...", where is the human?

---

> ### Author Response · Authors · 2025-11-20
> **Rebuttal (Part 1)**
>
> Thanks for reviewing our paper and providing valuable feedback. We will make clarifications point by point.
>
> ### W1: Since the 6 types are hand-designed, how isolate the reasoning ability and the prior knowledge (like Circuit Rule)?
>
> ### A1:
> Currently, all the knowledge involved in the black-boxes within the Oracle benchmark is certainly memorized by LLMs, yet when it’s presented in the form of black-box interaction, the performance of most models is poor. A powerful piece of evidence is that when faced with even extremely simple baseline tests, models lacking reasoning ability (e.g., gpt-4o) fail to meet the passing threshold.
>
> By hiding simple rules and constructing an interactive environment, **we can differentiate models with genuine reasoning capabilities from those that merely rely on memorization and recitation of relevant knowledge**. This is precisely the greatest feature of Black-Box Interaction and its advantage over traditional reasoning benchmarks (as discussed in the first paragraph of Appendix C.1). Any task design that adheres to the Black-Box Interaction concept possesses this feature, regardless of its specific design details or whether it is hand-designed.
>
> To further address your concern, we design an experiment that directly adds domain background knowledge into the input instruction and see if the performance gets improved. Two representative tasks, Circuit Rule Inference (CRI) and Encryption Rule Inference (ERI) and two representative LLMs, gemini-2.5-pro (reasoning model) and gemini-2.5-flash (non-reasoning model) are picked for experiment.
>
> | Background Knowledge | TASK 10@1 | Difficulty | gemini-2.5-pro | gemini-2.5-flash |
> | :---: | :---: | :---: | :---: | :---: |
> | with / without | CRI | Easy | 50% / 62% | 22% / 20% |
> | with / without | CRI | Hard | 30% / 37% | 11% / 10% |
> | with / without | ERI | Easy | 38.64% / 30.68% | 4.55% / 3.41% |
> | with / without | ERI | Hard | 5.56% / 8.33% | 0% / 0% |
>
> **As shown in the results, adding background knowledge explicitly will not lead to better performance for both reasoning and non-reasoning models. The key factor for achieving a high score is the model's strong reasoning ability.**
> The specific background knowledge prompt is in Rebuttal Part 3. It is noteworthy that the input background knowledge even contained some black-box design details. However, even with this prerequisite, explicitly adding this background knowledge still fails to improve the model's performance. Therefore, background knowledge will not affect the evaluation of reasoning ability in the Oracle benchmark.
>
> ---
>
> ### W2: The difficulty of the task is hard to understand.
>
> ### A2:
> It’s hard to quantify the difficulty of each black-box. Using the minimal exploration steps for an expert to determine the answer is a good suggestion. Sadly, it’s beyond our capacity to find enough experts. The benchmark is challenging and time-consuming for humans as well.
>
> The current division of tasks into easy and hard is based on two points:
>
> 1. **Human understanding of black-box difficulty**. Four designers ranked the black-box difficulty of each task based on a partial order relationship. They ultimately discussed and collectively determined the sets for easy and hard tasks.
>
> 2. **The black-box's inherent difficulty**. In the beginning of each subsection in Appendix G, we detail the basis for the difficulty division:
>
>     - (Code Intent Inference) Easy black-boxes cover common, simple algorithms (e.g., bubble sort, Fibonacci). Hard black-boxes cover complex, less common algorithms (e.g., heapsort, Sieve of Eratosthenes).
>
>     - (Circuit Rule Inference) Easy black-boxes mainly feature circuits with a sequence structure and those that implement basic functions (e.g., XOR). Hard black-boxes mainly feature circuits with a tree structure and those that implement more complex functions (e.g., ADD).
>
>     - (Physics System Inference) Easy black-boxes focus on fundamental motion involving one or two objects (e.g., projectile motion, collision). Hard black-boxes involve two or more objects or incorporate more complex mechanical systems (e.g., three-ball collision, damped simple harmonic motion).
>
>     - (Encryption Rule Inference) Easy black-boxes feature ciphers that do not change the frequency of characters in the plaintext. For hard black-boxes, the encrypted letters barely retain their original frequencies.
>
>     - (Interactive Puzzle Inference) Easy black-boxes are puzzles with simple rules and limited action space, while hard black-boxes have more complex rules and a much larger range of actions to choose from.
>
>     - (Game Strategy Inference) Easy black-boxes are typically with fixed game strategy or games with easy rules, while hard black-boxes are with dynamic game strategy or games with difficult rules.
>
> From the experimental results, the current division of easy and hard is reasonable. Almost all models perform significantly better on easy setting than on hard setting.

---

> ### Author Response · Authors · 2025-11-20
> **Rebuttal (Part 2)**
>
> ### Q1: What is the accuracy of Random for each task?
>
> ### A3:
> For Code Intent Inference (CII), Physics System Inference (PSI), Encryption Rule Inference (ERI) tasks, the result of a random answer will approach zero infinitely, since the action space is infinite. For Circuit Rule Inference (CRI), the accuracy for a random answer is 1/2^n, where n is the number of circuit output gates, typically above 8. For the Interactive Puzzle Inference (IPI) and Game Strategy Inference (GSI) tasks, the accuracy for a random answer depends on specific puzzle/game setting. For instance, in the heavy coin puzzle with 100 coins (IPI Easy 4), the accuracy for a random answer is 1/100. Since multiple test samples are designed for one black-box, we can guarantee that the probability of a false or random inference resulting in high accuracy is close to 0.
>
> ---
>
> ### Q2: L472, Figure 10, "Model and human performance ...", where is the human?
>
> ### A4:
> It’s a typo. There’s no human performance in this experiment. We will fix it. Thanks for pointing it out.
>
> ---
>
> **We will add the above content to the revised version of paper. Thanks again for giving a positive review. If you find our clarification helpful, we sincerely hope you may consider raising the rating. We are also very glad for further discussion.**

---

> ### Author Response · Authors · 2025-11-20
> **Rebuttal (Part 3)**
>
> We put the detailed prompt of prior background knowledge here.
>
> ### Circuit Rule Inference:
>
> Domain Knowledge:
> - Here is some domain knowledge that might help:
>     - NOT Gate: The most basic logic gate, also called an inverter. Its function is to reverse the input signal: if the input is 1 (High), the output is 0 (Low); if the input is 0, the output is 1.
>     - AND Gate: Implements the logical "AND" operation. The output is 1 only when all of its inputs are 1; otherwise, the output is 0.
>     - OR Gate: Implements the logical "OR" operation. The output is 1 as long as any of its inputs are 1; the output is 0 only when all inputs are 0.
>     - Boolean Algebra: This is the mathematical foundation for describing and analyzing logic circuits. AND, OR, and NOT correspond to Boolean multiplication (⋅), addition (+), and complement (').
>     - Transistor Implementation: In modern electronics, these logic gates (AND, OR, NOT) are typically not discrete components but are built using combinations of transistors (MOSFETs) on integrated circuits (ICs) using CMOS technology.
>     - Functional Completeness: Using only AND, OR, and NOT gates, it is possible to construct any complex logic function, making them functionally complete.
>     - NAND Gate: Equivalent to an AND gate followed by a NOT gate. It is very important because the NAND gate itself is "functionally complete," meaning all other logic functions can be implemented using only NAND gates.
>     - NOR Gate: Equivalent to an OR gate followed by a NOT gate. Similar to NAND, the NOR gate is also "functionally complete."
>     - XOR Gate (Exclusive OR): Implements the "exclusive OR" function (output is 1 when the two inputs are different). While not one of the most basic gates, it is very common and can be constructed from a combination of AND, OR, and NOT gates.
>     - Half-Adder: This is the foundation for implementing addition. It consists of one XOR gate (to calculate the "Sum") and one AND gate (to calculate the "Carry-out"), used for adding two single binary bits.
>     - Full-Adder: This is a more practical addition unit. It is built using two Half-Adders and one OR gate. It can process three inputs (two addend bits and one "Carry-in" from a lower bit) and produces a "Sum" and a "Carry-out" to the next bit.
>     - Ripple-Carry Adder (Multi-bit Adder): By chaining multiple "Full-Adders" together (connecting the Carry-out of one to the Carry-in of the next), it is possible to perform addition for multi-bit binary numbers (like 8-bit or 16-bit).
>     - Multiplexer (MUX): Acts like a "digital switch." It uses AND, OR, and NOT gates to select one of several inputs to be the single output, based on "select signals."
>     - Decoder: Its function is the opposite of a MUX. It takes an N-bit binary input (like an address) and activates exactly one of its 2^N outputs (e.g., for selecting a memory cell).

---

> ### Author Response · Authors · 2025-11-20
> **Rebuttal (Part 4)**
>
> We put the detailed prompt of prior background knowledge here.
>
> ### Encryption Rule Inference:
> - Here is some domain knowledge that might help:
>     - The Two Pillars: Substitution & Transposition Classical cryptography primarily relies on two main operations: Substitution (replacing one letter with another) and Transposition (rearranging the order of the letters).
>     - Caesar Cipher The simplest substitution cipher. It shifts the alphabet by a fixed number of positions (e.g., A becomes D, B becomes E, with a key of 3).
>     - Cracking Caesar: Brute Force Since the English alphabet has 26 letters, there are only 25 possible keys (shifts). An attacker can simply try all 25 shifts until the plaintext becomes readable.
>     - Simple Substitution Cipher Establishes a fixed, jumbled mapping for the alphabet (e.g., A->Q, B->X...). The key is this entire mapping. It is much more complex than the Caesar cipher.
>     - The Core Weakness: Frequency Analysis This is the most powerful weapon against simple substitution ciphers. It involves counting the frequency of each letter in the ciphertext and comparing it to the known letter frequencies of the plaintext language (e.g., in English, 'E' is the most common, followed by T, A, O) to deduce the mapping.
>     - Atbash Cipher A special type of simple substitution where the alphabet is reversed (A<->Z, B<->Y...). It has no key; knowing the algorithm is enough to decrypt it.
>     - Vigenère Cipher A famous polyalphabetic substitution cipher. It uses a keyword (e.g., "KEY") to cyclically select different Caesar ciphers, effectively hiding the simple letter frequencies and resisting basic frequency analysis.
>     - Cracking Vigenère: Kasiski Examination A method to deduce the key length of a Vigenère cipher. It works by finding repeated strings of letters in the ciphertext and calculating the greatest common divisor (GCD) of the distances between them.
>     - Cracking Vigenère: Index of Coincidence (IC) Another statistical method to estimate the key length. By calculating the IC of the ciphertext (which measures how different letter frequencies are from a uniform distribution), one can determine the most likely key length.
>     - Divide and Conquer Once the Vigenère key length (N) is found, the ciphertext can be split into N separate "columns." Each column is effectively encrypted with its own simple Caesar cipher, which can then be broken individually using frequency analysis.
>     - One-Time Pad (OTP) The only theoretically unbreakable cipher. It is a Vigenère cipher, but the key must be truly random, as long as the plaintext, and used only once.
>     - Transposition Ciphers These ciphers do not change the letters, only their positions. Common examples include the Rail Fence Cipher (writing text in a zigzag pattern) and Columnar Transposition (rearranging columns of a grid based on a keyword).
>     - Cracking Transposition: Anagramming The letter frequencies of the ciphertext are identical to the plaintext. Cracking often relies on anagramming—rearranging letters to form likely words, often helped by guessing probable words ("cribs") or analyzing common letter combinations (digraphs/trigraphs).
>     - Playfair Cipher The first practical digraphic substitution cipher. It uses a 5x5 keyed grid to encrypt letters in pairs, which hides single-letter frequencies and makes it harder to break than simple substitution.
>     - Hill Cipher A polygraphic substitution cipher that uses linear algebra (matrix multiplication). It is highly vulnerable to a Known-Plaintext Attack. An attacker only needs a few plaintext-ciphertext pairs to set up and solve a system of linear equations to recover the key matrix.

---

> ### Author Response · Authors · 2025-11-25
>
> Dear Reviewer JC5B:
>
> Thank you for your thoughtful review. We have addressed your concerns in detail in the rebuttal and would appreciate it if you could revisit our responses. Please let us know if any concerns remain. If the rebuttal satisfactorily addresses your points, we kindly ask you to consider reflecting this in your overall rating.
>
> Best
>
> Authors

---

> ### Author Response · Authors · 2025-11-27
>
> Dear Reviewer JC5B:
>
> As the deadline of rebuttal is approaching, we sincerely hope you can participate in the discussion and read our latest clarification. Thank you.
>
> Best
>
> Authors

---

### Author Response · Authors · 2025-12-01
**Summary of Rebuttal**

We summarize the rebuttal here. The reviewers' comments are categorized into three types: 1. Suggestions for Improvement (we make additional experiments and revise our paper), 2. Questions (we answer the raised questions), and 3. Misunderstandings (we make clarifications). We use W (weakness) and Q (question) for quick reference of our detailed response to each reviewer.
The submitted paper is updated based on the rebuttal. Sadly, while we believe we have sufficiently addressed all raised problems, reviewer JC5B and HrU1 did not respond.

### Suggestions for Improvement
- **(JC5B-W1) The influence of prior knowledge to reasoning.**

    We add experiment to show prior knowledge won't help. The Oracle benchmark tests pure reasoning.

- **(HrU1-W1(3), HrU1-Q3) The adequacy of the baseline test is questionable.**

    We test performance of gpt-4o and gpt-4.1. The baseline test is not entirely rigorous, but it can still filter poorly performing models. The baseline test was primarily designed for cost reduction. We will remove it in the formal benchmarking procedure.

- **(HrU1-Q9, NwnN-W1) Discuss some related work.**

    We supplement discussion.

- **(HrU1-W3(1), HrU1-Q8) The paper fail to tell why LLMs lack efficient and adaptive exploration strategies.**

    Since the models that performed well on the Oracle benchmark are all closed-source, we are unable to analyze the "why" question in detail. But we still have an explanation. Neither pre-train nor post-train involves placing the model in an unknown environment for exploration. Consequently, the model inherently lacks the ability for strategy optimization through multi-turn interaction with an environment.

- **(HrU1-W3(2), HrU1-Q6) The absence of metrics on token/turn efficiency.**

    We add an experiment that shows average reasoning tokens per turn.

- **(HrU1-W3(3), HrU1-Q7) Can CoT prompt improve model performance?**

    No. We add an experiment that shows the LLM's lack of an efficient and adaptive exploration strategy is a fundamental flaw, which cannot be compensated for by CoT prompting.

- **(HrU1-Q4) The performance of human in the benchmark.**

    We add human evaluation.

- **(HrU1-Q5) The influence of fewer and more exploration turns.**

    We supplement the results of 5 and 30 exploration turns.

### Questions

- **(JC5B-Q1) The accuracy of random behavior in Oracle benchmark.**

    We analyze it in detail.

### Misunderstandings

- **(HrU1-W1(1)) The Oracle benchmark fail to simulate real-world environment.**

    We make clarifications. The six tasks are all an abstraction of real-world tasks.

- **(JC5B-W2,HrU1-W1(2), HrU1-Q2) The criteria for easy and hard black-boxes are not defined.**

    The relevant details are already noted in the original paper. We make clarifications.

- **(HrU1-W2, HrU1-Q1) The paper lacks analysis of the black-box generation framework's reliability and potential biases.**

    The bias problem does not exist. The accuracy of automated generation is greater than 95%. Since the designed generation framework features the output of an interaction log, humans can easily detect the errors and intervene for correction when they happen. This ultimately ensures efficient and correct black-box expansion.

---

### Meta-Review · Area_Chair_ZANA · 2026-01-04

**Summary:**

This paper introduces a benchmark with the black-box interaction paradigm to evaluate the ability of LLMs to “understand” black-box functions by interacting with them. “Understanding” means the LLM can achieve high accuracy on a test set of black-box function input, output pairs. The benchmark includes 6 types of tasks such as coding intent, circuit rules, puzzles etc.


Reviewer JC5B and Reviewer HrU1 both raised questions about quantifying the difficulty of the tasks. The authors responded that it’s hard to quantify the difficulty but provided high level “qualitative” descriptions. I do not think the responses directly addressed the concern since there is still no clear quantification. Potentially, the authors could have investigated quantities like the function compositions, feedback complexity etc.

I think another important concern is insufficient investigation of “why” (Reviewer HrU1).
It’d be good to include deeper analyses to provide clearer understanding of why the models do or do not perform well. The paper starts off with good decompositions of the reasoning abilities involved in solving the tasks in the benchmark (abduction, deduction, induction) and it would be natural to understand those in the experiments too. I think the paper can also be strengthened by providing clearer relations among the tasks. For example, do some tasks measure abduction more than the others? The mixed results also leave more doubts. I think having more analyses on why a “less capable” model like gemini-2.0-flash may perform better than “more capable” models such as gpt-4.1. Analyses like this can offer valuable insights into model abilities. Simply showing the accuracy does not quite build new knowledge for the readers.

Given those outstanding concerns I recommend rejection.

Related work should appear earlier in the paper, ideally after the introduction.

**Reviewer Concerns:**

Reviewer JC5B also raised concerns about the isolation of reasoning and knowledge. The authors performed additional experiments to see whether adding knowledge makes models perform better or not. The results are mixed. For some models and tasks, there exist improvements but there are not for others. I think the authors could have offered deeper analyses into what causes the performance differences since the current results do not offer much insights to address this particular problem.

Reviewer HrU1 provided a very detailed review and shared concerns on the benchmarks’s resemblance of real-world reasoning (outstanding concern), reliability of the generated black-boxes given the heavy reliance on LLM agents (outstanding concern), insufficient investigation of “why” (outstanding concern; see my takeaway in the summary), lack of discussions on related work (addressed). The authors described the tasks in more detail to show that they are real-world tasks. However, I did not find the argument very convincing. These tasks are carefully constructed “puzzles” and it is hard to say whether they are real world usages of LLMs without comparing those tasks with the actual distribution over LLM use cases. While the concern was not addressed, I do think these are still interesting tasks to benchmark specific abilities of LLMs. About reliability, the authors analyzed bias and correctness, claiming that bias does not exist because the framework generates the code for the black-box and can detect errors in the code. I am worried that we still do not know how well the framework can detect errors or how to know whether the error detection process is correct.

Reviewer NwnN provided a very short review. The concern is about the discussion of related work and the authors added the discussion. While the reviewer stated a high confidence level, I believe that it is reasonable to discount it given how short the review is.

**Reviewer Scores:**

It'd be unlikely for Reviewer JC5B and Reviewer HrU1 to update their scores.

Reviewer NwnN might increase the score, but I would like to discount this review given how short it is.

---

### Decision · Program_Chairs · 2026-01-26

Reject